# Enhancing Low-Precision Sampling via Stochastic Gradient Hamiltonian Monte Carlo

**Ziyi Wang** *wang4538@purdue.edu*
*Department of Statistics*
*Purdue University*

**Yujie Chen** *chen1866@purdue.edu*
*Department of Statistics*
*Purdue University*

**Qifan Song**[*] *qfsong@purdue.edu*
*Department of Statistics*
*Purdue University*

**Ruqi Zhang**[*] *ruqiz@purdue.edu*
*Department of Computer Science*
*Purdue University*

Reviewed on OpenReview: *https://openreview.net/forum?id=uSLNzzuiDJ*

## Abstract

Low-precision training has emerged as a promising low-cost technique to enhance the training efficiency of deep neural networks without sacrificing much accuracy. Its Bayesian counterpart can further provide uncertainty quantification and improved generalization accuracy. This paper investigates low-precision sampling via Stochastic Gradient Hamiltonian Monte Carlo (SGHMC) with low-precision and full-precision gradient accumulators for both strongly log-concave and non-log-concave distributions. Theoretically, our results show that to achieve $\epsilon$-error in the 2-Wasserstein distance for non-log-concave distributions, low-precision SGHMC achieves quadratic improvement $(\tilde{\mathcal{O}}\left(\epsilon^{-2}\mu^{*-2}\log^2\left(\epsilon^{-1}\right)\right))$ compared to the state-of-the-art low-precision sampler, Stochastic Gradient Langevin Dynamics (SGLD) $(\tilde{\mathcal{O}}\left(\epsilon^{-4}\lambda^{*-1}\log^5\left(\epsilon^{-1}\right)\right))$. Moreover, we prove that low-precision SGHMC is more robust to the quantization error compared to low-precision SGLD due to the robustness of the momentum-based update w.r.t. gradient noise. Empirically, we conduct experiments on synthetic data, and MNIST, CIFAR-10 & CIFAR-100 datasets, which validate our theoretical findings. Our study highlights the potential of low-precision SGHMC as an efficient and accurate sampling method for large-scale and resource-limited machine learning.

## 1 Introduction

In recent years, while deep neural networks (DNNs) stand out for their state-of-art performance across various AI tasks, accompanied by an increase in model and computation complexity (Simonyan & Zisserman, 2014; He et al., 2016; Vaswani et al., 2017; Radford et al., 2018; Chen et al., 2023). Addressing this challenge, techniques that enable efficient DNN processing become imperative to enhance efficiency and facilitate the widespread deployment of DNNs in AI systems (Courbariaux et al., 2015; Sze et al., 2017). Consequently, there is a growing interest in utilizing low-precision optimization techniques to address the computational and memory costs associated with these complex models (Sze et al., 2017). By employing reduced precision for

---

[*]Equal advising

Table 1: Theoretical results of the achieved 2-Wasserstein distance and the required gradient complexity for both log-concave (*italic*) and non-log-concave (**bold**) target distributions, where $\epsilon$ is any sufficiently small constant, $\Delta$ is the quantization error, and $\mu^*$ and $\lambda^*$ denote the contraction rate of underdamped and overdamped Langevin dynamics respectively (Definition 1). Under non-log-concave target distributions, low-precision SGHMC achieves a better upper bound within shorter iterations compared with low-precision SGLD.

| | Condition | Gradient Complexity | Achieved 2-Wasserstein |
|---|---|---|---|
| Full-precision gradient accumulators | | | |
| *SGLD/SGHMC* (Theorem 4) | *Strongly log-concave* | $\tilde{\mathcal{O}}\left(\log\left(\epsilon^{-1}\right)\epsilon^{-2}\right)$ | $\tilde{\mathcal{O}}\left(\epsilon+\Delta\right)$ |
| **SGLD** (Theorem 7) | **Non-log-concave** | $\tilde{\mathcal{O}}\left(\epsilon^{-4}\lambda^{*-1}\log^5\left(\epsilon^{-1}\right)\right)$ | $\tilde{\mathcal{O}}\left(\epsilon+\log\left(\epsilon^{-1}\right)\sqrt{\Delta}\right)$ |
| **SGHMC** (Theorem 1) | **Non-log-concave** | $\tilde{\mathcal{O}}\left(\epsilon^{-2}\mu^{*-2}\log^2\left(\epsilon^{-1}\right)\right)$ | $\tilde{\mathcal{O}}\left(\epsilon+\sqrt{\log\left(\epsilon^{-1}\right)\Delta}\right)$ |
| Low-precision gradient accumulators | | | |
| *SGLD/SGHMC* (Theorem 5) | *Strongly log-concave* | $\tilde{\mathcal{O}}\left(\log\left(\epsilon^{-1}\right)\epsilon^{-2}\right)$ | $\tilde{\mathcal{O}}\left(\epsilon+\epsilon^{-1}\Delta\right)$ |
| *VC SGLD/VC SGHMC* (Theorem 6) | *Strongly log-concave* | $\tilde{\mathcal{O}}\left(\log\left(\epsilon^{-1}\right)\epsilon^{-2}\right)$ | $\tilde{\mathcal{O}}\left(\epsilon+\sqrt{\Delta}\right)$ |
| **SGLD** (Theorem 8) | **Non-log-concave** | $\tilde{\mathcal{O}}\left(\epsilon^{-4}\lambda^{*-1}\log^5\left(\epsilon^{-1}\right)\right)$ | $\tilde{\mathcal{O}}\left(\epsilon+\log^5\left(\epsilon^{-1}\right)\epsilon^{-4}\sqrt{\Delta}\right)$ |
| **VC SGLD** (Theorem 9) | **Non-log-concave** | $\tilde{\mathcal{O}}\left(\epsilon^{-4}\lambda^{*-1}\log^3\left(\epsilon^{-1}\right)\right)$ | $\tilde{\mathcal{O}}\left(\epsilon+\log^3\left(\epsilon^{-1}\right)\epsilon^{-2}\sqrt{\Delta}\right)$ |
| **SGHMC** (Theorem 2) | **Non-log-concave** | $\tilde{\mathcal{O}}\left(\epsilon^{-2}\mu^{*-2}\log^2\left(\epsilon^{-1}\right)\right)$ | $\tilde{\mathcal{O}}\left(\epsilon+\log^{3/2}\left(\epsilon^{-1}\right)\epsilon^{-2}\sqrt{\Delta}\right)$ |
| **VC SGHMC** (Theorem 3) | **Non-log-concave** | $\tilde{\mathcal{O}}\left(\epsilon^{-2}\mu^{*-2}\log^2\left(\epsilon^{-1}\right)\right)$ | $\tilde{\mathcal{O}}\left(\epsilon+\log\left(\epsilon^{-1}\right)\epsilon^{-1}\sqrt{\Delta}\right)$ |

both model and data representations ( e.g. mixed-precision, low-bits fixed-point, low-bit block floating point), significant improvements can be achieved in terms of DNN training speed and resource efficiency (Micikevicius et al., 2017; Gupta et al., 2015; Li et al., 2017; De Sa et al., 2017; Zhou et al., 2016). Notably, several recent studies (Wang et al., 2018; Banner et al., 2018; Wu et al., 2018; Lin et al., 2019; Sun et al., 2019) demonstrated the successful application of 8-bit training techniques in accelerating the training of different models, such as VGG (Wu et al., 2018), ResNet (Banner et al., 2018), LSTMs, Transformers (Sun et al., 2019), and vision-language models (Wortsman et al., 2023).

With the increasing demand and huge success of complicated architecture such as Large Languages Models (LLMs) and Vision-transformers, a wide range of quantization methods are adopted to reduce the computing and memory consumption while retaining acceptable performances(Liu et al., 2023; Zhao et al., 2023; Li et al., 2023; Xu et al., 2023; Xiao et al., 2023). Readers can find more information on low-precision optimization and model compression in the survey papers (Sze et al., 2017; Deng et al., 2020; Liang et al., 2021).

As a counterpart of low-precision optimization, low-precision sampling is relatively unexplored but has shown promising preliminary results. Zhang et al. (2022) studied the effectiveness of Stochastic Gradient Langevin Dynamics (SGLD) (Welling & Teh, 2011) in the context of low-precision arithmetic, highlighting its superiority over the optimization counterpart, Stochastic Gradient Descent (SGD). This superiority stems from SGLD's inherent robustness to system noise compared with SGD.

Other than SGLD, Stochastic Gradient Hamiltonian Monte Carlo (SGHMC) (Chen et al., 2014) is another popular gradient-based sampling method closely related to the underdamped Langevin dynamics. Recently, Cheng et al. (2018); Gao et al. (2022) showed that SGHMC converges to its target distribution faster than the best-known convergence rate of SGLD in the 2-Wasserstein distance under both strongly log-concave and non-log-concave assumptions. Beyond this, SGHMC is analogous to stochastic gradient methods augmented with momentum, which is shown to have more robust updates w.r.t. gradient estimation noise (Liu et al., 2020). Since the quantization-induced stochastic error in low-precision updates acts as extra gradient noise, we believe SGHMC is particularly suited for low-precision arithmetic.

Our main contributions in this paper are threefold:

- We conduct the first study of low-precision SGHMC, adopting the low-precision arithmetic (including full- and low-precision gradient accumulators and the variance correction (VC) version of low-precision gradient accumulators) to SGHMC.

- We present a thorough theoretical analysis of low-precision SGHMC for both strongly log-concave and non-log-concave target distributions. Beyond Zhang et al. (2022)'s analysis for strongly log-concave distributions, we introduce an intermediate process for quantization noise management to facilitate the analysis of non-log-concave target distributions. All our theoretical results are summarized in Table 1, where we compare the 2-Wasserstein convergence limit and the required gradient complexity. The table highlights the superiority of HMC-based low-precision algorithms over SGLD counterpart w.r.t. convergence speed and robustness to quantization error, especially under the non-log concave distributions.

- We provide promising empirical results across various datasets and models. We show the sampling capabilities of HMC-based low-precision algorithms and the effectiveness of the VC function in both strongly log-concave and non-log-concave target distributions. We also demonstrate the superior performance of HMC-based low-precision algorithms compared to SGLD in deep learning tasks. Our code is available here.

In summary, low-precision SGHMC emerges as a compelling alternative to standard SGHMC due to its ability to enhance speed and memory efficiency without sacrificing accuracy. These advantages position low-precision SGHMC as an attractive option for efficient and accurate sampling in scenarios where reduced precision representations are employed.

It is worth mentioning that low-precision gradient representations are also used in Federated Learning (FL) for either optimization (Gorbunov et al., 2021; Tyurin & Richtárik, 2022) or sampling tasks (Vono et al., 2022; Sun et al., 2022; Karagulyan & Richtárik, 2023). These methods use low-precision representations for between-node communication, aiming to mitigate communication bottlenecks, but still utilize full-precision arithmetic for local training. Thus these methods do not apply to the low-precision sampling challenge studied in this paper.

## 2 Preliminaries

### 2.1 Low-Precision Quantization

Two popular low-precision number representation formats are known as the *fixed point* (FP) and *block floating point* (BFP) (Song et al., 2018). Theoretical investigation of this paper only considers the fixed point case, where the quantization error (i.e., the gap between two adjacent representable numbers) is denoted as $\Delta$. For example, if we use 8 bits to represent a number where 1 bit is assigned for the sign, 2 bits for the integer part, and 5 bits for the fractional part, then the gap between two consecutive low-precision numbers is $2^{-5}$, i.e., $\Delta = 2^{-5}$. Furthermore, all representable numbers are truncated to an upper limit $\bar{U}$ and a lower limit $\bar{L}$.

Given the low-precision number representation, a quantization function is desired to round real-valued numbers to their low-precision counterparts. Two common quantization functions are *deterministic rounding* and *stochastic rounding*. The deterministic rounding function, denoted as $Q^d$, quantizes a number to its nearest representable neighbor. The stochastic rounding, denoted as $Q^s$ (refer to (19) in Appendix D), randomly quantizes a number to its close representable neighbor satisfying the unbiased condition, i.e. $\mathbb{E}[Q^s(\theta)] = \theta$. In what follows, $Q_W$ and $Q_G$ denote stochastic rounding quantizers for the weights and gradients respectively, allowing different quantization errors (i.e., different $\Delta$'s for $Q_W$ and $Q_G$). For simplicity in the analysis and experiments, we use the same number of bits to represent the weights and gradients.

## 2.2 Low-precision Stochastic Gradient Langevin Dynamics

When performing gradient updates in low-precision training, there are two common choices, *full-precision and low-precision gradient accumulators* depending on whether we store an additional copy of full-precision weights. Low-precision SGLD (Zhang et al., 2022) considers both choices.

Low-precision SGLD with full-precision gradient accumulators (SGLDLP-F) only quantizes weights before computing the gradient. The update rule can be defined as:

$$\mathbf{x}_{k+1} = \mathbf{x}_k - \eta Q_G \left( \widetilde{\nabla U}(Q_W(\mathbf{x}_k)) \right) + \sqrt{2\eta}\xi_{k+1}, \tag{1}$$

$\widetilde{\nabla U}$ is the unbiased gradient estimation of $U$. Zhang et al. (2022) showed that the SGLDLP-F outperforms its counterpart low-precision SGD with full-gradient accumulators (SGDLP-F). The computation costs can be further reduced using low-precision gradient accumulators by only keeping low-precision weights. Low-precision SGLD with low-precision gradient accumulators (SGLDLP-L) can be defined as the following:

$$\mathbf{x}_{k+1} = Q_W \left( \mathbf{x}_k - \eta Q_G(\widetilde{\nabla U}(\mathbf{x}_k)) + \sqrt{2\eta}\xi_{k+1} \right). \tag{2}$$

Zhang et al. (2022) studied the convergence property of both SGLDLP-F and SGLDLP-L under strongly-log-concave distributions and showed that a small step size deteriorates the performance of SGLDLP-L. To mitigate this problem, Zhang et al. (2022) proposed a variance-corrected quantization function (Algorithm 2 in Appendix D).

## 2.3 Stochastic Gradient Hamiltonian Monte Carlo

Given a dataset $D$, a model with weights (i.e., model parameters) $\mathbf{x} \in \mathbb{R}^d$, and a prior $p(\mathbf{x})$, we are interested in sampling from the posterior $p(\mathbf{x}|D) \propto \exp(-U(\mathbf{x}))$, where $U(\mathbf{x}) = -\log p(D|\mathbf{x}) - \log p(\mathbf{x})$ is the energy function. In order to sample from the target distribution, SGHMC (Chen et al., 2014) is proposed and strongly related to the underdamped Langevin dynamics. Cheng et al. (2018) proposed the following discretization of underdamped Langevin dynamics (10) with stochastic gradient:

$$\mathbf{v}_{k+1} = \mathbf{v}_k e^{-\gamma\eta} - u\gamma^{-1}(1 - e^{-\gamma\eta})\widetilde{\nabla U}(\mathbf{x}_k) + \xi_k^{\mathbf{v}} \tag{3}$$
$$\mathbf{x}_{k+1} = \mathbf{x}_k + \gamma^{-1}(1 - e^{-\gamma\eta})\mathbf{v}_k + u\gamma^{-2}(\gamma\eta + e^{-\gamma\eta} - 1)\widetilde{\nabla U}(\mathbf{x}_k) + \xi_k^{\mathbf{x}},$$

where $u, \gamma$ denote the hyperparameters of the inverse mass and friction respectively, and $\xi_k^{\mathbf{v}}, \xi_k^{\mathbf{x}}$ are normal distributed in $\mathbb{R}^d$ satisfying that :

$$\begin{aligned}
\mathbb{E}\xi_k^{\mathbf{v}}(\xi_k^{\mathbf{v}})^\top &= u(1 - e^{-2\gamma\eta}) \cdot \mathbf{I}, \\
\mathbb{E}\xi_k^{\mathbf{x}}(\xi_k^{\mathbf{x}})^\top &= u\gamma^{-2}(2\gamma\eta + 4e^{-\gamma\eta} - e^{-2\gamma\eta} - 3) \cdot \mathbf{I}, \\
\mathbb{E}\xi_k^{\mathbf{x}}(\xi_k^{\mathbf{v}})^\top &= u\gamma^{-1}(1 - 2e^{-\gamma\eta} + e^{-2\gamma\eta}) \cdot \mathbf{I}.
\end{aligned} \tag{4}$$

# 3 Low-Precision Stochastic Gradient Hamiltonian Monte Carlo

In this section, we investigate the convergence property of low-precision SGHMC under non-log-concave target distributions. We defer the convergence analysis of low-precision SGHMC under strongly log-concave target distributions, as well as the extension analysis under non-log-concave target distributions of low-precision SGLD (Zhang et al., 2022) to Appendix A and B respectively. All of our theorems are based on the fixed point representation and omit the clipping effect. We show that low-precision SGHMC exhibits superior convergence rates and mitigates the performance degradation caused by the quantization error than low-precision SGLD, especially for non-log-concave target distributions. Similar to Zhang et al. (2022), we also observe an overdispersion phenomenon in sampling distributions obtained by SGHMC with low-precision gradient accumulators, and we examine the effectiveness of variance-corrected quantization function in resolving this overdispersion problem.

In the statement of theorems, the big-O notation $\tilde{\mathcal{O}}$ gives explicit dependence on the quantization error $\Delta$ and concentration parameters $(\lambda^*, \mu^*)$ but hides multiplicative terms that polynomially depend on the other parameters (e.g., dimension $d$, friction $\gamma$, inverse mass $u$ and gradients variance $\sigma^2$). We refer readers to the appendix for all the theorems' proof. Before diving into theorems, we first introduce necessary assumptions for the convergence analysis as follows:

**Assumption 1** (Smoothness). *The energy function $U$ is $M$-smooth, i.e., there exists a positive constant $M$ such that*

$$\|\nabla U(\mathbf{x}) - \nabla U(\mathbf{y})\|^2 \leq M^2 \|\mathbf{x} - \mathbf{y}\|^2, \quad \text{for any } \mathbf{x}, \mathbf{y} \in \mathbb{R}^d.$$

**Assumption 2** (Dissaptiveness). *There exist constants $m_2, b > 0$, such that the following holds*

$$\langle \nabla U(\mathbf{x}), \mathbf{x} \rangle \geq m_2 \|\mathbf{x}\|^2 - b, \quad \text{for any } \mathbf{x} \in \mathbb{R}^d.$$

**Assumption 3** (Bounded Variance). *There exists a constant $\sigma^2 > 0$, such that the following holds*

$$\mathbb{E} \left\| \widetilde{\nabla U}(\mathbf{x}) - \nabla U(\mathbf{x}) \right\|^2 \leq \sigma^2, \quad \text{for any } \mathbf{x} \in \mathbb{R}^d.$$

Beyond the above assumptions, we further define $\kappa_1 = M/m_1$ (refer to Assumption 4 in Appendix A) and $\kappa_2 = M/m_2$ as the condition numbers for strongly log-concave and non-log-concave target distribution, respectively, and denote the global minimum of $U(\mathbf{x})$ as $\mathbf{x}^*$. All of our assumptions are standard and commonly used in the sampling literature. In particular, Assumption 2 is a standard assumption (Raginsky et al., 2017; Zou et al., 2019; Gao et al., 2022) in the analysis of sampling from non-log-concave distributions and is essential to guarantee the convergence of underdamped Langevin dynamics. Assumption 3 can be further relaxed, allowing the variance of $\widetilde{\nabla U}(\mathbf{x})$ scale up w.r.t $\mathbf{x}$. Please refer to the appendix H-P.

**Definition 1.** *Let $\lambda^*$ and $\mu^*$ denote the contraction rates for continuous-time overdamped Langevin dynamics and underdamped Langevin dynamics respectively. In other words, let $x_t$ follow the overdamped (or underdamped) Langevin dynamics initialized at $x_0 = 0$, $\pi_z$ be the invariant distribution, $p_t$ be the marginal distribution $x_t$, then $\lambda^*$ and $\mu^*$ satisfy*

$$\mathcal{W}_2^2(p_t, \pi_z) \leq C e^{-\lambda^* t/d}, \text{ or } \mathcal{W}_2^2(p_t, \pi_z) \leq C e^{-\mu^* t/d},$$

*for some constant $C$.*

The contraction rates $\mu^*$ and $\lambda^*$ are related to the nature of the Langevin dynamics. In general, the contraction rates exponentially depend on the dimension $d$. For example, two popular approaches to analyze the Wasserstein distance convergence property are the couplings method (Dalalyan & Riou-Durand, 2020; Eberle et al., 2019) and Bakry–Émery method based on which the exponential convergence of the kinetic Fokker–Planck equation is proved (Bakry & Émery, 2006; Baudoin, 2016; 2017). Unfortunately, both rates lead to exponential dependency on dimension in general. It raises a crucial open question of whether restricted models can exhibit improved dimensional dependence. While overdamped Langevin diffusions have seen corresponding advancements, as evidenced in (Eberle et al., 2019; Zimmer, 2017), analogous progress for underdamped Langevin diffusions remains underexplored. More detailed discussions of $\mu^*$ and $\lambda^*$ (and their uniform bounds) can be found in the Appendix C.

## 3.1 Full-Precision Gradient Accumulators

Adopting the update rule in equations (3), we propose low-precision SGHMC with full gradient accumulators (SGHMCLP-F) as the following:

$$\mathbf{v}_{k+1} = \mathbf{v}_k e^{-\gamma\eta} - u\gamma^{-1}(1 - e^{-\gamma\eta})Q_G(\widetilde{\nabla U}(Q_W(\mathbf{x}_k))) + \xi_k^{\mathbf{v}} \tag{5}$$

$$\mathbf{x}_{k+1} = \mathbf{x}_k + \gamma^{-1}(1 - e^{-\gamma\eta})\mathbf{v}_k + u\gamma^{-2}(\gamma\eta + e^{-\gamma\eta} - 1)Q_G(\widetilde{\nabla U}(Q_W(\mathbf{x}_k))) + \xi_k^{\mathbf{x}},$$

which keeps full-precision parameters $\mathbf{v}_k, \mathbf{x}_k$ at each iteration and quantizes them to low-precision representations before taking the gradient. Our analysis for non-log-concave distributions utilizes similar techniques in Raginsky et al. (2017). We are now ready to present our first theorem:

**Theorem 1.** *Assuming 1, 2 and 3 hold. Let $p^*$ denote the target distribution of $(\mathbf{x}, \mathbf{v})$. If $\gamma^2 \leq 4Mu$ and setting the step size $\eta = \tilde{\mathcal{O}}\left(\frac{\mu^* \epsilon^2}{\log(1/\epsilon)}\right)$ satisfying*

$$\eta \leq \min\left\{\frac{\gamma}{4\left(8Mu + u\gamma + 22\gamma^2\right)}, \sqrt{\frac{4u^2}{4Mu + 3\gamma^2}}, \frac{6\gamma bu}{\left(4Mu + 3\gamma^2\right)d}, \frac{1}{8\gamma}, \frac{\gamma m_2}{12(21u + \gamma)M^2}, \frac{8(\gamma^2 + 2u)}{(20u + \gamma)\gamma}\right\},$$

*then after $K$ steps starting at the initial point $\mathbf{x}_0 = \mathbf{v}_0 = 0$, the output $(\mathbf{x}_K, \mathbf{v}_K)$ of SGHMCLP-F in (5) satisfies*

$$\mathcal{W}_2(p(\mathbf{x}_K, \mathbf{v}_K), p^*) \leq \tilde{\mathcal{O}}\left(\epsilon + \widetilde{A}\sqrt{\log\left(\frac{1}{\epsilon}\right)}\right),$$

*for some $K$ satisfying*

$$K = \tilde{\mathcal{O}}\left(\frac{1}{\epsilon^2 \mu^{*2}} \log^2\left(\frac{1}{\epsilon}\right)\right),$$

*where constants are defined as: $\widetilde{A} = \max\left\{\sqrt{\Delta^2 d + \sigma^2}, \sqrt[4]{\Delta^2 d + \sigma^2}\right\}$.*

Similar to the convergence result of full-precision SGHMC or SGLD (Raginsky et al., 2017; Gao et al., 2022), the above upper bound of the 2-Wasserstein distance contains an $\epsilon$ term and a $\log(\epsilon^{-1})$ term. The difference is that for the SGHMCLP-F algorithm, the quantization error $\Delta$ affects the multiplicative constant of the $\log(\epsilon^{-1})$ term. Focusing on the effect of quantization error $\Delta$, due to the fact that $\log(x) \leq x^{1/e}$, one can tune the choice of $\epsilon$ and $\eta$ and obtain a $\tilde{\mathcal{O}}\left(\Delta^{e/(1+2e)}\right)$ 2-Wasserstein bound. As for the non-convergence of our result (i.e, $\log(\epsilon^{-1})$ term), we note that even for full-precision sample algorithms, the best non-asymptotic convergence result in the 2-Wasserstein distance (Zou et al., 2019; Raginsky et al., 2017; Gao et al., 2022) also contain a $\log(\epsilon^{-1})$ factor which is brought by stochastic gradient noise, and diverge as $\epsilon \to 0$. The non-convergence of our Wasserstein upper bound is due to the accumulation of stochastic gradient noise and stochastic discretion error. Conceptually, these random errors may average out over iterations when the iteration number increases to infinity (i.e., the law of large numbers), as in the classical ergodic theory of Markov chain (Theorem 17.25, 17.28 of Kallenberg & Kallenberg (1997)). However, our mathematical tools lead to an upper bound that involves some weighted summation of the norm of these random errors over iterations rather than the summation of these random errors. Under strongly log-concave target distributions with no discretion error, this sum is bounded as $t \to \infty$ and proportional to the stepsize, allowing for a sufficiently small step size to zero the bound Dalalyan & Karagulyan (2019); Cheng et al. (2018). However, for general cases, this sum grows to infinity. It is yet an open question to sharpen this type of analysis.

With the same technical tools, we conduct a similar convergence analysis of SGLDLF-P for non-log-concave target distributions. The details are deferred in Theorem 7 of Appendix B. Comparing Theorems 1 and 7, we show that SGHMCLP-F can achieve lower 2-Wasserstein distance (i.e., $\tilde{\mathcal{O}}\left(\log^{1/2}\left(\epsilon^{-1}\right)\Delta^{1/2}\right)$ versus $\tilde{\mathcal{O}}\left(\log\left(\epsilon^{-1}\right)\Delta^{1/2}\right)$) for non-log-concave target distribution within fewer iterations (i.e., $\tilde{\mathcal{O}}\left(\epsilon^{-2}\mu^{*-2}\log^2\left(\epsilon^{-1}\right)\right)$ versus $\tilde{\mathcal{O}}\left(\epsilon^{-4}\lambda^{*-1}\log^5\left(\epsilon^{-1}\right)\right)$). Furthermore, by the same argument in the previous paragraph, after carefully choosing the stepsize $\eta$, the 2-Wasserstein distance of the SGLDLF-P algorithm can be further bounded by $\tilde{\mathcal{O}}\left(\Delta^{e/(2+2e)}\right)$ which is worse than the bound $\tilde{\mathcal{O}}\left(\Delta^{e/(1+2e)}\right)$ obtained by SGHMC. We verify the advantage of SGHMCLF-P over SGLDLF-P by our simulations in section 4.

### 3.2 Low-Precision Gradient Accumulators

The storage and computation costs of low-precision algorithms can be further reduced by low-precision gradient accumulators. We can adopt low-precision SGHMC with low-precision gradient accumulators (SGHMCLP-L) as

$$\mathbf{v}_{k+1} = Q_W\left(\mathbf{v}_k e^{-\gamma\eta} - u\gamma^{-1}(1 - e^{\gamma\eta})Q_G(\widetilde{\nabla U}(\mathbf{x}_k)) + \xi_k^{\mathbf{v}}\right), \tag{6}$$

$$\mathbf{x}_{k+1} = Q_W\left(\mathbf{x}_k + \gamma^{-1}(1 - e^{-\gamma\eta})\mathbf{v}_k + u\gamma^{-2}(\gamma\eta + e^{-\gamma\eta} - 1)Q_G(\widetilde{\nabla U}(\mathbf{x}_k)) + \xi_k^{\mathbf{x}}\right).$$

Similar to the observation of Zhang et al. (2022), we also empirically find that the output $\mathbf{x}_K$'s distribution has a larger variance than the target distribution (see Figures 1 (a) and 2 (a)), as the update rule (6) introduces extra rounding noise. Our theorem in the section aims to support this argument. We present the convergence theorem of SGHMCLP-L under non-log-concave target distributions.

**Theorem 2.** *Assuming 1, 2 and 3 hold. Let $p^*$ denote the target distribution of $(\mathbf{x}, \mathbf{v})$. If $\gamma^2 \leq 4Mu$ and setting the step size $\eta = \tilde{\mathcal{O}}\left(\frac{\mu^* \epsilon^2}{\log(1/\epsilon)}\right)$ satisfying*

$$\eta \leq \min\left\{\frac{\gamma}{4\left(8Mu + u\gamma + 22\gamma^2\right)}, \sqrt{\frac{4u^2}{4Mu + 3\gamma^2}}, \frac{6\gamma bu}{(4Mu + 3\gamma^2)\,d}, \frac{1}{8\gamma}, \frac{\gamma m_2}{12(21u + \gamma)M^2}, \frac{8(\gamma^2 + 2u)}{(20u + \gamma)\gamma}\right\},$$

*then after $K$ steps starting at the initial point $\mathbf{x}_0 = \mathbf{v}_0 = 0$, the output $(\mathbf{x}_K, \mathbf{v}_K)$ of SGHMCLP-L in (6) satisfies*

$$\mathcal{W}_2(p(\mathbf{x}_K, \mathbf{v}_K), p^*) = \tilde{\mathcal{O}}\left(\epsilon + \sqrt{\max\{\sigma^2, \sigma\}\log\left(\frac{1}{\epsilon}\right)} + \frac{\log^{3/2}\left(\frac{1}{\epsilon}\right)}{\epsilon^2}\sqrt{\Delta}\right), \tag{7}$$

*for some $K$ satisfying*

$$K = \tilde{\mathcal{O}}\left(\frac{1}{\epsilon^2 \mu^{*2}}\log^2\left(\frac{1}{\epsilon}\right)\right).$$

For non-log-concave target distribution, the output of the naïve SGHMCLP-L has a worse convergence upper bound than Theorem 1. The source of the observed problem is the variance introduced by the quantization $Q_W$, causing actual variances of $(\mathbf{x}_k, \mathbf{v}_k)$ to be larger than the variances needed. In Theorem 8, we generalize the result of the naïve SGLDLP-L in (Zhang et al., 2022) to non-log-concave target distributions, and we defer this theorem to appendix B. Similarly, we observe that SGHMCLP-L needs fewer iterations than SGLDLP-L in terms of the order w.r.t. $\epsilon$ and $\log(\epsilon^{-1})$ ($\tilde{\mathcal{O}}\left(\epsilon^{-2}\mu^{*-2}\log^2\left(\epsilon^{-1}\right)\right)$ versus $\tilde{\mathcal{O}}\left(\epsilon^{-4}\lambda^{*-1}\log^5\left(\epsilon^{-1}\right)\right)$) and achieves better upper bound $\tilde{\mathcal{O}}\left(\epsilon^{-2}\log^{3/2}\left(\epsilon^{-1}\right)\sqrt{\Delta}\right)$ versus $\tilde{\mathcal{O}}\left(\epsilon^{-4}\log^5\left(\epsilon^{-1}\right)\sqrt{\Delta}\right)$.

By the same argument in Theorem 1's discussion, after carefully choosing the stepsize $\eta$, the 2-Wasserstein distance between samples obtained by SGHMCLP-L and non-log-concave target distributions can be further bounded as $\tilde{\mathcal{O}}\left(\Delta^{e/(3+6e)}\right)$, whilst the distance between the samples obtained by SGLDLP-L to the target can be bounded as $\tilde{\mathcal{O}}\left(\Delta^{e/10(1+e)}\right)$. Thus, low-precision SGHMC is more robust to the quantization error than SGLD.

### 3.3 Variance Correction

To resolve the overdispersion caused by low-precision gradient accumulators, Zhang et al. (2022) proposed a quantization function $Q^{vc}$ (refer to Algorithm 2 in Appendix D) that directly samples from the discrete weight space instead of quantizing a real-valued Gaussian sample. This quantization function aims to reduce the discrepancy between the ideal sampling variance (i.e., the required variance of full-precision counterpart algorithms) and the actual sampling variance in our low-precision algorithms. We adopt the variance-corrected quantization function to low-precision SGHMC (VC SGHMCLP-L) and study its convergence property for non-log-concave target distributions. We extend the convergence analysis of VC SGLDLP-L in Zhang et al. (2022) to the case of the non-log-concave distributions as well. The details are deferred to Appendix B for comparison purposes. Let $\mathrm{Var}_{\mathbf{v}}^{hmc} = u(1 - e^{-2\gamma\eta})$ and $\mathrm{Var}_{\mathbf{x}}^{hmc} = u\gamma^{-2}(2\gamma\eta + 4e^{-\gamma\eta} - e^{-2\gamma\eta} - 3)$, which are the variances added by the underdamped Langevin dynamics in (3). The VC SGHMCLP-L can be done as follows:

$$\mathbf{v}_{k+1} = Q^{vc}\left(\mathbf{v}_k e^{-\gamma\eta} - u\gamma^{-1}(1 - e^{-\gamma\eta})Q_G(\widetilde{\nabla U}(\mathbf{x}_k)), \mathrm{Var}_{\mathbf{v}}^{hmc}, \Delta\right), \tag{8}$$

$$\mathbf{x}_{k+1} = Q^{vc}\left(\mathbf{x}_k + \gamma^{-1}(1 - e^{-\gamma\eta})\mathbf{v}_k + u\gamma^{-2}(\gamma\eta + e^{-\gamma\eta} - 1)Q_G(\widetilde{\nabla U}(\mathbf{x}_k)), \mathrm{Var}_{\mathbf{x}}^{hmc}, \Delta\right).$$

The variance corrected quantization function $Q^{vc}$ aims to output a low-precision random variable with the desired mean and variance. When the desired variance $v$ is larger than $\Delta^2/4$, which is the largest possible

variance introduced by the quantization $Q^s$, the variance-corrected quantization first adds a small Gaussian noise to compensate for the variance and then adds a categorical random variable with a desired variance. When $v$ is less than $\Delta^2/4$ the variance-corrected quantization computes the actual variance introduced by $Q^s$. If it is larger than $v$, a categorical random variable is added to the weights to match the desired variance $v$. If it is less than $v$, we will not be able to match the variance after quantization. However, this case arises only with exceptionally small step sizes. With the variance-corrected quantization $Q^{vc}$ in hand, we now present the convergence analysis of the VC SGHMCLP-L for non-log-concave distributions.

**Theorem 3.** *Assuming 1, 2 and 3 hold and $\mathbb{E}\left\|Q_G(\widetilde{\nabla U}(x))\right\|_2^2 \leq G^2$. Let $p^*$ be the target distribution of $\mathbf{x}$. If $\gamma^2 \leq 4Mu$ and setting the step size $\eta = \tilde{\mathcal{O}}\left(\frac{\mu^*\epsilon^2}{\log(1/\epsilon)}\right)$ satisfying*

$$\eta \leq \min\left\{\frac{\gamma}{4\left(8Mu + u\gamma + 22\gamma^2\right)}, \sqrt{\frac{4u^2}{4Mu + 3\gamma^2}}, \frac{6\gamma bu}{\left(4Mu + 3\gamma^2\right)d}, \frac{1}{8\gamma}, \frac{\gamma m_2}{12(21u + \gamma)M^2}, \frac{8(\gamma^2 + 2u)}{(20u + \gamma)\gamma}\right\},$$

*then after $K$ steps starting at the initial point $\mathbf{x}_0 = \mathbf{v}_0 = 0$ the output $(\mathbf{x}_K)$ of the VC SGHMCLP-L in (9) satisfies*

$$\mathcal{W}_2(p(\mathbf{x}_K), p^*) = \tilde{\mathcal{O}}\left(\epsilon + \sqrt{\max\{\sigma^2, \sigma\}\log\left(\frac{1}{\epsilon}\right)} + \frac{\log\left(\frac{1}{\epsilon}\right)}{\epsilon}\sqrt{\Delta}\right), \tag{9}$$

*for some $K$ satisfying*

$$K = \tilde{\mathcal{O}}\left(\frac{1}{\epsilon^2\mu^{*2}}\log^2\left(\frac{1}{\epsilon}\right)\right).$$

Compared with Theorem 1, we cannot show that the variance corrected quantization fully resolves the overdispersion problem observed for non-log-concave target distributions. However comparing with Theorem 2, we show in Theorem 3 that the variance-corrected quantization can improve the upper bound w.r.t. $\epsilon$ from $\tilde{\mathcal{O}}\left(\epsilon^{-2}\log^{3/2}\left(\epsilon^{-1}\right)\sqrt{\Delta}\right)$ to $\tilde{\mathcal{O}}\left(\epsilon^{-1}\log\left(\epsilon^{-1}\right)\sqrt{\Delta}\right)$. In Theorem 9, we generalize the result of the VC SGLDLP-L in (Zhang et al., 2022) to non-log-concave target distributions, and we defer this theorem to appendix B. Similarly, we observe that VC SGHMCLP-L needs fewer iterations than VC SGLDLP-L in terms of the order w.r.t. $\epsilon$ and $\log(\epsilon^{-1})$ $(\tilde{\mathcal{O}}\left(\epsilon^{-2}\mu^{*-2}\log^2\left(\epsilon^{-1}\right)\right)$ versus $\tilde{\mathcal{O}}\left(\epsilon^{-4}\lambda^{*-1}\log^5\left(\epsilon^{-1}\right)\right))$.

Beyond the above analysis, we apply similar mathematical tools and study the convergence property of VC SGHMCLP-L and VC SGLDLP-L in terms of $\Delta$ for non-log-concave target distributions. Based on the Theorem 2 and 3, the variance-corrected quantization can improve the upper bound from $\tilde{\mathcal{O}}\left(\Delta^{e/(3+6e)}\right)$ to $\tilde{\mathcal{O}}\left(\Delta^{e/(2+4e)}\right)$. Compared with VC SGLDLP-L, the VC SGHMCLP-L has a better upper bound (i.e. $\tilde{\mathcal{O}}\left(\Delta^{e/(2+4e)}\right)$ versus $\tilde{\mathcal{O}}\left(\Delta^{e/6(1+e)}\right)$). Interestingly, the naïve SGHMCLP-L has similar dependence on the quantization error $\Delta$ with VC SGLDLP-L but saves more computation resources since the variance corrected quantization requires sampling discrete random variables. We verify our findings in Table 4.

## 4 Experiments

We evaluate the performance of the proposed low-precision SGHMC algorithms across various experiments: Gaussian and Gaussian mixture distributions (Section 4.1), Logistic Regression and Multi-Layer Perceptron (MLP) applied to the MNIST dataset (Section 4.2), and ResNet-18 on both CIFAR-10 and CIFAR-100 datasets (Section 4.3). Additionally, we compare the accuracy of our proposed algorithms with their SGLD counterparts. Throughout all experiments, low-precision arithmetic is implemented using *qtorch* (Zhang et al., 2019). Beyond our theoretical settings, our experiments encompass a range of low-precision setups, including fixed point, block floating point, as well as quantization of weights, gradients, errors, and activations. For more details of our low-precision settings used in experiments, please refer to Appendix D

### 4.1 Sampling from standard Gaussian and Gaussian mixture distributions

We first demonstrate the performance of low-precision SGHMC for fitting synthetic distributions. We use the standard Gaussian distribution and Gaussian mixture distribution to represent strongly log-concave and

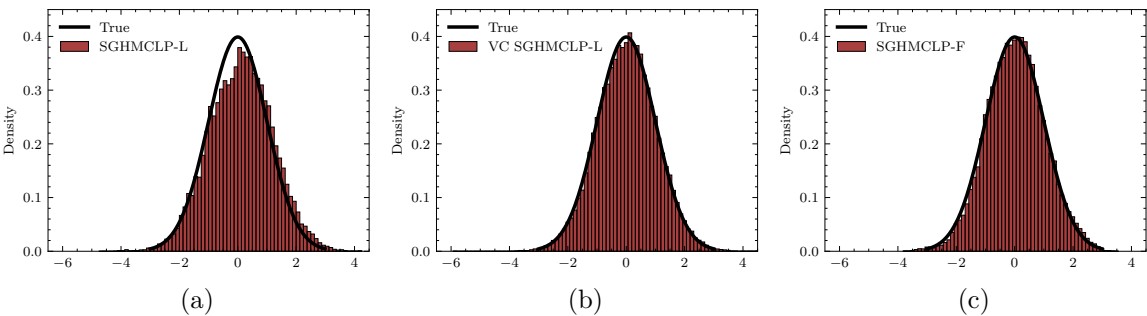

Figure 1: Low-precision SGHMC on a Gaussian distribution. (a): SGHMCLP-L. (b): VC SGHMCLP-L. (c): SGHMCLP-F. VC SGHMCLP-L and SGHMCLP-F converge to the true distribution, whereas naïve SGHMCLP-L suffers a larger variance.

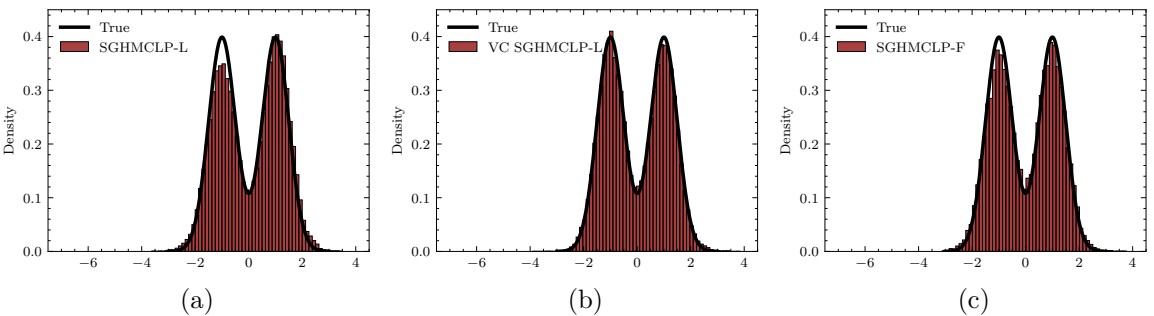

Figure 2: Low-precision SGHMC with on a Gaussian mixture distribution. (a): SGHMCLP-L. (b): VC SGHMCLP-L. (c): SGHMCLP-F. VC SGHMCLP-L and SGHMCLP-F converge to the true distribution, whereas naïve SGHMCLP-L suffers a larger variance.

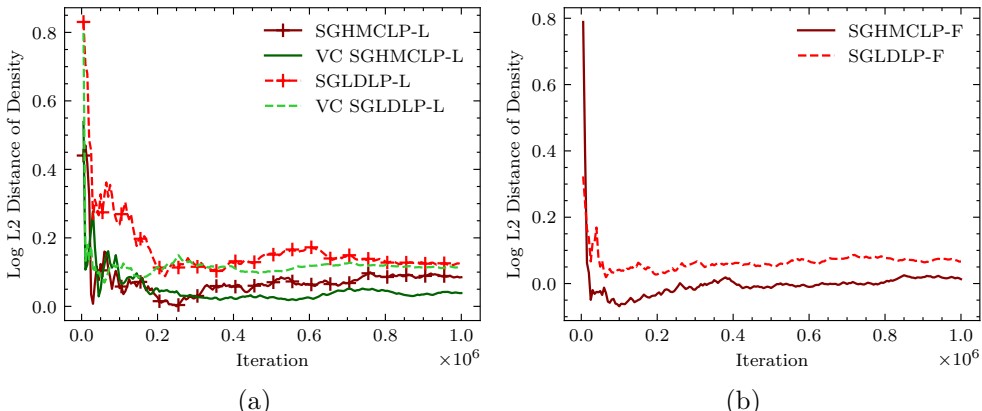

Figure 3: Log $L_2$ distance from sample density estimation obtained by low-precision SGHMC and SGLD to the Gaussian mixture distribution. (a) Low-precision gradient accumulators. (b): Full-precision gradient accumulators. Overall, SGHMC methods enjoy a faster convergence speed. In particular, SGHMCLP-L achieves a lower distance compared to SGLDLP-L and VC SGLDLP-L.

non-log-concave distribution, respectively. The density of the Gaussian mixture example is defined as

$$e^{-U(\mathbf{x})} = e^{2\|\mathbf{x}-1\|^2} + e^{2\|\mathbf{x}+1\|^2}.$$

We use 8-bit fixed point representation with 4 of them representing the fractional part. For hyper-parameters please the Appendix D. The simulation results are shown in Figure 1 and 2. From Figure 1(a) and 2(a),

we see that the sample from naïve SGHMCLP-L has a larger variance than the target distribution. This verifies the results we prove in Theorem 2. In Figure 1(b) and 2(b), we verify that the variance-corrected quantizer mitigates this problem by matching variance of the quantizer to the variance $\mathrm{Var}_{\mathbf{x}}^{hmc}$ defined by the underdamped Langevin dynamics (10). In Figure 3, we compare the performance of low-precision SGHMC with low-precision SGLD for sampling from Gaussian mixture distribution. Since calculating the 2-Wasserstein distance over long iterations is time-consuming, instead of computing the Wasserstein distance, we resort to $L_2$ distance of the sample density estimation to the true density function. It shows that low-precision SGHMC enjoys faster convergence speed and smaller distance, especially SGHMCLP-L compared to SGLDLP-L and VC SGLDLP-L.

We also study in which case the variance-corrected quantizer is advantageous over the naïve stochastic quantization function. We test the 2-Wasserstein sampling error of VC SGHMCLP-L and SGHMCLP-L over different variances. The result is shown in Figure 4. We find that when the variance $\mathrm{Var}_{\mathbf{x}}^{hmc}$ is close to the largest quantization variance $\Delta^2/4$, the variance corrected quantization function shows the largest advantage over the naïve quantization. When the variance $\mathrm{Var}_{\mathbf{x}}^{hmc}$ is less than $\Delta^2/4$, the correction has a chance to fail. When the variance $\mathrm{Var}_{\mathbf{x}}^{hmc}$ is 100 times the quantization variance, the advantage of variance-corrected quantizer shows less advantage. One possible reason is that the quantization variance eliminated by the variance-corrected quantizer is not critical compared to $\mathrm{Var}_{\mathbf{x}}^{hmc}$ which is the intrinsic variance for SGHMC. We advocate for the adoption of

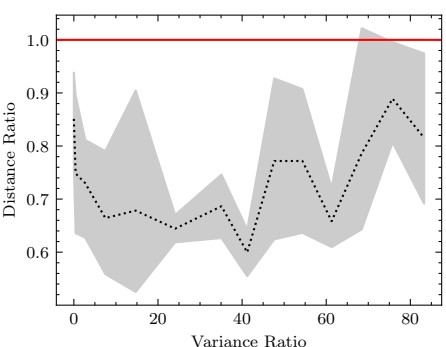

Figure 4: Mean (dotted line) and 95% confidence interval (shaded area) of 2-Wasserstein error ratio between VC SGHMCLP-L & SGHMCLP-L (Smaller means the variance correction is more effective), computed over 5 experimental runs. The x-axis represents the ratio between $\mathrm{Var}_{\mathbf{x}}^{hmc}$ and $\Delta^2/4$.

variance-corrected quantization under the specific condition where the ideal variance approximates $\Delta^2/4$. Our observations indicate that this scenario yields the most significant performance gains. Conversely, in other situations, we suggest employing naïve low-precision gradient accumulators, as they offer comparable performance while conserving computational resources.

## 4.2  MNIST

In this section, we further examine the sampling performance of low-precision SGHMC and SGLD on strongly log-concave distributions and non-log-concave distributions on real-world data. We use logistic and multilayer perceptron (MLP) models to represent the class of strongly log-concave and non-log-concave distributions, respectively. The results are shown in Figure 5 and 6. We use $\mathcal{N}\left(0, 10^{-2}\right)$ as the prior distribution and fixed point number representation, where we set 2 integer bits and various fractional bits. A smaller number of fractional bits corresponds to a larger quantization gap $\Delta$. For MLP model, we use two-layer MLP with 100 hidden units and ReLu nonlinearities. We report the training negative log-likelihood (NLL) with different numbers of fractional bits in Figure 5 and 6. For detailed hyperparameters and experiment setup, please see Appendix D.

From the results on MNIST, we can see that when using full-precision gradient accumulators, low-precision SGHMC are robust to the quantization error. Even when we use only 2 fractional bits, SGHMCLP-F can still converge to a distribution with a small and stable NLL but with more iterations. However, regarding low-precision gradient accumulators, SGHMCLP-L and SGLDLP-L are less robust to the quantization error. As the precision error increases, both SGHMCLP-L and SGLDLP-L have a worse convergence pattern compared to SGHMCLP-F and SGLDLP-F. We showed empirically that SGHMCLP-L and VC SGHMCLP-L outperform SGLDLP-L and VC SGLDLP-L. As shown in Figure 5 and 6, when we increase the quanti-

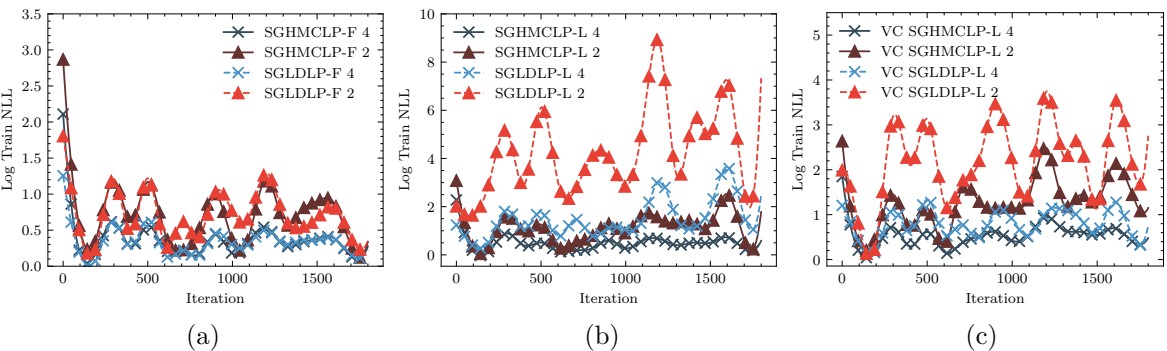

Figure 5: Training NLL of low-precision SGHMC and SGLD on logistic model with MNIST in terms of different numbers of fractional bits. (a): Full-precision gradient accumulators. (b): Low-precision gradient accumulators. (c): Variance-corrected quantizer. SGHMCLP-F achieves comparable results with SGLDLP-F. However, both SGHMCLP-L and VC SGHMCLP-L show more robustness to quantization error, especially when the number of representable bits is low. Please be aware of the different scales of y-axis across three figures.

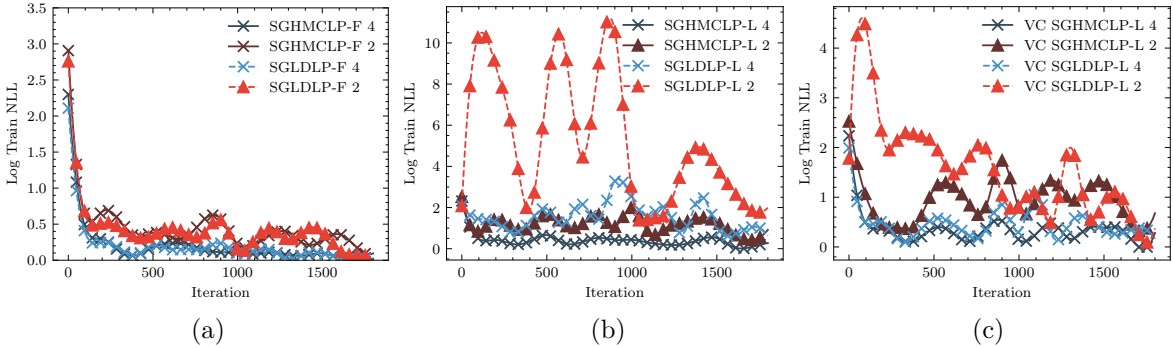

Figure 6: Training NLL of low-precision SGHMC and SGLD on MLP with MNIST in terms of different numbers of fractional bits. (a): Full-precision gradient accumulators. (b): Low-precision gradient accumulators. (c): Variance-corrected quantizer. SGHMCLP-F achieves comparable results with SGLDLP-F. However, both SGHMCLP-L and VC SGHMCLP-L show more robustness to quantization error, especially when the number of representable bits is low. Please be aware of the different scales of y-axis across three figures.

zation error, SGHMCLP-L and VC SGHMCLP-L are more robust than SGLDLP-L and VC SGLDLP-L, respectively.

## 4.3 CIFAR-10 & CIFAR-100

We consider image tasks CIFAR-10 and CIFAR-100 on the ResNet-18. We use 8-bit number representation following Zhang et al. (2022). We report the test errors averaging over 3 runs in Tables 2 and 4. For detailed hyperparameters and experiment setup, please see Appendix D.

**Fixed Point** We employ fixed point representations for both weights and gradients while retaining full precision for activations and errors following previous work (Zhang et al., 2022). From the figure, unsurprisingly the full-precision algorithms outperform their low-precision counterparts. But with long enough iterations the performance gap between SGHMC/SGLD and SGHMCLP-F/SGLDLP-F converges toward zero. Similar to the results in previous sections, SGHMCLP-F is comparable with SGDLP-F, and the naïve SGHMCLP-L significantly outperforms naïve SGLDLP-L and SGDLP-L across datasets and architectures. For example, SGHMCLP-L outperforms SGLDLP-L by 1.19% on CIFAR-10, and SGHMCLP-L outperforms SGLDLP-L by 0.58% on CIFAR-100. Furthermore, from the result in Figure 7, we empirically show that

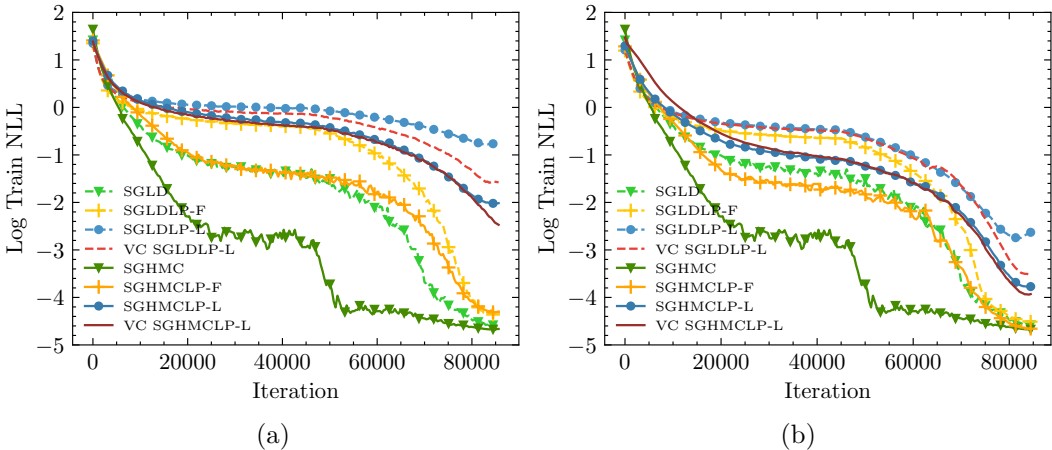

Figure 7: Log of training NLL of low-precision SGHMC and SGLD on ResNet-18 with CIFAR-100. (a): 8-bits Fixed Point. (b): 8-bits Block Floating Point. For fixed point representations, low-precision SGHMC shows faster convergence and SGHMCLP-L outperforms SGLDLP-L and VC SGLDLP-L.

the convergence speed of SGHMC is way better than the convergence speed of SGLD. SGHMCLP-L even achieves faster convergence than SGLDLP-F. When the variance $\text{Var}_{\mathbf{x}}^{hmc}$ is comparable with or less than $\Delta^2/4$, we recommend implementing SGHMCLP-L rather than VC SGHMCLP-L. This is the case when we assess the performance of low-precision SGHMC on CIFAR-10 and CIFAR-100. Notably, even in the absence of the performance enhancement provided by the variance-corrected quantization function, the test results indicate that SGHMCLP-L's performance is on par with its SGLD counterpart with variance correction. This result verifies our findings in Theorems 2 and 9.

**Block Floating Point**   We also consider the block floating point (BFP) representation adopted with deep models, which causes less quantization error and thus performs better compared with fixed point representation (Song et al., 2018). Given sufficient iterations, the performance differences between SGHMC/SGLD and SGHMCLP-F/SGLDLP-F almost disappear. As illustrated in plot b) of Figure 7, SGHMCLP-F outperforms full-precision SGLD in block floating point low-precision format. By using BFP, the performance of all low-precision methods improves over fixed point representation. The naïve SGHMCLP-L outperforms the naïve SGLDLP-L 0.82%. Moreover, the naïve SGHMCLP-L achieves comparable results with the VC SGLDLP-L method, and SGHMCLP-L can save more computation resources since the variance-corrected quantization function would need to sample an additional categorical random vector $\mathbf{c} \in \mathbb{R}^d$ at each iteration. Let $\text{Var}_{\mathbf{x}}^{sgld} = 2\eta$ denote the variance added by overdamped Langevin dynamics in (14). For most deep learning tasks, a small step size is preferred, and thus there is a large chance that $\text{Var}_{\mathbf{x}}^{sgld} \leq \Delta^2/4$ in which case we recommend running the naïve SGHMCLP-L to achieve comparable accuracy and save more computation resources.

**Expected Calibration Error**   To study the model calibration of low-precision SGHMC, we further report the results of expected calibration error (ECE) (Guo et al., 2017) in Table 3 and 5. We observe that sometimes SGLDLP-L and SGLDLP-F achieve a lower ECE than the full-precision SGLD counterpart, implying that the corresponding sample distributions deviate from the true target posterior. We conjecture that it is caused by the implicit regularization effect of the operator $Q_W$. On the other hand, we observe that SGHMCLP-F and SGHMCLP-L have almost the same ECE as full-precision SGHMC in CIFAR-10, showing that low-precision arithmetic does not degrade the calibration ability of SGHMC. In the CIFAR-100 dataset, HMC-based low-precision algorithms outperform their SGLD counterparts, especially SGHMCLP-F, which outperforms SGLDLP-F around 1.4% in fixed point representation for the CIFAR-100 task. For the low-precision gradient a ccumulators method, SGHMCLP-L and VC SGHMCLP-L achieve comparable or better ECE with SGLDLP-L and VC SGLDLP-L and dramatically outperform low-precision SGD.

Table 2: Test errors (%) of full-precision gradient accumulators on CIFAR with ResNet-18. SGHMCLP-F achieves comparable results with SGLDLP-F.

|  | CIFAR-10 | CIFAR-100 |
|---|---|---|
| **32-bit Float** | | |
| SGD | $4.73 \pm 0.10$ | $\mathbf{22.34} \pm \mathbf{0.22}$ |
| SGLD | $\mathbf{4.52} \pm \mathbf{0.07}$ | $22.40 \pm 0.04$ |
| SGHMC | $4.78 \pm 0.08$ | $22.37 \pm 0.04$ |
| **8-bit Fixed Point** | | |
| SGD | $5.19 \pm 0.09$ | $23.71 \pm 0.18$ |
| SGLD | $\mathbf{5.07} \pm \mathbf{0.04}$ | $\mathbf{23.36} \pm \mathbf{0.10}$ |
| SGHMC | $5.08 \pm 0.08$ | $23.54 \pm 0.10$ |
| **8-bit Block Floating Point** | | |
| SGD | $4.75 \pm 0.21$ | $22.86 \pm 0.14$ |
| SGLD | $\mathbf{4.58} \pm \mathbf{0.07}$ | $22.70 \pm 0.22$ |
| SGHMC | $4.93 \pm 0.09$ | $\mathbf{22.39} \pm \mathbf{0.11}$ |

Table 3: ECE (%) of full-precision gradient accumulators on CIFAR with ResNet-18. SGHMCLP-F achieves comparable ECE with SGLDLP-F.

|  | CIFAR-10 | CIFAR-100 |
|---|---|---|
| **32-bit Float** | | |
| SGD | 2.50 | 4.97 |
| SGLD | 1.12 | 3.71 |
| SGHMC | **0.72** | **1.52** |
| **8-bit Fixed Point** | | |
| SGD | 2.79 | 7.11 |
| SGLD | **0.86** | 3.57 |
| SGHMC | 1.11 | **1.92** |
| **8-bit Block Floating Point** | | |
| SGD | 2.43 | 5.97 |
| SGLD | 1.01 | 3.87 |
| SGHMC | 1.12 | **3.65** |

Table 4: Test errors (%) of low-precision gradient accumulators on CIFAR with ResNet-18. SGHMCLP-L and VC SGHMCLP-L outperform SGLDLP-L and VC SGLDLP-L, respectively. SGHMCLP-L achieves comparable results with VC SGLDLP-L.

|  | CIFAR-10 | CIFAR-100 |
|---|---|---|
| **32-bit Float** | | |
| SGD | $4.73 \pm 0.10$ | $\mathbf{22.34} \pm \mathbf{0.22}$ |
| SGLD | $\mathbf{4.52} \pm \mathbf{0.07}$ | $22.40 \pm 0.04$ |
| SGHMC | $4.78 \pm 0.08$ | $22.37 \pm 0.04$ |
| **8-bit Fixed Point** | | |
| SGD | $8.50 \pm 0.22$ | $28.42 \pm 0.35$ |
| SGLD | $7.81 \pm 0.07$ | $27.15 \pm 0.35$ |
| VC SGLD | $7.03 \pm 0.23$ | $26.73 \pm 0.12$ |
| SGHMC | $6.63 \pm 0.10$ | $26.57 \pm 0.10$ |
| **VC SGHMC** | $\mathbf{6.60} \pm \mathbf{0.06}$ | $\mathbf{26.43} \pm \mathbf{0.19}$ |
| **8-bit Block Floating Point** | | |
| SGD | $5.86 \pm 0.18$ | $26.75 \pm 0.11$ |
| SGLD | $5.75 \pm 0.05$ | $26.11 \pm 0.38$ |
| VC SGLD | $5.51 \pm 0.01$ | $25.14 \pm 0.11$ |
| SGHMC | $5.38 \pm 0.06$ | $25.29 \pm 0.03$ |
| **VC SGHMC** | $\mathbf{5.15} \pm \mathbf{0.08}$ | $\mathbf{24.45} \pm \mathbf{0.16}$ |

Table 5: ECE (%) of low-precision gradient accumulators on CIFAR with ResNet-18. Low-precision SGHMC are less affected by the quantization error.

|  | CIFAR-10 | CIFAR-100 |
|---|---|---|
| **32-bit Float** | | |
| SGD | 2.50 | 4.97 |
| SGLD | 1.12 | 3.71 |
| SGHMC | **0.72** | **1.52** |
| **8-bit Fixed Point** | | |
| SGD | 5.12 | 12.92 |
| SGLD | 1.67 | **1.11** |
| VC SGLD | **0.60** | 2.89 |
| SGHMC | 0.72 | 2.46 |
| **VC SGHMC** | 0.70 | 2.44 |
| **8-bit Block Floating Point** | | |
| SGD | 4.62 | 13.93 |
| SGLD | 0.67 | 5.63 |
| VC SGLD | **0.60** | 5.09 |
| SGHMC | 0.78 | **4.94** |
| **VC SGHMC** | 0.67 | 5.02 |

## 5 Conclusion

We provide the first comprehensive investigation for low-precision SGHMC in both strongly log-concave and non-log-concave target distributions with several variants of low-precision training. In particular, we prove that for non-log-concave distributions, low-precision SGHMC with full-precision, low-precision, and variance-corrected gradient accumulators all achieve an acceleration in iterations and have a better convergence upper bound w.r.t the quantization error compared to low-precision SGLD counterparts. Moreover, we study the improvement of variance-corrected quantization applied to low-precision SGHMC under different cases. Under certain conditions, the naïve SGHMCLP-L can replace the VC SGLDLP-L to get comparable results,

saving more computation resources. We conduct empirical experiments on Gaussian, Gaussian mixture distribution, logistic regression, and Bayesian deep learning tasks to justify our theoretical findings.

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

# A   Additional Results for Low-precision Stochastic Gradients Hamiltonian Monte Carlo

In this section, we mainly summarize the theoretical results of Low-precision SGHMC under strongly log-concave target distribution. The underdamped Langevin dynamics can be defined as:

$$
\begin{aligned}
d\mathbf{v}_t &= -\gamma\mathbf{v}_t dt - u\nabla U(\mathbf{x}_t)dt + \sqrt{2\gamma u}d\mathbf{B}_t \\
d\mathbf{x}_t &= \mathbf{v}_t dt,
\end{aligned}
\tag{10}
$$

where $(\mathbf{x}_t, \mathbf{v}_t) \in \mathbb{R}^{2d}$, and $u$, $\gamma$ denote the hyperparameters of inverse mass and friction respectively. We introduce the the strongly-log-concave assumption as:

**Assumption 4** (Strongly Log-Convex). *The energy function $U$ is $m$-strongly log-convex, i.e., there exists a positive constant $m$ such that,*

$$
U(\mathbf{y}) \geq U(\mathbf{x}) + \langle\nabla U(\mathbf{x}), \mathbf{y} - \mathbf{x}\rangle + \frac{m_1}{2}\|\mathbf{y} - \mathbf{x}\|^2, \quad \text{for any } \mathbf{x}, \mathbf{y} \in \mathbb{R}^d.
$$

Once we introduce the continuous underdamped Langevin dynamics (10), we are ready to find a contraction rate for (10). According to

**Theorem 4.** *Suppose Assumptions 1, 3, and 4 hold and the minimum satisfies $\|\mathbf{x}^*\|^2 < \mathcal{D}^2$. Furthermore, let $p^*$ denote the target distribution of $\mathbf{x}$ and $\mathbf{v}$. Given any sufficiently small $\epsilon$, if we set the step size to be*

$$
\eta = \min\left\{\frac{\epsilon\kappa_1^{-1}}{\sqrt{479232/5(d/m_1 + \mathcal{D}^2)}}, \frac{\epsilon^2}{1440\kappa_1 u^2\left[(M^2 + 1)\frac{\Delta^2 d}{4} + \sigma^2\right]}\right\},
$$

*then after $K$ steps starting with initial points $\mathbf{x}_0 = \mathbf{v}_0 = 0$, the output $(\mathbf{x}_K, \mathbf{v}_K)$ of the SGHMCLP-F in (5) satisfies*

$$
\mathcal{W}_2(p(\mathbf{x}_K, \mathbf{v}_K), p^*) \leq \tilde{\mathcal{O}}(\epsilon + \Delta),
$$

*for some $K$ satisfying*

$$
K \leq \frac{\kappa_1}{\eta}\log\left(\frac{36\left(\frac{d}{m_1} + \mathcal{D}^2\right)}{\epsilon}\right) = \tilde{\mathcal{O}}\left(\epsilon^{-2}\log\left(\epsilon^{-1}\right)\Delta^2\right).
$$

Theorem 1 in Zhang et al. (2022) implies that for strongly log-concave target distribution, the low-precision SGLD with full-precision gradient accumulators can achieve $\epsilon$ accuracy within $\tilde{\mathcal{O}}\left(\epsilon^{-2}\log\left(\epsilon^{-1}\right)\Delta^2\right)$ iterations. Thus, the theorem of SGHMCLP-F does not showcase any advantage over SGLDLP-F. This is not surprising, since the quantization applied to the gradients in the full-precision gradient accumulator algorithm is equivalent to adding extra noise to the stochastic gradients. As theoretically shown by Cheng et al. (2018) for strongly-log-concave target distribution, SGHMC doesn't exhibit any advantage over the overdamped Langevin algorithm when stochastic gradients are used. Now we present the convergence analysis of SGHMCLP-L under strongly log-concave target distributions.

**Theorem 5.** *Let Assumption 1, 4 and 3 hold and the minimum satisfies $\|\mathbf{x}^*\|^2 < \mathcal{D}^2$. Furthermore, let $p^*$ denote the target distribution of $\mathbf{v}$ and $\mathbf{x}$. Given any sufficiently small $\epsilon$, if we set the step size $\eta$ to be*

$$
\eta = \min\left\{\frac{\epsilon\kappa_1^{-1}}{\sqrt{663552/5\left(\frac{d}{m_1} + \mathcal{D}^2\right)}}, \frac{\epsilon^2}{2880\kappa_1 u\left(\frac{\Delta^2 d}{4} + \sigma^2\right)}\right\},
$$

*then after $K$ steps starting with initial points $\mathbf{x}_0 = \mathbf{v}_0 = 0$, the output $(\mathbf{x}_K, \mathbf{v}_K)$ of the SGHMCLP-L in (6) satisfies*

$$
\mathcal{W}_2(p(\mathbf{x}_K, \mathbf{v}_K), p^*) = \tilde{\mathcal{O}}\left(\epsilon + \frac{\Delta}{\epsilon}\right),
\tag{11}
$$

*for some $K$ such that*

$$K \leq \frac{\kappa_1}{\eta} \log \left( \frac{36 \left( \frac{d}{m_1} + \mathcal{D}^2 \right)}{\epsilon} \right) = \tilde{\mathcal{O}} \left( \epsilon^{-2} \log \left( \epsilon^{-1} \right) \Delta^2 \right).$$

Comparing Theorem 4 and Theorem 5, we show that for strongly log-concave target distribution the naïve SGHMCLP-L has worse convergence upper bound than SGHMCLP-F. Since SGHMCLP-L directly quantizes the weights after each update, a small stepsize update is often quantized to zero, resulting in the sample distribution converging to a Dirac distribution at the initial point. In such cases, ensuring convergence becomes challenging. Compared with Theorem 2 in Zhang et al. (2022), We cannot show the advantages of low-precision SGHMC over SGLD. Next, we present the theorem for VC SGHMCLP-L under strongly log-concave target distribution.

**Theorem 6.** *Let Assumption 1, 4 and 3 hold and the minimum satisfies $\|\mathbf{x}^*\|^2 < \mathcal{D}^2$. Furthermore, let $p^*$ denote the target distribution of $\mathbf{x}$ and $\mathbf{v}$. Given any sufficiently small $\epsilon$, if we set the stepsize to be*

$$\eta = \min \left\{ \frac{\epsilon}{479232/5 \left( \frac{d}{m_1} + \mathcal{D}^2 \right) \kappa_1}, \frac{\epsilon^2}{90u^2 \Delta^2 d\kappa_1 + 360u^2 \sigma^2 \kappa_1} \right\}$$

*after $K$ steps starting from the initial point $\mathbf{x}_0 = \mathbf{v}_0 = 0$ the output $(\mathbf{x}_K, \mathbf{v}_K)$ of the VC SGHMCLP-L in (9) satisfies*

$$\mathcal{W}_2(p(\mathbf{x}_K, \mathbf{v}_K), p^*) = \tilde{\mathcal{O}} \left( \epsilon + \sqrt{\Delta} \right), \tag{12}$$

*for some $K$ satisfied*

$$K \leq \frac{\kappa_1}{\eta} \log \left( \frac{36 \left( \frac{d}{m_1} + \mathcal{D}^2 \right)}{\epsilon} \right) = \tilde{\mathcal{O}} \left( \epsilon^{-2} \log \left( \epsilon^{-1} \right) \Delta^2 \right).$$

Theorem 6 shows that the variance corrected quantization function can solve the overdispersion problem we observe for the naïve SGHMCLP-L algorithm for strongly log-concave distribution. The $\mathcal{W}_2$ distance between the sample distribution and target distribution can be arbitrarily close to $\tilde{\mathcal{O}}(\sqrt{\Delta})$. Compared to the Theorem 3 in Zhang et al. (2022), the VC SGHMCLP-L doesn't showcase its advantage over VC SGLDLP-L for strongly log-concave distribution.

## B  Stochastic Gradient Langevin Dynamics Result

In order to sample from the target distribution, Langevin dynamics-based samplers, such as overdamped Langevin MCMC and underdamped Langevin MCMC methods, are widely used when the evaluation of $U(\mathbf{x})$ is expansive due to a large sample size. The continuous-time overdamped Langevin MCMC can be represented by the following stochastic differential equation(SDE):

$$d\mathbf{x}_t = -\nabla U(\mathbf{x}_t) + \sqrt{2}d\mathbf{B}_t, \tag{13}$$

where $\mathbf{B}_t$ represents the standard Brownian motion in $\mathbb{R}^d$. Under some mild conditions, it can be proved that the invariant distribution of (13) converges the target distribution $\exp(-U(\mathbf{x}))$. To reduce the computational cost of evaluating $\nabla U(\mathbf{x})$, Welling & Teh (2011) proposed the Stochastic Gradient Langevin Dynamics (SGLD) and updates the weights using stochastic gradients:

$$\mathbf{x}_{k+1} = \mathbf{x}_k - \eta \nabla \tilde{U}(\mathbf{x}_k) + \sqrt{2\eta}\xi_{k+1}, \tag{14}$$

where $\eta$ is the stepsize, the $\xi_{k+1}$ is a standard Gaussian noise, and $\nabla \tilde{U}(\mathbf{x}_k)$ is an unbiased estimation of $\nabla U(\mathbf{x}_k)$. Despite the additional noise induced by stochastic gradient estimations, SGLD can still converge to the target distribution.

In this section, we present the theoretical result for SGLD. We start from the SGLDLP-F's result.

**Theorem 7.** *Suppose Assumptions 1, 2 and 3 hold. Let $p^*$ denote the target distribution of $\mathbf{x}$, $\widetilde{A}$ have the same definition in Theorem 1. After $K$ steps starting with initial point $\mathbf{x}_0 = 0$, if we set the stepsize to be $\eta = \tilde{\mathcal{O}}\left(\left(\frac{\epsilon}{\log(1/\epsilon)}\right)^4\right)$. The output $\mathbf{x}_K$ of SGLDLP-F in (1) satisfies*

$$\mathcal{W}_2(p(\mathbf{x}_K), p^*) \leq \tilde{\mathcal{O}}\left(\epsilon + \widetilde{A}\log\left(\frac{1}{\epsilon}\right)\right), \tag{15}$$

*for some $K$ satisfied*

$$K = \tilde{\mathcal{O}}\left(\frac{1}{\epsilon^4\lambda^*}\log^5\left(\frac{1}{\epsilon}\right)\right).$$

Theorem 7 shows that the low-precision SGLD with full-precision gradient accumulators can converge to the non-log-concave target distribution if provided a small gradient variance and quantization error. Next, we present the SGLDLP-L's result.

**Theorem 8.** *Let Assumptions 1, 2 and 3 hold. Let $p^*$ denote the target distribution of $\mathbf{x}$. If we set the step size to be $\eta = \tilde{\mathcal{O}}\left(\left(\frac{\epsilon}{\log(1/\epsilon)}\right)^4\right)$, after $K$ steps starting at the initial point $\mathbf{x}_0 = 0$ the output $\mathbf{x}_K$ of the SGLDLP-L in (2) satisfies*

$$\mathcal{W}_2(p(\mathbf{x}_K), p^*) = \tilde{\mathcal{O}}\left(\epsilon + \sqrt{\max\{\sigma^2, \sigma\}}\log\left(\frac{1}{\epsilon}\right) + \frac{\log^5\left(\frac{1}{\epsilon}\right)}{\epsilon^4}\sqrt{\Delta}\right), \tag{16}$$

*for some $K$ satisfied*

$$K = \tilde{\mathcal{O}}\left(\frac{1}{\epsilon^4\lambda^*}\log^5\left(\frac{1}{\epsilon}\right)\right).$$

The VC SGLDLP-L can be done as:

$$\mathbf{x}_{k+1} = Q^{vc}\left(\mathbf{x}_k - \eta Q_G(\nabla\tilde{U}(\mathbf{x}_k)), 2\eta, \Delta\right) \tag{17}$$

We present the convergence analysis of VC SGLDLP-L in the following theorem:

**Theorem 9.** *Let Assumption 1, 2 and 3 hold. Let $p^*$ denote the target distribution of $\mathbf{x}$. If we set the stepsize to be $\eta = \tilde{\mathcal{O}}\left(\frac{\epsilon^4}{\log^4\left(\frac{1}{\epsilon}\right)}\right)$, after $K$ steps from the initial point $\mathbf{x}_0 = 0$ the output $\mathbf{x}_K$ of VC SGLDLP-L in (17) satisfies*

$$\mathcal{W}_2(p(\mathbf{x}_K), p^*) = \tilde{\mathcal{O}}\left(\epsilon + \sqrt{\max\{\sigma^2, \sigma\}\log\left(\frac{1}{\epsilon}\right)} + \frac{\log^3\left(\frac{1}{\epsilon}\right)}{\epsilon^2}\sqrt{\Delta}\right), \tag{18}$$

*for some $K$ satisfied*

$$K = \tilde{\mathcal{O}}\left(\frac{1}{\epsilon^4\lambda^*}\log^5\left(\frac{1}{\epsilon}\right)\right).$$

## C   Uniform Bound of Contraction Rate

According to reference (Raginsky et al., 2017), under Assumptions 1 and 2, one can choose $\lambda^*$ to be the uniform lower bound of the contraction rate, i.e.,

$$\lambda^* := \inf\left\{\frac{\int_{\mathbb{R}^d}\|\nabla g\|^2\,dp^*}{\int_{\mathbb{R}^d}g^2\,dp^*} : g \in \mathcal{C}^1\left(\mathbb{R}^d\right) \cap L^2(p^*), g = 0, \int_{\mathbb{R}^d}g\,dp^* = 0\right\},$$

which satisfied

$$\frac{1}{\lambda^*} \leq \frac{2}{m_2(d+b)} + \frac{4C(d+b)}{m_2}exp\left(\frac{2}{m_2}(M+B)(b+d) + (A+B)\right),$$

where $A$, $B$ denote bounds such that $|U(0)| \leq A, \|\nabla \widetilde{U}(0)\| \leq B$. In other words, asymptotically w.r.t. the dimension, we have $\lambda^{*-1} = \exp(\mathcal{O}(d))$.

Similarly, (Zou et al., 2019) derives a contraction rate as

$$\mu^* = \frac{2d}{768\gamma e^\Lambda} \min\left\{\lambda M u e^\Lambda, \Lambda^{1/2} M u, \gamma \Lambda^{1/2}\right\},$$

where the constants are defined as:

$$\lambda = \frac{2m_2}{4M + u^{-1}\gamma^2}$$
$$\Lambda = \frac{12(1 + 2\alpha + 2\alpha^2)(d + \mathcal{A})Mu}{5\gamma^2\lambda(1 - 2\lambda)}$$
$$\mathcal{A} = \frac{2m_2(U(x^*) + M\|x^*\|^2)}{4M + u^{-1}\gamma^2} + \frac{b}{2}.$$

Note that the above rate also satisfies $\mu^{*-1} = \exp(\mathcal{O}(d))$.

## D   Technical Detail

In this section, we disclose more details of empirical experiments. We can define the stochastic quantization function $Q^s$ as:

$$Q^s(\theta) = \begin{cases} \Delta \left\lfloor \frac{\theta}{\Delta} \right\rfloor, & \text{w.p. } \left\lceil \frac{\theta}{\Delta} \right\rceil - \frac{\theta}{\Delta} \\ \Delta \left\lceil \frac{\theta}{\Delta} \right\rceil, & \text{w.p. } 1 - \left(\left\lceil \frac{\theta}{\Delta} \right\rceil - \frac{\theta}{\Delta}\right). \end{cases} \tag{19}$$

In practice, to implement stochastic rounding based on the rule (19), the computer still needs a full-precision $\text{Unif}(0,1)$ random number generator (note that we ignore the discretization gap between full precision values and real values), then compares this random number with the residual and rounds up if it is smaller otherwise rounds down. This full precision random number generator can be shared for all rounding steps, hence won't affect memory usage too much. For more details about the implementation of stochastic rounding, please refer to (Gupta et al., 2015; Croci et al., 2022).

Now, we show the details of the experiment setup. For the standard normal distribution experiment, we use 8-bit fixed point low-precision representation with 4 of them representing fractional parts. Moreover, we set the step size $\eta = 0.09$, inverse mass $u = 2$, and friction $\gamma = 3$. Similarly, for Gaussian mixture distribution, we also use 8-bit fixed point low-precision representation with 4 of them representing fractional parts for both low-precision SGHMC and SGLD, but we set the step size $\eta = 0.1$, inverse mass $u = 1$, and friction $\gamma = 3$.

Next, for both logistic, MLP models, low-precision SGLD and SGHMC in MNIST task, we set $\mathcal{N}\left(0, 10^{-2}\right)$ as the prior distribution, and step size $\eta = 0.01$. Moreover, for SGHMC, we set the inverse mass $u = 2$, and friction $\gamma = 2$.

Then we introduce the training detail of low-precision SGHMC for CIFAR-10 & CIFAR-100. We adopt the quantization framework from previous research Wu et al. (2018); Wang et al. (2018); Yang et al. (2019) to apply quantization to weights, activations, backpropagation errors, and gradients. Please see the Algorithm 1. We use $\mathcal{N}\left(0, 10^{-4}\right)$ as the prior distribution. Furthermore, we set the set the step size $\eta = 0.1$, and $u = 2, \gamma = 2$ for low-precision SGHMC.

Algorithm 1 is a practical version of the three different types of low-precision SGHMC updates proposed in our main text, i.e. equations (5), (6), and (9). Additional components in the algorithm include (i) How to compute the stochastic gradient via forward/back-propagation (steps colored by red in the Algorithm box). Due to the low-precision nature of the algorithm, we also quantize all intermediate results along the propagation process by proper quantizers $Q_A$ and $Q_E$. (ii) Additional optional scale/re-scale step (colored

by blue in the Algorithm box). The reason for adding this step is that: in practice, we found that the momentum term $\mathbf{v}_k$ tends to be close to 0. When $\mathbf{v}_k$ is represented in the low-precision fixed-point format, the information carried by $\mathbf{v}_k$ is lost since the low-precision fixed-point is loose around 0, and only 2 or 3 bits are used representing the momentum (i.e., the other bits are wasted). With this observation, we store a scaled-up momentum to fully utilize all bits, thus the information carried will be kept in an optimal way.

Algorithm 2 is proposed by (Zhang et al., 2022). The rationale behind Algorithm 2 is that: if we directly quantize the SGLD update result, i.e. the mean shift plus a Gaussian noise variable, it essentially introduces an additional quantization noise to the sample, leading to a larger sampling variance. Instead of quantizing the mean shift plus Gaussian noise, we can first quantize the mean shift, then plus a low-precision discrete random variable. In this way, we guarantee the sampler yields low-precision values and have the freedom to design the variance of the low-precision discrete random variable, such that overall sampling variance (i.e., variance due to stochastic round and low-precision discrete random variable) matches the idea sampling variance of full-precision Langevin update.

When implementing low-precision SGHMC on classification tasks in the MNIST, CIFAR-10 and CIFAR-100 dataset, we observed that the momentum term $\mathbf{v}$ tend to gather in a small range around zero in which case the low-precision representations of $\mathbf{v}$ end up in using few bits, thus the momentum information is seriously lost and cause in performance degradation. In order to tackle this problem and fully utilize all the low-precision representations, we borrowed the idea of rescaling from the bit-centering trick and adopted the low-precision SGHMC method. The detailed algorithm is listed in Algorithms 1.

Now, we give a brief introduction of the variance-corrected quantization function $Q^{vc}$. Instead of adding real value Gaussian noise and quantizing the weights, we can design a categorical sampler that samples from the space $\{\Delta, -\Delta, 0\}$ with the desired expectation $\mu$ and variance $v$ as

$$\mathrm{Cat}(\mu, v) = \begin{cases} \Delta, & w.p. \frac{v + \mu^2 + \mu\Delta}{2\Delta^2} \\ -\Delta, & w.p. \frac{v + \mu^2 - \mu\Delta}{2\Delta^2} \\ 0, & \text{otherwise.} \end{cases} \tag{20}$$

Based on the sampler (20), one can design the variance-corrected quantization function $Q^{vc}$ in the Algorithm 2.

# E  Proof of Main Theorems

## E.1  Proof of Theorem 1

In this section we analyze the Wasserstein distance between the sample $(\mathbf{x}_k, v_K)$ in (5) and the target distribution, given the target distribution satisfies the assumption 1 and 2. We follow the proof in Raginsky et al. (2017). To analyze the Wasserstein distance, we first calculate the distance between solutions of low-precision discrete underdamped Langevin dynamics and solutions of the ideal continuous underdamped Langevin dynamics, also the distance between solutions of the ideal continuous underdamped Langevin dynamics and the target distribution.

Again let $p_k = (\mathbf{x}_k, v_k)$ denote the low-precision sample from (5) at $k$-th iteration, let $\hat{p}_t = (\hat{x}_t, \hat{v}_t)$ denote the sample from the ideal continuous underdamped Langevin dynamics in (41) at time $t$. Then the Wasserstein distance between the $p_k$ and the target distribution $p^*$ can be bounded as:

$$\mathcal{W}_2(p_K, p^*) \leq \mathcal{W}_2(p_K, \hat{p}_{K\eta}) + \mathcal{W}_2(\hat{p}_{K\eta}, p^*).$$

Then we bound the first term $\mathcal{W}_2(p_K, \hat{p}_{K\eta})$ by invoking the weighted CKP inequality Bolley & Villani (2005),

$$\mathcal{W}_2^2(p_K, \hat{p}_{K\eta}) \leq \Lambda \left( \sqrt{D_{KL}(p_K || \hat{p}_{K\eta})} + \sqrt[4]{D_{KL}(p_K || \hat{p}_{K\eta})} \right),$$

---

**Algorithm 1** Low-Precision Training for SGHMC.

---

**given:** $L$ layers DNN $\{f_1 \ldots, f_L\}$. Weight, gradient, activation, and error quantizers $Q_W, Q_G, Q_A, Q_E$. Variance-corrected quantization $Q^{vc}$, and quantization gap of weights $\Delta$. Data batch sequence $\{(\theta_k, h_k)\}_{k=1}^K$, where the $\theta_k$ is the input, and $h_k$ is the target. The loss function $\mathcal{L}(\mathbf{a}, \mathbf{h})$ measures the loss between the prediction $\mathbf{a}$ and target $\mathbf{h}$. And $\mathbf{x}_k^{fp}$ denotes the full-precision buffer of the weight. Let $\mathrm{Var}_{\mathbf{v}}^{hmc} = u(1 - e^{-2\gamma\eta})$ and $\mathrm{Var}_{\mathbf{x}}^{hmc} = u\gamma^{-2}(2\gamma\eta + 4e^{-\gamma\eta} - e^{-2\gamma\eta} - 3)$ and $S_{\mathbf{v}} = 1$. {Initialize the scaling parameter}

**for** $k = 1 : K$ **do**

  **1. Forward Propagation:**

    $a_k^{(0)} = \theta_k$

    $a_k^{(l)} = Q_A(f_l(a_k^{(l-1)}, \mathbf{x}_k^l)), \forall l \in [1, L]$

  **2. Backward Propagation:**

    $e^{(L)} = \nabla_{a_k^{(L)}} \mathcal{L}(a_k^{(L)}, h_k)$

    $e^{(l-1)} = Q_E\left(\frac{\partial f_l(a_k^{(l)})}{\partial a_k^{(l-1)}} e_k^{(l)}\right), \forall l \in [1, L]$

    $g_k^{(l)} = Q_G\left(\frac{\partial f_l}{\partial \theta_k^{(l)}} e_k^{(l)}\right), \forall l \in [1, L]$

  **3. SGHMC Update:**

    **full-precision gradient accumulators:**

    $\mathbf{v}_{k+1}^{(l)} \leftarrow \mathbf{v}_k^{(l)} - u\gamma^{-1}(1 - e^{-\gamma\eta})g_k^{(l)} + \xi_k^{\mathbf{v}}, \forall l \in [1, L],$

    $\mathbf{x}_{k+1}^{(l),fp} \leftarrow \mathbf{x}_k^{(l),fp} + \gamma^{-1}(1 - e^{-\gamma\eta})\mathbf{v}_k^{(l)} + u\gamma^{-2}(\gamma\eta + e^{-\gamma\eta} - 1)g_k^{(l)} + \xi_k^{\mathbf{x}}, \quad \mathbf{x}_{k+1}^{(l)} \leftarrow Q_W\left(\mathbf{x}_{k+1}^{(l),fp}\right), \forall l \in [1, L]$

    **low-precision gradient accumulators:**

    $\mathbf{v}_k^{(l)} = \mathbf{v}_k^{(l)} * S_{\mathbf{v}}^{(l)}, \forall l \in [1, L]$ {Restore the velocity before update}

    $\mu(\mathbf{v}_{k+1}^{(l)}) \leftarrow \mathbf{v}_k^{(l)} e^{-\gamma\eta} - u\gamma^{-1}(1 - e^{-\gamma\eta})g_k^{(l)}, \forall l \in [1, L]$

    $S_{\mathbf{v}}^{(l)} = \frac{\left\|\mu(\mathbf{v}_{k+1}^{(l)})\right\|_\infty}{U}, \forall l \in [1, L]$ {Update the Scaling}

    $\mathbf{v}_{k+1}^{(l)} \leftarrow Q_W((\mu(\mathbf{v}_{k+1}^{(l)}) + \xi_k^{\mathbf{v}})/S_{\mathbf{v}}^{(l)}), \forall l \in [1, L]$

    $\mathbf{x}_{k+1}^{(l)} \leftarrow Q_W\left(\mathbf{x}_k^{(l)} + \gamma^{-1}(1 - e^{-\gamma\eta})\mathbf{v}_k^{(l)} + u\gamma^{-2}(\gamma\eta + e^{-\gamma\eta} - 1)g_k^{(l)} + \xi_k^{\mathbf{x}}\right), \forall l \in [1, L]$

    **Variance-corrected low-precision gradient accumulators:**

    $\mathbf{v}_k^{(l)} = \mathbf{v}_k^{(l)} * S_v^{(l)}, \forall l \in [1, L]$ {Restore the velocity before update}

    $\mu(\mathbf{v}_{k+1}^{(l)}) = \mathbf{v}_k^{(l)} e^{-\gamma\eta} - u\gamma^{-1}(1 - e^{-\gamma\eta})g_k^{(l)}, \forall l \in [1, L]$

    $\mu(\mathbf{x}_{k+1}^{(l)}) = \mathbf{x}_k^{(l)} + \gamma^{-1}(1 - e^{-\gamma\eta})\mathbf{v}_k^{(l)} + u\gamma^{-2}(\gamma\eta + e^{-\gamma\eta} - 1)g_k^{(l)}, \forall l \in [1, L]$

    $S_{\mathbf{v}}^{(l)} = \frac{\left\|\mu(\mathbf{v}_{k+1}^{(l)})\right\|_\infty}{U}, \forall l \in [1, L]$ {Update the Scaling}

    $\mathbf{v}_{k+1}^{(l)} \leftarrow Q^{vc}\left(\mu(\mathbf{v}_{k+1}^{(l)})/S_{\mathbf{v}}^{(l)}, Var_{\mathbf{v}}^{hmc}/(S_{\mathbf{v}}^{(l)})^2, \Delta\right), \forall l \in [1, L]$

    $\mathbf{x}_{k+1}^{(l)} \leftarrow Q^{vc}\left(\mu(\mathbf{x}_{k+1}^{(l)}), Var_{\mathbf{x}}^{hmc}, \Delta\right), \forall l \in [1, L]$

**end for**

**output:** samples $\{(\mathbf{v}_k^{(l)}, \mathbf{x}_k^{(l)})\}$

---

where $\Lambda = 2\inf_{\theta>0}\sqrt{1/\theta\left(3/2 + \log\mathbb{E}_{\hat{p}_{K\eta}}\left[exp(\theta(\|\hat{x}_{K\eta}\|^2 + \|\hat{v}_{K\eta}\|^2))\right]\right)}$. We define a Lyapunov function for every $(x, v) \in \mathbb{R}^d \times \mathbb{R}^d$

$$\mathcal{E}(\mathbf{x}, \mathbf{v}) = \|\mathbf{x}\|^2 + \|\mathbf{x} + 2\mathbf{v}/\gamma\|^2 + 8u(U(\mathbf{x}) - U(\mathbf{x}^*))/\gamma^2.$$

Note that $\|a\|^2 + \|b\|^2 \geq \|a - b\|^2/2$ and $U(x) \geq U(x^*)$, we can have:

$$\mathcal{E}(x, v) \geq \|x\|^2 + \|x + 2v/\gamma\|^2 \geq \max\{\|x\|^2, 2\|v/\gamma\|^2\}.$$

---

**Algorithm 2** Variance-Corrected Quantization Function $Q^{vc}$. (Zhang et al., 2022)

---

    **input**: $(\mu, v, \Delta)$  {$Q^{vc}$ returns a variable with mean $\mu$ and variance $v$}
    $v_0 \leftarrow \Delta^2/4$    {$\Delta^2/4$ is the largest possible variance that stochastic rounding can cause}
    **if** $v > v_0$ **then** {add a small Gaussian noise and sample from the discrete grid to make up the remaining variance}
        $x \leftarrow \mu + \sqrt{v - v_0}\xi$, where $\xi \sim \mathcal{N}(0, I_d)$
        $r \leftarrow x - Q^d(x)$
        **for all** $i$ **do**
            **sample** $c_i$ from $\mathrm{Cat}(|r_i|, v_0)$ as in (20)
        **end for**
        $\theta \leftarrow Q^d(x) + \mathrm{sign}(r) \odot c$
    **else** {sample from the discrete grid to achieve the target variance}
        $r \leftarrow \mu - Q^s(\mu)$
        **for all** $i$ **do**
            $v_s \leftarrow \left(1 - \frac{|r_i|}{\Delta}\right) \cdot r_i^2 + \frac{|r_i|}{\Delta} \cdot (-r_i + \mathrm{sign}(r_i)\Delta)^2$
            **if** $v > v_s$ **then**
                **sample** $c_i$ from $\mathrm{Cat}(0, v - v_s)$ as in (20)
                $\theta_i \leftarrow Q^s(\mu)_i + c_i$
            **else**
                $\theta_i \leftarrow Q^s(\mu)_i$
            **end if**
        **end for**
    **end if**
    clip $\theta$ if outside representable range
    **return** $\theta$

---

Given assumptions 4 and 2 hold and apply Lemma B.4 in Zou et al. (2019), we can get

$$
\begin{aligned}
\Lambda \leq &2 \inf_{0 < \theta \leq \min\{\frac{\gamma}{128u}, \frac{m_2}{32}\}} \sqrt{\frac{1}{\theta}\left(\frac{3}{2} + 2\theta\mathcal{E}(\mathbf{X}_0, \mathbf{V}_0) + \frac{32M\theta u(4d + 2b + m_2\|\mathbf{x}^*\|^2)}{\gamma^2 m_2}\right)} \\
\leq &2\sqrt{2\mathcal{E}(\mathbf{X}_0, \mathbf{V}_0) + \frac{32M\theta u(4d + 2b + m_2\|\mathbf{x}^*\|^2) + 16(12um_2 + 3\gamma^2)}{\gamma^2 m_2}} := \bar{\Lambda}.
\end{aligned}
$$

It remains to bound the divergence between the distribution $p_K$ and $\hat{p}_{K\eta}$. We first define a continuous interpolation of the low-precision sample $(\mathbf{x}_k, \mathbf{v}_k)$,

$$
\begin{aligned}
d\mathbf{v}_t &= -\gamma\mathbf{v}_t dt - uG_t dt + \sqrt{2\gamma u}dB_t \tag{21} \\
d\mathbf{x}_t &= \mathbf{v}_t dt, \tag{22}
\end{aligned}
$$

where $G_t = \sum_{k=0}^{K} \tilde{g}(\mathbf{x}_k)\mathbf{1}_{t \in [k\eta, (k+1)\eta)}$. Integrating this equation from time 0 to $t$, we can get

$$
\begin{aligned}
\mathbf{v}_t &= \mathbf{v}_0 - \int_0^t \gamma\mathbf{v}_s ds - \int_0^t uG_s dt + \int_0^t \sqrt{2\gamma u}dB_s \\
\mathbf{x}_t &= \mathbf{x}_0 + \int_0^t \mathbf{v}_s ds.
\end{aligned}
$$

Notice that when $t = k\eta$, the solution of (21) has the same distribution with the low-precision sample $(\mathbf{x}_k, \mathbf{v}_k)$. Now by Girsanov formula, we can compute the Radon-Nikodym derivative of $\hat{p}_{K\eta}$ with respect to

$p_K$ as follows:

$$\frac{d\hat{p}_{K\eta}}{dp_K} = exp\left\{\sqrt{\frac{\gamma u}{2}}\int_0^t (\nabla U(\mathbf{x}_s) - G_s)d\mathbf{B}s - \frac{\gamma u}{4}\int_0^T \|\nabla U(\mathbf{x}_s) - G_s\|ds\right\}.$$

It follows that

$$D_{KL}(p_K\|\hat{p}_{K\eta}) = \mathbb{E}_{p_K}\left[\log\left(\frac{d\hat{p}_{K\eta}}{dp_K}\right)\right] \tag{23}$$

$$= \frac{\gamma u}{4}\mathbb{E}\int_0^{K\eta}\|\nabla U(\mathbf{x}_s) - G_s\|^2\, ds$$

$$= \frac{\gamma u}{4}\sum_{k=0}^{K-1}\int_{k\eta}^{(k+1)\eta}\mathbb{E}\left[\|\nabla U(\mathbf{x}_s) - G_s\|^2\right]ds$$

$$= \frac{\gamma u}{4}\sum_{k=0}^{K-1}\int_{k\eta}^{(k+1)\eta}\mathbb{E}\left[\|\nabla U(\mathbf{x}_s) - \tilde{g}(\mathbf{x}_k)\|^2\right]ds.$$

Furthermore, in the $k$-th interval, we have

$$\mathbb{E}\left[\|\nabla U(\mathbf{x}_s) - \tilde{g}(\mathbf{x}_k)\|^2\right] \leq 2\mathbb{E}\left[\|\nabla U(\mathbf{x}_s) - \nabla U(\mathbf{x}_k)\|^2\right] + 2\mathbb{E}\left[\|\nabla U(\mathbf{x}_k) - \tilde{g}(\mathbf{x}_k)\|^2\right]. \tag{24}$$

We now bound the first term in the RHS of the (24). By the smooth Assumption1, we have

$$\mathbb{E}\left[\|\nabla U(\mathbf{x}_s) - \nabla U(\mathbf{x}_k)\|^2\right] \leq M^2\mathbb{E}\left[\|\mathbf{x}_s - \mathbf{x}_k\|^2\right].$$

Notice that

$$\mathbf{x}_s = \mathbf{x}_k + \int_{k\eta}^s \mathbf{v}_r dr$$

$$= \mathbf{x}_k + \int_{k\eta}^s\left(\mathbf{v}_{k\eta}e^{-\gamma(r-k\eta)} - u\left(\int_{k\eta}^r e^{-\gamma(r-z)}\tilde{g}(\mathbf{x}_k)dz\right) + \sqrt{2\gamma u}\int_{k\eta}^r e^{-\gamma(r-z)}dB_z\right)dr.$$

This further implies that:

$$\|\mathbf{x}_s - \mathbf{x}_k\|^2 = \left\|\int_{k\eta}^s\left(\mathbf{v}_{k\eta}e^{-\gamma(r-k\eta)} - u\left(\int_{k\eta}^r e^{-\gamma(r-z)}\tilde{g}(\mathbf{x}_k)dz\right) + \sqrt{2\gamma u}\int_{k\eta}^r e^{-\gamma(r-z)}dB_z\right)dr\right\|^2$$

$$\leq 3\left\|\int_{k\eta}^s \mathbf{v}_{k\eta}e^{\gamma(k\eta-r)}dr\right\|^2 + 3\left\|\int_{k\eta}^s\int_{k\eta}^r u\tilde{g}(\mathbf{x}_k)e^{\gamma(z-r)}dzdr\right\|^2 + 6ru\left\|\int_{k\eta}^s\int_0^s e^{-\gamma(r-z)}dB_zdr\right\|^2$$

$$\leq 3\eta^2\|\mathbf{v}_k\|^2 + 3u^2\eta^4\|\tilde{g}(\mathbf{x}_k)\|^2 + 3\left[\frac{u}{\gamma^2}\left(2\gamma(s-k\eta) + 4e^{-\gamma(s-k\eta)} - e^{-2\gamma(s-k\eta)} - 3\right)d\right]$$

$$\leq 3\eta^2\left(\|\mathbf{v}_k\|^2 + u^2\eta^2\|\tilde{g}(\mathbf{x}_k)\|^2 + 2du\right), \tag{25}$$

where we use inequality $1-x \leq e^{-x} \leq 1-x+x^2/2$ for $x > 0$ and $k\eta \leq s \leq (k+1)\eta$ to get the last inequality. Given this analysis we can bound the first term in the RHS of (24)

$$\mathbb{E}\left[\|\nabla U(\mathbf{x}_s) - \nabla U(\mathbf{x}_k)\|^2\right] \leq 3M^2\eta^2\left(\mathbb{E}\|v_k\|^2 + u^2\eta^2\mathbb{E}\|\tilde{g}(\mathbf{x}_k)\|^2 + 2du\right).$$

By lemma 12, the second term in the RHS of (24) can be bounded as:

$$\mathbb{E}\left[\|\nabla U(\mathbf{x}_k) - \tilde{g}(\mathbf{x}_k)\|^2\right] \leq (M^2+1)\frac{\Delta^2 d}{4} + \sigma^2.$$

We need to introduce a lemma to bound the $\sup_k\|\mathbf{x}_k\|^2$, $\sup_k\|v_k\|^2$ and $\sup_k\|\tilde{g}(\mathbf{x}_k)\|^2$.

**Lemma 10.** *Under Assumptions 1 and 2, if we set the step size statisfied the following condition:*

$$\eta \leq min \left\{ \frac{\gamma}{4\left(8Mu + u\gamma + 22\gamma^2\right)}, \sqrt{\frac{4u^2}{4Mu + 3\gamma^2}}, \frac{6\gamma bu}{\left(4Mu + 3\gamma^2\right)d}, \right.$$
$$\left. \frac{1}{8\gamma}, \frac{\gamma m_2}{12(21u + \gamma)M^2}, \frac{8(\gamma^2 + 2u)}{(20u + \gamma)\gamma} \right\},$$

*then for all $k \geq 0$ the $\mathbb{E}\left[\|\mathbf{x}_k\|^2\right]$, $\mathbb{E}\left[\|v_k\|^2\right]$ and $\mathbb{E}\left[\|\tilde{g}(\mathbf{x}_k)\|^2\right]$ can be bounded as*

$$\mathbb{E}\left[\|\mathbf{x}_k\|^2\right] \leq \overline{\mathcal{E}} + C_0\left((M^2 + 1)\frac{\Delta^2 d}{4} + \sigma^2\right)$$

$$\mathbb{E}\left[\|v_k\|^2\right] \leq \gamma^2\overline{\mathcal{E}}/2 + \gamma^2 C_0/2\left((M^2 + 1)\frac{\Delta^2 d}{4} + \sigma^2\right)$$

$$\mathbb{E}\left[\|\tilde{g}(\mathbf{x}_k)\|^2\right] \leq 2\left((M^2 + 1)\frac{\Delta^2 d}{4} + \sigma^2\right) + 4M^2\overline{\mathcal{E}} + 4G^2$$

*where $\overline{\mathcal{E}}$ and $C_0$ are defined as:*

$$\overline{\mathcal{E}} = \mathbb{E}\left[\mathcal{E}(\mathbf{x}_0, \mathbf{v}_0)\right] + \frac{24(21u + \gamma)uM}{m_2\gamma^3}G^2 + \frac{96(d + b)uM}{m_2\gamma^2}, \quad G = \|\nabla U(0)\|$$

$$C_0 = \frac{96u\left(\gamma^2 + 2u\right)}{m_2\gamma^4}.$$

The proof of Lemma 10 can be found in Appendix F.3. We now ready to bound $\mathbb{E}\left[\|\nabla U(\mathbf{x}_s - \tilde{g}(\mathbf{x}_k))\|^2\right]$ as:

$$\mathbb{E}\left[\|\nabla U(\mathbf{x}_s) - \tilde{g}(\mathbf{x}_k)\|^2\right] \leq 2\mathbb{E}\left[\|\nabla U(\mathbf{x}_s) - \nabla U(\mathbf{x}_k)\|^2\right] + 2\mathbb{E}\left[\|\nabla U(\mathbf{x}_k) - \tilde{g}(\mathbf{x}_k)\|^2\right]$$

$$\leq 6M^2\eta^2\left(\mathbb{E}\|v_k\|^2 + u^2\eta^2\mathbb{E}\|\tilde{g}(\mathbf{x}_k)\|^2 + 2du\right) + 2\left((M^2 + 1)\frac{\Delta^2 d}{4} + \sigma^2\right)$$

$$\leq 6M^2\eta^2\left((\gamma^2/2 + 4M^2u^2\eta^2)\overline{\mathcal{E}} + (\gamma^2 C_0/2 + 2u^2\eta^2)\left((M^2 + 1)\frac{\Delta^2 d}{4} + \sigma^2\right) + 4u^2\eta^2 G^2 + 2du\right)$$

$$+ 2\left((M^2 + 1)\frac{\Delta^2 d}{4} + \sigma^2\right)$$

$$\leq 6M^2\eta^2\left[(\gamma^2/2 + 4M^2u^2\eta^2)\overline{\mathcal{E}} + 4u^2\eta^2 G^2 + 2du\right]$$

$$+ \left(6M^2\eta^2(\gamma^2 C_0/2 + 2u^2\eta^2) + 2\right)\left((M^2 + 1)\frac{\Delta^2 d}{4} + \sigma^2\right).$$

Thus the divergence can be bounded as:

$$D_{KL}(p_K\|\hat{p}_{K\eta}) \leq \frac{3\gamma u}{2}M^2 K\eta^3\left[(\gamma^2/2 + 4M^2u^2\eta^2)\overline{\mathcal{E}} + 4u^2\eta^2 G^2 + 2du\right]$$

$$+ \frac{\gamma u}{4}K\eta\left(6M^2\eta^2(\gamma^2 C_0/2 + 2u^2\eta^2) + 2\right)\left((M^2 + 1)\frac{\Delta^2 d}{4} + \sigma^2\right).$$

By the weighted CKP inequality and given $K\eta \geq 1$,

$$\mathcal{W}_2(p_K, \hat{p}_{K\eta}) \leq \overline{\Lambda}\left(\sqrt{D_{KL}(p_K\|\hat{p}_{K\eta})} + \sqrt[4]{D_{KL}(p_K\|\hat{p}_{K\eta})}\right)$$

$$\leq \overline{\Lambda}\left(\widetilde{C_0}\sqrt{\eta} + \widetilde{C_1}\widetilde{A}\right)\sqrt{K\eta},$$

where the constants $\widetilde{C_0}$, $\widetilde{C_1}$ and $\widetilde{A}$ are defined as:

$$\widetilde{C_0} = \sqrt{\frac{3\gamma u}{2} M^2 \left[(\gamma^2/2 + 4M^2 u^2 \eta^2)\overline{\mathcal{E}} + 4u^2 \eta^2 G^2 + 2du\right]} + \sqrt[4]{\frac{3\gamma u}{2} M^2 \left[(\gamma^2/2 + 4M^2 u^2 \eta^2)\overline{\mathcal{E}} + 4u^2 \eta^2 G^2 + 2du\right]}$$

$$\widetilde{C_1} = \sqrt{\frac{\gamma u}{4} \left(6M^2 \eta^2 (\gamma^2 C_0/2 + 2u^2 \eta^2) + 2\right)} + \sqrt[4]{\frac{\gamma u}{4} \left(6M^2 \eta^2 (\gamma^2 C_0/2 + 2u^2 \eta^2) + 2\right)}$$

$$\widetilde{A} = \max\left\{\sqrt{\left((M^2 + 1)\frac{\Delta^2 d}{4} + \sigma^2\right)}, \sqrt[4]{\left((M^2 + 1)\frac{\Delta^2 d}{4} + \sigma^2\right)}\right\}.$$

Finally by the Lemma A.2 in Zou et al. (2019), we can have

$$\mathcal{W}_2(\hat{p}_{K\eta}, p^*) \leq \Gamma_0 e^{-\mu^* K\eta},$$

where $\mu^* = e^{-\widetilde{\mathcal{O}}(d)}$ denotes the concentration rate of the underdamped Langevin dynamics and $\Gamma_0$ is a constant of order $\mathcal{O}(1/\mu^*)$. Combining this inequality with the previous analysis we can prove:

$$\mathcal{W}_2(p_K, p^*) \leq \overline{\Lambda}\left(\widetilde{C_0}\sqrt{\eta} + \widetilde{C_1}\widetilde{A}\right)\sqrt{K\eta} + \Gamma_0 e^{-\mu^* K\eta}. \tag{26}$$

To bound the Wasserstein distance, we need to set

$$\overline{\Lambda}\widetilde{C_0}\sqrt{K\eta^2} = \frac{\epsilon}{2} \quad \text{and} \quad \Gamma_0 e^{-\mu^* K\eta} = \frac{\epsilon}{2}. \tag{27}$$

Solving the equation (27), we can have

$$K\eta = \frac{\log\left(\frac{2\Gamma_0}{\epsilon}\right)}{\mu^*} \quad \text{and} \quad \eta = \frac{\epsilon^2}{4\overline{\Lambda}^2 \widetilde{C_0}^2 K\eta}.$$

Combining these two we can have

$$\eta = \frac{\epsilon^2 \mu^*}{4\overline{\Lambda}^2 \widetilde{C_0}^2 \log\left(\frac{2\Gamma_0}{\epsilon}\right)} \quad \text{and} \quad K = \frac{4\overline{\Lambda}^2 \widetilde{C_0}^2 \log^2\left(\frac{2\Gamma_0}{\epsilon}\right)}{\epsilon^2 (\mu^*)^2}.$$

Plugging in (26) completes the proof.

### E.2 Proof of Theorem 2

In this section, we analyze the convergence of SGHMCLP-L when the target distribution is non-log-concave. In this proof, unlike in the SGHMCLP-F algorithm where gradients are unbiased, additional noise applied to the state $\mathbf{x}$ causes deviation from the underdamped Langevin dynamics, leading us to establish an intermediate process to address this noise.

Recall the continuous interpolation of the SGHMCLP-L,

$$\mathbf{v}_t = \mathbf{v}_0 - \int_0^t \gamma \mathbf{v}_s ds - u \int_0^t G_s ds + \sqrt{2\gamma u} \int_0^t e^{-\gamma(t-s)} dB_s + \int_0^t \alpha_v(s) ds$$

$$\mathbf{x}_t = \mathbf{x}_0 + \int_0^t \mathbf{v}_s ds + \int_0^t \alpha_x(s) ds,$$

where $G_s = \sum_{k=0}^{\infty} Q_G\left(\nabla U(x'_k)\right)\mathbf{1}_{s \in (k\eta, (k+1)\eta)}$. And we define an intermediate process by let $\mathbf{v}'_t = \mathbf{v}_t + \alpha_x(t)$:

$$v'_t = v'_0 - \int_0^t \gamma\left(v'_s - \alpha_x(s)\right)ds - u\int_0^t G_s ds + \sqrt{2\gamma u}\int_0^t e^{-\gamma(t-s)}dB_s + \int_0^t \left(\alpha_v(s) + \frac{1}{t}\alpha_x(t)\right)ds$$

$$x'_t = x'_0 + \int_0^t v'_s ds. \tag{28}$$

By integrating the underdamped Langevin dynamic (10), we can have:

$$\mathbf{v}_t = \mathbf{v}_0 - \int_0^t \gamma\left(\mathbf{v}_s - \alpha_x(s)\right)ds - u\int_0^t \nabla U(\mathbf{x}_s)ds + \sqrt{2\gamma u}\int_0^t e^{-\gamma(t-s)}dB_s$$

$$\mathbf{x}_t = \mathbf{x}_0 + \int_0^t \mathbf{v}_s ds. \tag{29}$$

Notice that the process $x_t'$ has the same distribution with $\mathbf{x}_t$, thus in the following analysis we study the convergence of the intermediate process $p_k' = (x_{k\eta}', v_{k\eta}')$. By taking the difference of equation (28) with (29) and the Girsanov formula, we can derive the Radon-Nikodym derivative of $\hat{P}_{K\eta}$ w.r.t $p_K'$:

$$\frac{d\hat{p}_{K\eta}}{dp_K'} = exp\left\{\sqrt{\frac{u}{2\gamma}}\int_0^T (\gamma\alpha_x(s) + \alpha_v(s) + \frac{1}{T}\alpha_x(T) + \nabla U(\mathbf{x}_s) - G_s)d\mathbf{B}s \right.$$

$$\left. -\frac{u}{4\gamma}\int_0^T \|\gamma\alpha_x(s) + \alpha_v(s) + \frac{1}{T}\alpha_x(T) + \nabla U(\mathbf{x}_s) - G_s\|^2 ds\right\}.$$

Thus the divergence can be bouned as:

$$D_{KL}(p_K\|\hat{p}_{K\eta}) = \mathbb{E}_{p_K}\left[\log\left(\frac{d\hat{p}_{K\eta}}{dp_K}\right)\right]$$

$$= \frac{u}{4\gamma}\int_0^T \mathbb{E}\left\|\gamma\alpha_x(s) + \alpha_v(s) + \frac{1}{T}\alpha_x(T) + \nabla U(\mathbf{x}_s) - G_s\right\|^2 ds$$

$$= \frac{u}{4\gamma T}\mathbb{E}\left[\|\alpha_x(T)\|^2\right] + \frac{u}{4\gamma}\sum_{k=0}^K \int_{k\eta}^{(k+1)\eta} \mathbb{E}\left[\|\gamma\alpha_v(s) + \alpha_x(s) + \nabla U(\mathbf{x}_s) - G_s\|^2\right]ds$$

$$\leq \frac{u}{4\gamma T\eta^2}\mathbb{E}\left[\|\alpha_k^{\mathbf{x}}\|^2\right] + \frac{u}{4\gamma}\sum_{k=0}^K \int_{k\eta}^{(k+1)\eta} \mathbb{E}\left[\|\gamma\alpha_v(s)\|^2\right]ds + \frac{u}{4\gamma}\sum_{k=0}^K \int_{k\eta}^{(k+1)\eta} \mathbb{E}\left[\|\alpha_x(s)\|^2\right]ds$$

$$+ \frac{u}{4\gamma}\sum_{k=0}^K \int_{k\eta}^{(k+1)\eta} \mathbb{E}\left[\|\nabla U(\mathbf{x}_s) - G_s\|^2\right]ds$$

$$\leq \frac{u}{4\gamma T\eta^2}\mathbb{E}\left[\|\alpha_k^{\mathbf{x}}\|^2\right] + \frac{u}{4\gamma}\sum_{k=0}^K \int_{k\eta}^{(k+1)\eta} \mathbb{E}\left[\|\gamma\alpha_k^{\mathbf{v}}/\eta\|^2\right]ds + \frac{u}{4\gamma}\sum_{k=0}^K \int_{k\eta}^{(k+1)\eta} \mathbb{E}\left[\|\alpha_k^{\mathbf{x}}/\eta\|^2\right]ds$$

$$+ \frac{u}{4\gamma}\sum_{k=0}^K \int_{k\eta}^{(k+1)\eta} \mathbb{E}\left[\|\nabla U(\mathbf{x}_s) - Q_G(\nabla U(\mathbf{x}_k))\|^2\right]ds$$

$$\leq \frac{u}{4\gamma T\eta^2}\mathbb{E}\left[\|\alpha_k^{\mathbf{x}}\|^2\right] + \frac{u}{4\gamma}\sum_{k=0}^K \int_{k\eta}^{(k+1)\eta} \mathbb{E}\left[\|\gamma\alpha_k^{\mathbf{v}}/\eta\|^2\right]ds + \frac{u}{4\gamma}\sum_{k=0}^K \int_{k\eta}^{(k+1)\eta} \mathbb{E}\left[\|\alpha_k^{\mathbf{x}}/\eta\|^2\right]ds \tag{30}$$

$$+ \frac{u}{4\gamma}\sum_{k=0}^K \int_{k\eta}^{(k+1)\eta} \mathbb{E}\left[\|\nabla U(\mathbf{x}_s) - \nabla U(\mathbf{x}_k)\|^2\right]ds + \frac{u}{4\gamma}\sum_{k=0}^K \int_{k\eta}^{(k+1)\eta} \mathbb{E}\left[\|\nabla U(\mathbf{x}_k) - Q_G(\nabla U(\mathbf{x}_k))\|^2\right]ds.$$

By assumption 1, we know that:

$$\mathbb{E}\left[\|\nabla U(\mathbf{x}_s) - \nabla U(\mathbf{x}_k)\|^2\right] \leq M^2\mathbb{E}\left[\|\mathbf{x}_s - \mathbf{x}_k\|^2\right].$$

From the same analysis in (25), we can derive:

$$\mathbb{E}\left[\|\nabla U(\mathbf{x}_s) - \nabla U(\mathbf{x}_k)\|^2\right] \leq 3M^2\eta^2\left(\mathbb{E}\left[\|\mathbf{v}_k'\|^2\right] + u^2\eta^2\mathbb{E}\left[\|Q_G(\nabla U(\mathbf{x}_k))\|^2\right] + 2du\right).$$

Now we need to derive a uniform bound of $\mathbb{E}\left[\|\mathbf{x}_k\|^2\right]$ and $\mathbb{E}\left[\|\mathbf{v}_k'\|^2\right]$.

**Lemma 11.** *Let Assumptions 2 and 1 hold. If we set the step size to the following condition*

$$\eta \leq \min \left\{ \frac{\gamma}{4\left(8Mu + u\gamma + 22\gamma^2\right)}, \sqrt{\frac{4u^2}{4Mu + 3\gamma^2}}, \frac{6\gamma bu}{\left(4Mu + 3\gamma^2\right)d}, \frac{\gamma m_2}{6\left(22u + \gamma\right)M^2} \right\},$$

*then for all* $k > 0$ $\mathbb{E}\left[\|\mathbf{x}_k\|^2\right]$ *and* $\mathbb{E}\left[\|v_k\|^2\right]$ *can be bouned as follow:*

$$\mathbb{E}\left[\|\mathbf{x}_k\|^2\right] \leq \mathcal{E} + C\Delta^2 d, \quad \mathbb{E}\left[\|v'_k\|^2\right] \leq \gamma^2 \mathcal{E}/2 + \gamma^2 C\Delta^2 d/2,$$

*where constants* $\mathcal{E}$ *and* $C$ *are defined as:*

$$\mathcal{E} = \mathbb{E}\left[\mathcal{E}(\mathbf{x}_0, \mathbf{v}_0)\right] + \frac{54\left(4u + \gamma^2\right)u}{m_2\gamma^4}\sigma^2 + \frac{12(22u + \gamma)uM^3}{m_2\gamma^3}G^2 + \frac{96\left(d + b\right)uM}{m_2\gamma^2}$$

$$C = \frac{27\left(4u + \gamma^2\right)u}{2m_2\gamma^4}.$$

The proof of Lemma 11 can be found in Appendix F.5. Thus,

$$\mathbb{E}\left[\|\nabla U(\mathbf{x}_s) - \nabla U(\mathbf{x}_k)\|^2\right] \leq 3M^2\eta^2 \left(\mathbb{E}\left[\|v_k\|^2\right] + u^2\eta^2 \left(\frac{\Delta^2 d}{4} + \sigma^2 + 2M^2\mathbb{E}\left[\|\mathbf{x}_k\|^2\right] + 2G^2\right) + 2du\right)$$

$$\leq 3M^2\eta^2 \left(\gamma^2 \mathcal{E}/2 + \gamma^2 C\Delta^2 d/2 + u^2\eta^2 \left(\frac{\Delta^2 d}{4} + \sigma^2 + 2M^2\mathcal{E} + 2M^2C\Delta^2 d + 2G^2\right) + 2du\right)$$

$$\leq 3M^2\eta^2 \left(\left(\gamma^2 + 2u^2M^2\right)\mathcal{E} + \left(\gamma^2 + 2u^2M^2\right)C\Delta^2 d + u^2\sigma^2 + 2u^2G^2 + 2du\right).$$

Now we can go back to the divergence of $p_K$ and $\hat{p}_{K\eta}$,

$$D_{KL}(p_K||\hat{p}_{K\eta})$$

$$\leq \frac{u}{4\gamma T\eta^2}\mathbb{E}\left[\|\alpha_k^{\mathbf{x}}\|^2\right] + \frac{u}{4\gamma}\sum_{k=0}^{K}\int_{k\eta}^{(k+1)\eta}\mathbb{E}\left[\|\gamma\alpha_k^{\mathbf{v}}/\eta\|^2\right]ds + \frac{u}{4\gamma}\sum_{k=0}^{K}\int_{k\eta}^{(k+1)\eta}\mathbb{E}\left[\|\alpha_k^{\mathbf{x}}/\eta\|^2\right]ds$$

$$+ \frac{u}{4\gamma}3M^2K\eta^3\left(\left(\gamma^2 + 2u^2M^2\right)\mathcal{E} + \left(\gamma^2 + 2u^2M^2\right)C\Delta^2 d + u^2\sigma^2 + 2u^2G^2 + 2du\right) + \frac{u}{4\gamma}K\eta\left(\frac{\Delta^2 d}{4} + \sigma^2\right)$$

$$\leq \frac{u}{4\gamma}3M^2K\eta^3\left(\left(\gamma^2 + 2u^2M^2\right)\mathcal{E} + \left(\gamma^2 + 2u^2M^2\right)C\Delta^2 d + u^2\sigma^2 + 2u^2G^2 + 2du\right) + \frac{u}{4\gamma}K\eta\left(\frac{\Delta^2 d}{4} + \sigma^2\right)$$

$$+ \frac{u\Delta^2 d}{16\gamma T\eta^2} + \frac{uK\Delta^2 d}{8\gamma\eta}$$

$$\leq \frac{u}{4\gamma}3M^2K\eta^3\left(\left(\gamma^2 + 2u^2M^2\right)\mathcal{E} + u^2\sigma^2 + 2u^2G^2 + 2du\right) + \frac{u}{4\gamma}K\eta\sigma^2$$

$$+ \left(\frac{u}{4\gamma}3M^2K\eta^3 C\left(\gamma^2 + 2u^2M^2\right) + \frac{uK\eta}{16\gamma} + \frac{u}{16\gamma T\eta^2} + \frac{uK}{8\gamma\eta}\right)\Delta^2 d$$

$$=: C_0 K\eta^3 + C_1 K\eta\sigma^2 + C_2 K\Delta^2,$$

where the constants $C_0$, $C_1$ and $C_2$ are defined as:

$$C_0 = \frac{u}{4\gamma}3M^2\left(\left(\gamma^2 + 2u^2M^2\right)\mathcal{E} + u^2\sigma^2 + 2u^2G^2 + 2du\right)$$

$$C_1 = \frac{u}{4\gamma}$$

$$C_2 = \left(\frac{u}{4\gamma}3M^2\eta^3 C\left(\gamma^2 + 2u^2M^2\right) + \frac{u}{16\gamma} + \frac{u}{16\gamma T^2\eta} + \frac{u}{8\gamma\eta}\right)d.$$

By the weighted CKP inequality and given $K\eta \geq 1$,

$$\mathcal{W}_2(p_K, \hat{p}_{K\eta}) \leq \overline{\Lambda} \left( \sqrt{D_{KL}(p_K \| \hat{p}_{K\eta})} + \sqrt[4]{D_{KL}(p_K \| \hat{p}_{K\eta})} \right)$$

$$\leq \left( \widetilde{C_0}\sqrt{\eta} + \widetilde{C_1}\widetilde{A} \right) \sqrt{K\eta} + \widetilde{C_2}\sqrt{K\Delta}, \tag{31}$$

where the constants are defined as:

$$\widetilde{C_0} = \left( \sqrt{C_0} + \sqrt[4]{C_0} \right)$$

$$\widetilde{C_1} = \left( \sqrt{C_1} + \sqrt[4]{C_1} \right)$$

$$\widetilde{C_2} = \left( \sqrt{C_2} + \sqrt[4]{C_2} \right)$$

$$\widetilde{A} = \max \left\{ \sigma, \sqrt{\sigma} \right\}.$$

From the same analysis in (26), we can have:

$$\mathcal{W}_2(p_K, p^*) \leq \overline{\Lambda} \left( \widetilde{C_0}\sqrt{\eta} + \widetilde{C_1}\widetilde{A} \right) \sqrt{K\eta} + \widetilde{C_2}\sqrt{K\eta} + \Gamma_0 e^{-\mu^* K\eta}. \tag{32}$$

In order to bound the Wasserstein distance, we need to set

$$\overline{\Lambda}\widetilde{C_0}\sqrt{K\eta^2} = \frac{\epsilon}{2} \quad \text{and} \quad \Gamma_0 e^{-\mu^* K\eta} = \frac{\epsilon}{2}. \tag{33}$$

Solving the equation (33), we can have

$$K\eta = \frac{\log\left(\frac{2\Gamma_0}{\epsilon}\right)}{\mu^*} \quad \text{and} \quad \eta = \frac{\epsilon^2}{4\overline{\Lambda}^2 \widetilde{C_0}^2 K\eta}.$$

Combining these two we can have

$$\eta = \frac{\epsilon^2 \mu^*}{4\overline{\Lambda}^2 \widetilde{C_0}^2 \log\left(\frac{2\Gamma_0}{\epsilon}\right)} \quad \text{and} \quad K = \frac{4\overline{\Lambda}^2 \widetilde{C_0}^2 \log^2\left(\frac{2\Gamma_0}{\epsilon}\right)}{\epsilon^2 (\mu^*)^2}.$$

Plugging in (32) completes the proof.

### E.3 Proof of Theorem 3

In this section, we analyze the convergence of VC SGHMCLP-L when the target distribution is non-log-concave. Similarly, the gradients are unbiased, additional noise applied to the state $\mathbf{x}$ causes deviation from the underdamped Langevin dynamics. This proof is similar to the proof of Theorem 2, however, the variance-corrected quantization function gives us a bound for the difference between the quantized value and the full-precision value. This bound can scale with the learning rate. This fact leads to the advantage of variance-corrected quantization over naive stochastic rounding.

Similarily, from the analysis in (57), we know that

$$\mathbb{E}\left[ \|\alpha_k^{\mathbf{v}}\|^2 \right] \leq \gamma \eta \mathcal{A}, \tag{34}$$

where $A = \max\left\{ \Delta\sqrt{d}\left(A' + \mathcal{G}\right), 4ud \right\}$. By the analysis in (55), we know that if $\text{Var}_{\mathbf{x}}^{hmc} \geq \frac{\Delta^2}{4}$, we can have

$$\mathbb{E}\left[ \|\alpha_k^{\mathbf{x}}\|^2 \right] \leq 4ud\eta^2 \tag{35}$$

by (58), if $\text{Var}_{\mathbf{x}}^{hmc} < \frac{\Delta^2}{4}$,

$$\mathbb{E}\left[ \|\alpha_k^{\mathbf{x}}\|^2 \right] \leq \eta B, \tag{36}$$

where $B = \max\left\{2\Delta\sqrt{d}A' + u\eta\sqrt{d}\mathcal{G}, 4ud\eta\right\}$. Thus, we can define the following:

$$\mathbb{E}\left[\|\alpha_k^{\mathsf{x}}\|^2\right] = \eta\mathcal{B}, \tag{37}$$

where $\mathcal{B}$ is defined as:

$$\mathcal{B} = \begin{cases} 4ud\eta, & \text{if } \mathrm{Var}_{\mathsf{x}}^{hmc} \geq \frac{\Delta^2}{4} \\ B, & \text{else.} \end{cases}$$

Combining the bound of $\mathbb{E}\left[\|\alpha_k^{\mathsf{x}}\|^2\right]$, $\mathbb{E}\left[\|\alpha_k^{\mathsf{v}}\|^2\right]$ with (30), we can show,

$D_{KL}(p_K\|\hat{p}_{K\eta})$

$\leq \dfrac{u}{4\gamma T\eta^2}\mathbb{E}\left[\|\alpha_k^{\mathsf{x}}\|^2\right] + \dfrac{u}{4\gamma}\sum_{k=0}^{K}\int_{k\eta}^{(k+1)\eta}\mathbb{E}\left[\|\gamma\alpha_k^{\mathsf{v}}/\eta\|^2\right]ds + \dfrac{u}{4\gamma}\sum_{k=0}^{K}\int_{k\eta}^{(k+1)\eta}\mathbb{E}\left[\|\alpha_k^{\mathsf{x}}/\eta\|^2\right]ds$

$+ \dfrac{u}{4\gamma}3M^2K\eta^3\left(\left(\gamma^2 + 2u^2M^2\right)\mathcal{E} + \left(\gamma^2 + 2u^2M^2\right)C\Delta^2 d + u^2\sigma^2 + 2u^2G^2 + 2du\right) + \dfrac{u}{4\gamma}K\eta\left(\dfrac{\Delta^2 d}{4} + \sigma^2\right)$

$\leq \dfrac{u}{4\gamma}3M^2K\eta^3\left(\left(\gamma^2 + 2u^2M^2\right)\mathcal{E} + \left(\gamma^2 + 2u^2M^2\right)C\Delta^2 d + u^2\sigma^2 + 2u^2G^2 + 2du\right) + \dfrac{u}{4\gamma}K\eta\left(\dfrac{\Delta^2 d}{4} + \sigma^2\right)$

$+ \dfrac{u\mathcal{B}}{4\gamma T} + \dfrac{uK\mathcal{A}}{4} + \dfrac{uK\mathcal{B}}{4\gamma}$

$\leq \dfrac{u}{4\gamma}3M^2K\eta^3\left(\left(\gamma^2 + 2u^2M^2\right)\mathcal{E} + \left(\gamma^2 + 2u^2M^2\right)C\Delta^2 d + u^2\sigma^2 + 2u^2G^2 + 2du\right) + \dfrac{u}{4\gamma}K\eta\left(\dfrac{\Delta^2 d}{4} + \sigma^2\right)$

$+ \dfrac{uK\mathcal{A}}{4} + \dfrac{uK\mathcal{B}}{2\gamma}$

$\leq \dfrac{u}{4\gamma}3M^2K\eta^3\left(\left(\gamma^2 + 2u^2M^2\right)\mathcal{E} + u^2\sigma^2 + 2u^2G^2 + 2du\right) + \dfrac{u}{4\gamma}K\eta\sigma^2 + \dfrac{u}{16\gamma}K\eta\Delta^2 d + \dfrac{uK\mathcal{A}}{4} + \dfrac{uK\mathcal{B}}{2\gamma}$

$=: C_0 K\eta^3 + C_1 K\eta\sigma^2 + C_2 K\eta\Delta^2 + C_3 K\mathcal{A} + C_4 K\mathcal{B},$

where the constants are defined as

$$C_0 = \dfrac{u}{4\gamma}3M^2\left(\left(\gamma^2 + 2u^2M^2\right)\mathcal{E} + u^2\sigma^2 + 2u^2G^2 + 2du\right)$$
$$C_1 = \dfrac{u}{4\gamma}$$
$$C_2 = \dfrac{u}{16\gamma}d$$
$$C_3 = \dfrac{u}{4}$$
$$C_4 = \dfrac{u}{2\gamma}.$$

By the weighted CKP inequality and given $K\eta \geq 1$,

$$\mathcal{W}_2(p_K, \hat{p}_{K\eta}) \leq \overline{\Lambda}\left(\sqrt{D_{KL}(p_K\|\hat{p}_{K\eta})} + \sqrt[4]{D_{KL}(p_K\|\hat{p}_{K\eta})}\right)$$
$$\leq \left(\widetilde{C}_0\sqrt{\eta} + \widetilde{C}_1\widetilde{A} + \widetilde{C}_2\sqrt{\Delta}\right)\sqrt{K\eta} + \widetilde{C}_3\sqrt{K\mathcal{A}} + \widetilde{C}_4\sqrt{K\mathcal{B}},$$

where the constants are defined as:

$$\widetilde{C_0} = \overline{\Lambda}\left(\sqrt{C_0} + \sqrt[4]{C_0}\right)$$

$$\widetilde{C_1} = \overline{\Lambda}\left(\sqrt{C_1} + \sqrt[4]{C_1}\right)$$

$$\widetilde{C_2} = \overline{\Lambda}\left(\sqrt{C_2} + \sqrt[4]{C_2}\right)$$

$$\widetilde{C_3} = \overline{\Lambda}\left(\sqrt{C_3} + \sqrt[4]{C_3}\right)$$

$$\widetilde{C_4} = \overline{\Lambda}\left(\sqrt{C_4} + \sqrt[4]{C_4}\right)$$

$$\widetilde{A}^2 = \overline{\Lambda}\max\left\{\sigma^2, \sqrt{\sigma^2}\right\}.$$

From the same analysis of (26), we can have:

$$\mathcal{W}_2(p_K, p^*) \leq \left(\widetilde{C_0}\sqrt{\eta} + \widetilde{C_1}\widetilde{A}\right)\sqrt{K\eta} + \widetilde{C_2}\sqrt{K\eta}\Delta + \widetilde{C_3}\sqrt{K\mathcal{A}} + \widetilde{C_4}\sqrt{K\mathcal{B}} + \Gamma_0 e^{-\mu^* K\eta}. \tag{38}$$

To bound the Wasserstein distance, we need to set

$$\overline{\Lambda}\widetilde{C_0}\sqrt{K\eta^2} = \frac{\epsilon}{2} \quad \text{and} \quad \Gamma_0 e^{-\mu^* K\eta} = \frac{\epsilon}{2}. \tag{39}$$

Solving the equation (39), we can have

$$K\eta = \frac{\log\left(\frac{2\Gamma_0}{\epsilon}\right)}{\mu^*} \quad \text{and} \quad \eta = \frac{\epsilon^2}{4\overline{\Lambda}^2\widetilde{C_0}^2 K\eta}.$$

Combining these two we can have

$$\eta = \frac{\epsilon^2\mu^*}{4\overline{\Lambda}^2\widetilde{C_0}^2\log\left(\frac{2\Gamma_0}{\epsilon}\right)} \quad \text{and} \quad K = \frac{4\overline{\Lambda}^2\widetilde{C_0}^2\log^2\left(\frac{2\Gamma_0}{\epsilon}\right)}{\epsilon^2\left(\mu^*\right)^2}.$$

Plugging in (38) completes the proof.

### E.4 Proof of Theorem 4

In this section, we analyze the convergence of full-precision gradient accumulators (SGHMCLP-F) introduced in Section 3.1 when the target distribution is strongly log-concave. Again SGHMCLP-F uses biased gradients because we quantize the parameter before taking the gradients. We need to derive the upper bound given biased gradients.

The SGHMCLP-F update follows

$$\mathbf{vv}_{k+1} = \mathbf{v}_k e^{-\gamma\eta} - u\gamma^{-1}(1 - e^{-\gamma\eta})Q_G(\widetilde{\nabla U}(Q_W(\mathbf{x}_k))) + \xi_k^{\mathbf{v}}$$

$$\mathbf{vx}_{k+1} = \mathbf{x}_k + \gamma^{-1}(1 - e^{-\gamma\eta})\mathbf{v}_k + u\gamma^{-2}(\gamma\eta + e^{-\gamma\eta} - 1)Q_G(\widetilde{\nabla U}(Q_W(\mathbf{x}_k))) + \xi_k^{\mathbf{x}},$$

In this section, we prove the convergence of SGHMCLP-F in terms of 2-Wasserstein distance for strongly-log-concave target distribution via coupling argument. To simplify the notation we define the quantized stochastic gradients at $\mathbf{x}$ as:

$$\tilde{g}(\mathbf{x}) := Q_G(\widetilde{\nabla U}(Q_W(\mathbf{x})))$$

$$=: \nabla U(\mathbf{x}) + \xi. \tag{40}$$

**Lemma 12.** *For any $\mathbf{x} \in \mathbb{R}^d$, the random noise $\xi$ of the low-precision gradients defined in* (40) *satisfies:*

$$\|\mathbb{E}\xi\|^2 \leq M^2 \frac{\Delta^2 d}{4}$$

$$\mathbb{E}[\|\xi\|^2] \leq (M^2 + 1)\frac{\Delta^2 d}{4} + \sigma^2.$$

The proof of Lemma 12 can be found in Appendix F.1. We follow the proof in Cheng et al. (2018). Denote by $\mathcal{B}(\mathbb{R}^d)$ the Borel $\sigma$-field of $\mathbb{R}^d$. Given probability measures $\mu$ and $\nu$ on $(\mathbb{R}^d, \mathcal{B}(\mathbb{R}^d))$, we define a *transference plan* $\zeta$ between $\mu$ and $\nu$ as a probability measure on $(\mathbb{R}^d \times \mathbb{R}^d, \mathcal{B}(\mathbb{R}^d \times \mathbb{R}^d))$ such that for all sets $A \in \mathbb{R}^d$, $\zeta(A \times \mathbb{R}^d) = \mu(A)$ and $\zeta(\mathbb{R}^d \times A) = \nu(A)$. We denote $\Gamma(\mu, \nu)$ as the set of all transference plans. A pair of random variables $(\mathbf{x}, \mathbf{y})$ is called a coupling if there exists a $\zeta \in \Gamma(\mu, \nu)$ such that $(\mathbf{x}, \mathbf{y})$ is distributed according to $\zeta$. (With some abuse of notation, we will also refer to $\zeta$ as the coupling.)

To calculate the Wasserstein distance from the proposed sample $(\mathbf{x}_K, \mathbf{v}_K)$ and the target distribution sample $(\mathbf{x}^*, \mathbf{v}^*)$, we define sample $q_k = (\mathbf{x}_k, \mathbf{x}_k + \mathbf{v}_k)$ and the target distribution sample $q^* = (\mathbf{x}^*, \mathbf{x}^* + \mathbf{v}^*)$. Let $p_k = (\mathbf{x}_k, \mathbf{v}_k)$ and $\widehat{\Phi}_\eta$ be the operator that maps from $p_k$ to $p_{k+1}$ i.e.

$$p_{k+1} = \widehat{\Phi}_\eta p_k.$$

The solution $(\mathbf{x}_t, \mathbf{v}_t)$ of the continuous underdamped Langevin dynamics with exact gradient satisfies the following equations:

$$\mathbf{v}_t = \mathbf{v}_0 e^{-\gamma t} - u\left(\int_0^t e^{-\gamma(t-s)}\nabla U(\mathbf{x}_s)ds\right) + \sqrt{2\gamma u}\int_0^t e^{-\gamma(t-s)}dB_s, \tag{41}$$

$$\mathbf{x}_t = \mathbf{x}_0 + \int_0^t \tilde{\mathbf{v}}_s ds.$$

Let $\Phi_\eta$ denote the operator that maps $p_0$ to the solution of continuous underdamped Langevin dynamics in (41) after time step $\eta$. Notice the solution $(\tilde{\mathbf{v}}_t, \tilde{\mathbf{x}}_t)$ of the discrete underdamped Langevin dynamics as in (10) with an exact gradient can be written as

$$\tilde{\mathbf{v}}_t = \tilde{\mathbf{v}}_0 e^{-\gamma t} - u\left(\int_0^t e^{-\gamma(t-s)}\nabla U(\tilde{\mathbf{x}}_0)ds\right) + \sqrt{2\gamma u}\int_0^t e^{-\gamma(t-s)}dB_s, \tag{42}$$

$$\tilde{\mathbf{x}}_t = \tilde{\mathbf{x}}_0 + \int_0^t \tilde{\mathbf{v}}_s ds.$$

We can also define a similar operator for the discrete underdamped Langevin dynamics solution $\tilde{p}_t = (\tilde{\mathbf{x}}_t, \tilde{\mathbf{v}}_t)$, let $\widetilde{\Phi}_t$ be the operator that maps $\tilde{p}_0$ to $\tilde{p}_t$. Furthermore the SGHMCLP-F can be written as:

$$\mathbf{v}_t = \mathbf{v}_0 e^{-\gamma t} - u\left(\int_0^t e^{-\gamma(t-s)}\tilde{g}(\mathbf{x}_0)ds\right) + \sqrt{2\gamma u}\int_0^t e^{-\gamma(t-s)}dB_s, \tag{43}$$

$$\mathbf{x}_t = \tilde{\mathbf{x}}_0 + \int_0^t \mathbf{v}_s ds.$$

Given $\tilde{g}(\mathbf{x}_0) = \nabla U(\mathbf{x}_0) + \xi_0$ and $\mathbf{x}_0 = \tilde{\mathbf{x}}_0$, we know:

$$\mathbf{v}_t = \tilde{\mathbf{v}}_t - u\left(\int_0^t e^{-\gamma(t-s)}ds\right)\xi \tag{44}$$

$$\mathbf{x}_t = \tilde{\mathbf{x}}_t - u\left(\int_0^t \left(\int_0^r e^{-\gamma(t-s)}ds\right)dr\right)\xi.$$

**Lemma 13.** *Let $q_0$ be some initial distribution and $\widetilde{\Phi}_\eta$ and $\Phi_\eta$ be the operator we defined above for discrete Langevin dynamics with exact full-precision gradients and low-precision gradients respectively. If the stepszie $1 > \eta > 0$, then the Wasserstein distance satisfies*

$$\mathcal{W}_2^2(\Phi_\eta q_0, q^*) \leq \left(\mathcal{W}_2(\widetilde{\Phi}_\eta q_0, q^*) + \sqrt{5}/2u\eta\sqrt{d}M\Delta\right)^2 + 5u^2\eta^2\left((M^2+1)\frac{\Delta^2 d}{4} + \sigma^2\right).$$

The proof of Lemma 13 can be found in Appendix F.2. The lemma 13 says that if starting from the same distribution after one step of low-precision update the Wasserstein distance from the target distribution is bounded by the distance after one step of exact gradients plus $\mathcal{O}(\eta^2 \Delta^2)$. Furthermore from the corollary 7 in Cheng et al. (2018) we know that for any $i \in \{1, \cdots, K\}$:

$$\mathcal{W}_2^2(\Phi_\eta q_i, q^*) \le e^{-\eta/2\kappa_1} \mathcal{W}_2^2(q_i, q^*), \tag{45}$$

where $\kappa_1 = M/m_1$ is the condtion number. Let $\mathcal{E}_K$ denote the $26 \left( d/m_1 + \mathcal{D}^2 \right)$, and from the discretization error bound from Theorem 9 and Lemma 8 (sandwich inequality) in Cheng et al. (2018), we get

$$\mathcal{W}_2(\Phi_\eta q_i, \widetilde{\Phi}_\eta q_i) \le 2\mathcal{W}_2(\Phi_\eta p_i, \widetilde{\Phi}_\eta p_i) \le \eta^2 \sqrt{\frac{8\mathcal{E}_K}{5}}.$$

By triangle inequality:

$$\mathcal{W}_2(\widetilde{\Phi}_\eta q_i, q^*) \le \mathcal{W}_2(\Phi_\eta q_i, \widetilde{\Phi}_\eta q_i) + \mathcal{W}_2(\Phi_\eta q_i, q^*)$$

$$\le \eta^2 \sqrt{\frac{8\mathcal{E}_K}{5}} + e^{-\eta/2\kappa_1} \mathcal{W}_2(q_i, q^*).$$

Combine this with the result in Lemma 13 we have,

$$\mathcal{W}_2^2(\widehat{\Phi}_\eta q_i, q^*) \le \left( e^{-\eta/2\kappa_1} \mathcal{W}_2(q_i, q^*) + \eta^2 \sqrt{\frac{8\mathcal{E}_K}{5}} + \sqrt{5}/2u\eta\sqrt{d}M\Delta \right)^2 + 5u^2\eta^2 \left( (M^2+1)\frac{\Delta^2 d}{4} + \sigma^2 \right). \tag{46}$$

By invoking the Lemma 7 in Dalalyan & Karagulyan (2019) we can bound the 2-Wasserstein distance by:

$$\mathcal{W}_2(q_K, q^*) \le e^{-K\eta/2\kappa_1} \mathcal{W}_2(q_0, q^*) + \frac{\eta^2 \sqrt{\frac{8\mathcal{E}_K}{5}} + \frac{u\eta M\Delta\sqrt{5d}}{2}}{1 - e^{-\eta/2\kappa_1}}$$

$$+ \frac{5u^2\eta^2 \left( (M^2+1)\frac{\Delta^2 d}{4} + \sigma^2 \right)}{\eta^2 \sqrt{\frac{8\mathcal{E}_K}{5}} + \frac{u\eta M\Delta\sqrt{5d}}{2} + \sqrt{1 - e^{-\eta/\kappa_1}}\sqrt{5u^2\eta^2 \left( (M^2+1)\frac{\Delta^2 d}{4} + \sigma^2 \right)}}.$$

Finally, by sandwich inequality we have:

$$\mathcal{W}_2(p_K, p^*) \le 4e^{-K\eta/2\kappa} \mathcal{W}_2(p_0, p^*) + 4\frac{\eta^2 \sqrt{\frac{8\mathcal{E}_K}{5}} + \frac{u\eta M\Delta\sqrt{5d}}{2}}{1 - e^{-\eta/2\kappa}}$$

$$+ \frac{20u^2\eta^2 \left( (M^2+1)\frac{\Delta^2 d}{4} + \sigma^2 \right)}{\eta^2 \sqrt{\frac{8\mathcal{E}_K}{5}} + \frac{u\eta M\Delta\sqrt{5d}}{2} + \sqrt{1 - e^{-\eta/\kappa}}\sqrt{5u^2\eta^2 \left( (M^2+1)\frac{\Delta^2 d}{4} + \sigma^2 \right)}}.$$

Now we let the first term less than $\epsilon/3$, from the lemma 13 in (Cheng et al., 2018) we know that $\mathcal{W}_2(p_K, p^*) \le 3 \left( \frac{d}{m_1} + \mathcal{D}^2 \right)$. So we can choose $K$ as the following,

$$K \le \frac{2\kappa_1}{\eta} \log \left( 36 \left( \frac{d}{m_1} + \mathcal{D}^2 \right) \right).$$

Next, we choose a step size $\eta \le \frac{\epsilon \kappa_1^{-1}}{\sqrt{479232/5(d/m_1 + \mathcal{D}^2)}}$ to ensure the second term is controlled below $\epsilon/3 + \frac{16\kappa_1 uM\Delta\sqrt{5d}}{2}$. Since $1 - e^{-\eta/2\kappa_1} \ge \eta/4\kappa_1$ and definition of $\mathcal{E}_K$,

$$4\frac{\eta^2 \sqrt{\frac{8\mathcal{E}_K}{5}} + \frac{u\eta M\Delta\sqrt{5d}}{2}}{1 - e^{-\eta/2\kappa}} \le 4\frac{\eta^2 \sqrt{\frac{8\mathcal{E}_K}{5}} + \frac{u\eta M\Delta\sqrt{5d}}{2}}{\eta/4\kappa_1} \le 16\kappa_1 \left( \eta\sqrt{\frac{8\mathcal{E}_K}{5}} + \frac{uM\Delta\sqrt{5d}}{2} \right)$$

$$\le \epsilon/3 + \frac{16\kappa_1 uM\Delta\sqrt{5d}}{2}.$$

Finally by choosing the step size satisfied that,

$$\eta \le \frac{\epsilon M \Delta \sqrt{5d}}{120u \left[ (M^2 + 1)\frac{\Delta^2 d}{4} + \sigma^2 \right]},$$

the third term can be bounded as:

$$\frac{20u^2\eta^2 \left( (M^2 + 1)\frac{\Delta^2 d}{4} + \sigma^2 \right)}{\eta^2 \sqrt{\frac{8\mathcal{E}_K}{5}} + \frac{u\eta M \Delta \sqrt{5d}}{2} + \sqrt{1 - e^{-\eta/\kappa}} \sqrt{5u^2\eta^2 \left( (M^2 + 1)\frac{\Delta^2 d}{4} + \sigma^2 \right)}}$$

$$\le \frac{20u^2\eta^2 \left( (M^2 + 1)\frac{\Delta^2 d}{4} + \sigma^2 \right)}{\frac{u\eta M \Delta \sqrt{5d}}{2}} = 40u\eta \frac{\left( (M^2 + 1)\frac{\Delta^2 d}{4} + \sigma^2 \right)}{M \Delta \sqrt{5d}} \le \epsilon/3.$$

This complete the proof.

### E.5 Proof of Theorem 5

In this section, we analyze the convergence of SGHMCLP-L when the target distribution is strongly log-concave. We mainly follow the proof in Cheng et al. (2018), the difference is we need to handle the noise. Recall the SGHMCLP-L update rule:

$$\mathbf{v}_{k+1} = Q_W \left( \mathbf{vv}_k e^{-\gamma\eta} - u\gamma^{-1}(1 - e^{\gamma\eta})Q_G(\widetilde{\nabla U}(\mathbf{x}_k)) + \xi_k^{\mathbf{v}} \right)$$

$$\mathbf{x}_{k+1} = Q_W \left( \mathbf{x}_k + \gamma^{-1}(1 - e^{-\gamma\eta})\mathbf{v}_k + u\gamma^{-2}(\gamma\eta + e^{-\gamma\eta} - 1)Q_G(\widetilde{\nabla U}(\mathbf{x}_k)) + \xi_k^{\mathbf{x}} \right).$$

If we let $\alpha_k^{\mathbf{x}}$ and $\alpha_k^{\mathbf{v}}$ denote the quantization error,

$$\alpha_k^{\mathbf{x}} = Q_W \left( \mathbf{v}_k e^{-\gamma\eta} - u\gamma^{-1}(1 - e^{\gamma\eta})Q_G(\widetilde{\nabla U}(\mathbf{x}_s)) + \xi_k^{\mathbf{v}} \right) - \left( \mathbf{v}_k e^{-\gamma\eta} - u\gamma^{-1}(1 - e^{\gamma\eta})Q_G(\widetilde{\nabla U}(\mathbf{x}_s)) + \xi_k^{\mathbf{v}} \right)$$

$$\alpha_k^{\mathbf{v}} = Q_W \left( \mathbf{x}_s + \gamma^{-1}(1 - e^{-\gamma\eta})v_k + u\gamma^{-2}(\gamma\eta + e^{-\gamma\eta} - 1)Q_G(\widetilde{\nabla U}(\mathbf{x}_s)) + \xi_k^{\mathbf{x}} \right)$$

$$- \left( \mathbf{x}_s + \gamma^{-1}(1 - e^{-\gamma\eta})v_k + u\gamma^{-2}(\gamma\eta + e^{-\gamma\eta} - 1)Q_G(\widetilde{\nabla U}(\mathbf{x}_s)) + \xi_k^{\mathbf{x}} \right),$$

we can rewrite the update rule as:

$$\mathbf{v}_{k+1} = \mathbf{v}_k e^{-\gamma\eta} - u\gamma^{-1}(1 - e^{\gamma\eta})Q_G(\widetilde{\nabla U}(\mathbf{x}_s)) + \xi_k^{\mathbf{v}} + \alpha_k^{\mathbf{v}}$$

$$\mathbf{x}_{k+1} = \mathbf{x}_k + \gamma^{-1}(1 - e^{-\gamma\eta})\mathbf{v}_k + u\gamma^{-2}(\gamma\eta + e^{-\gamma\eta} - 1)Q_G(\widetilde{\nabla U}(\mathbf{x}_k)) + \xi_k^{\mathbf{x}} + \alpha_k^{\mathbf{x}}. \tag{47}$$

Similarly, we can define a continuous interpolation of (47) for $t \in (0, \eta]$.

$$\mathbf{v}_t = \mathbf{v}_0 e^{-\gamma t} - u \left( \int_0^t e^{-\gamma(t-s)} \left( \nabla U(\mathbf{x}_0) + \zeta \right) ds \right) + \sqrt{2\gamma u} \int_0^t e^{-\gamma(t-s)} dB_s + \int_0^t \alpha_v(s) ds$$

$$\mathbf{x}_t = \mathbf{x}_0 + \int_0^t \mathbf{v}_s ds + \int_0^t \alpha_x(s) ds, \tag{48}$$

where the $\zeta = Q_G \left( \widetilde{\nabla U}(\hat{x}_0) \right) - \widetilde{\nabla U}(\hat{x}_0)$ the function $\alpha_v(s)$, $\alpha_x(s)$ are defined as:

$$\alpha_v(s) = \sum_{k=0}^{\infty} \alpha_k^{\mathbf{v}}/\eta \mathbf{1}_{s \in (k\eta, (k+1)\eta)}$$

$$\alpha_x(s) = \sum_{k=0}^{\infty} \alpha_k^{\mathbf{x}}/\eta \mathbf{1}_{s \in (k\eta, (k+1)\eta)}.$$

If we let $\hat{p}_0 = (\hat{x}_0, \hat{v}_0)$ be the initial sample and $\hat{p}_t = (\hat{x}_t, \hat{v}_t)$ be the sample that satisfies the previous equations, we can define an operator $\hat{\Phi}_t$ that maps $\hat{p}_0$ to $\hat{p}_t$ i.e., $\hat{p}_t = \hat{\Phi}_t \hat{p}_0$. Notice that since $\hat{p}_t$ is the continuous interpolation of (6), thus $\hat{p}_{k\eta} = p_k = (\mathbf{x}_k, v_k)$. Similarly, we define $q_k = (\mathbf{x}_k, v_k + \mathbf{x}_k) =: (\mathbf{x}_k, \omega_k)$ as a tool to analyze the convergence of $p_k$.

We are now ready to compute the Wasserstein distance between $\hat{\Phi}_\eta q_0$ and $q^*$. Let $\Gamma_1$ be all of the couplings between $\widetilde{\Phi}_\eta q_0$ and $q^*$, and $\Gamma_2$ be all of the couplings between $\widehat{\Phi}_\eta q_0$ and $q^*$. Let $r_1$ be the optimal coupling between $\widetilde{\Phi}_\eta q_0$ and $q^*$. By taking the difference between (48) and (42),

$$\begin{bmatrix} x \\ \omega \end{bmatrix} = \begin{bmatrix} \widetilde{x} \\ \widetilde{\omega} \end{bmatrix} + u \begin{bmatrix} \left(\int_0^\eta \left(\int_0^r e^{-\gamma(s-r)}ds\right) dr\right) \zeta + \int_0^\eta \alpha_x(s)ds \\ \left(\int_0^\eta \left(\int_0^r e^{-\gamma(s-r)}ds\right) dr + \int_0^\eta e^{-\gamma(s-\eta)}ds\right) \zeta + \int_0^\eta \alpha_x(s) + \alpha_v(s)ds \end{bmatrix}.$$

Let us now analyze the Wasserstein distance between $\hat{\Phi}_\eta q_0$ and $q^*$,

$$\mathcal{W}_2^2 \left(\hat{\Phi}_\eta q_0, q^*\right) \tag{49}$$

$$\leq \mathbb{E}_{r_1} \left\| \begin{bmatrix} \widetilde{x} \\ \widetilde{\omega} \end{bmatrix} + u \begin{bmatrix} \left(\int_0^\eta \left(\int_0^r e^{-\gamma(s-r)}ds\right) dr\right) \zeta + \int_0^\eta \alpha_x(s)ds \\ \left(\int_0^\eta \left(\int_0^r e^{-\gamma(s-r)}ds\right) dr + \int_0^\eta e^{-\gamma(s-\eta)}ds\right) \zeta + \int_0^\eta (\alpha_x(s) + \alpha_v(s)) ds \end{bmatrix} - \begin{bmatrix} x^* \\ \omega^* \end{bmatrix} \right\|^2$$

$$\leq \mathbb{E}_{r_1} \left\| \begin{bmatrix} \widetilde{x} \\ \widetilde{\omega} \end{bmatrix} - \begin{bmatrix} x^* \\ \omega^* \end{bmatrix} \right\|^2 + u^2 \mathbb{E} \left\| \begin{bmatrix} \left(\int_0^\eta \left(\int_0^r e^{-\gamma(s-r)}ds\right) dr\right) \zeta + \int_0^\eta \alpha_x(s)ds \\ \left(\int_0^\eta \left(\int_0^r e^{-\gamma(s-r)}ds\right) dr + \int_0^\eta e^{-\gamma(s-\eta)}ds\right) \zeta + \int_0^\eta (\alpha_x(s) + \alpha_v(s)) ds \end{bmatrix} \right\|^2$$

$$\leq \mathcal{W}_2^2 \left(\widetilde{\Phi}_\eta q_0, q^*\right) + 4u^2 \left(\left(\int_0^\delta \left(\int_0^r e^{-\gamma(s-r)}ds\right) dr\right)^2 + \left(\int_0^\delta e^{-\gamma(s-\delta)}ds\right)^2\right) \left(\frac{\Delta^2 d}{4} + \sigma^2\right)$$

$$+ u^2 \mathbb{E} \left[\left\|\int_0^\eta (\alpha_x(s)) ds\right\|^2\right] + u^2 \mathbb{E} \left[\left\|\int_0^\eta (\alpha_x(s) + \alpha_v(s)) ds\right\|^2\right]$$

$$\leq \mathcal{W}_2^2 \left(\widetilde{\Phi}_\eta q_0, q^*\right) + 4u^2 \left(\frac{\eta^4}{4} + \eta^2\right) \left(\frac{\Delta^2 d}{4} + \sigma^2\right) + u^2 \mathbb{E} \left[\|\alpha_k^{\mathbf{x}}\|^2\right] + u^2 \mathbb{E} \left[\|\alpha_k^{\mathbf{x}} + \alpha_k^{\mathbf{v}}\|^2\right]$$

$$\leq \mathcal{W}_2^2 \left(\widetilde{\Phi}_\eta q_0, q^*\right) + 5u^2\eta^2 \left(\frac{\Delta^2 d}{4} + \sigma^2\right) + 2u^2 \left(\mathbb{E} \|\alpha_k^{\mathbf{x}}\|^2 + \mathbb{E} \|\alpha_k^{\mathbf{v}}\|^2\right)$$

$$\leq \mathcal{W}_2^2 \left(\widetilde{\Phi}_\eta q_0, q^*\right) + 5u^2\eta^2 \left(\frac{\Delta^2 d}{4} + \sigma^2\right) + 2u^2 (A + B), \tag{50}$$

where the constant $A$, $B$ are the uniform bounds of $\mathbb{E}[\|\alpha_k^{\mathbf{x}}\|]$ and $\mathbb{E}[\|\alpha_k^{\mathbf{v}}\|]$ respectively. Furthermore from the corollary 7 in Cheng et al. (2018) we know that for any $i \in \{1, \cdots, K\}$:

$$\mathcal{W}_2^2(\Phi_\eta q_i, q^*) \leq e^{-\eta/2\kappa_1} \mathcal{W}_2^2(q_i, q^*), \tag{51}$$

where $\kappa_1 = M/m_1$ is the condition number. From the discretization error bound from theorem 9 and lemma 8(sandwich inequality) in Cheng et al. (2018), we get

$$\mathcal{W}_2(\Phi_\eta q_i, \widetilde{\Phi}_\eta q_i) \leq 2\mathcal{W}_2(\Phi_\eta p_i, \widetilde{\Phi}_\eta p_i) \leq \eta^2 \sqrt{\frac{8\mathcal{E}_K}{5}}.$$

By triangle inequality:

$$\mathcal{W}_2(\widetilde{\Phi}_\eta q_i, q^*) \leq \mathcal{W}_2(\Phi_\eta q_i, \widetilde{\Phi}_\eta q_i) + \mathcal{W}_2(\Phi_\eta q_i, q^*)$$

$$\leq \eta^2 \sqrt{\frac{8\mathcal{E}_K}{5}} + e^{-\eta/2\kappa_1} \mathcal{W}_2(q_i, q^*),$$

further implies the following inequality:

$$\mathcal{W}_2^2 \left(\hat{\Phi}_\eta q_i, q^*\right) \leq \left(e^{-\eta/2\kappa_1} \mathcal{W}_2(q_i, q^*) + \eta^2 \sqrt{\frac{8\mathcal{E}_K}{5}}\right)^2 + 5u^2\eta^2 \left(\frac{\Delta^2 d}{4} + \sigma^2\right) + 2u^2 (A + B). \tag{52}$$

By invoking the Lemma 7 in Dalalyan & Karagulyan (2019) we can bound the Wasserstein distance by:

$$\mathcal{W}_2(q_K, q^*) \leq e^{-K\eta/2\kappa_1} \mathcal{W}_2(q_0, q^*) + \frac{\eta^2 \sqrt{\frac{8\mathcal{E}_K}{5}}}{1 - e^{-\eta/2\kappa_1}}$$
$$+ \frac{5u^2\eta^2 \left(\frac{\Delta^2 d}{4} + \sigma^2\right) + 2u^2 (A + B)}{\eta^2 \sqrt{\frac{8\mathcal{E}_K}{5}} + \sqrt{1 - e^{-\eta/2\kappa_1}} \sqrt{5u^2\eta^2 \left(\frac{\Delta^2 d}{4} + \sigma^2\right) + 2u^2 (A + B)}}.$$

Finally, by sandwich inequality we have:

$$\mathcal{W}_2(p_K, p^*) \leq 4e^{-K\eta/2\kappa_1} \mathcal{W}_2(q_0, q^*) + \frac{4\eta^2 \sqrt{\frac{8\mathcal{E}_K}{5}}}{1 - e^{-\eta/2\kappa_1}} \tag{53}$$
$$+ \frac{20u^2\eta^2 \left(\frac{\Delta^2 d}{4} + \sigma^2\right) + 8u^2 (A + B)}{\eta^2 \sqrt{\frac{8\mathcal{E}_K}{5}} + \sqrt{1 - e^{-\eta/2\kappa_1}} \sqrt{5u^2\eta^2 \left(\frac{\Delta^2 d}{4} + \sigma^2\right) + 2u^2 (A + B)}}.$$

And in this case, we know that $\mathbb{E}\left[\|\alpha_k^{\mathsf{x}}\|\right]$ and $\mathbb{E}\left[\|\alpha_k^{\mathsf{y}}\|\right]$ can be bouned by $\frac{\Delta^2 d}{4}$. Finally, we can have:

$$\mathcal{W}_2(p_K, p^*) \leq 4e^{-K\eta/2\kappa_1} \mathcal{W}_2(q_0, q^*) + \frac{4\eta^2 \sqrt{\frac{8\mathcal{E}_K}{5}}}{1 - e^{-\eta/2\kappa_1}}$$
$$+ \frac{20u^2\eta^2 \left(\frac{\Delta^2 d}{4} + \sigma^2\right) + 4u^2\Delta^2 d}{\eta^2 \sqrt{\frac{8\mathcal{E}_K}{5}} + \sqrt{1 - e^{-\eta/2\kappa_1}} \sqrt{5u^2\eta^2 \left(\frac{\Delta^2 d}{4} + \sigma^2\right) + u^2\Delta^2 d}}.$$

Now we let the first term less than $\epsilon/3$, from the lemma 13 in (Cheng et al., 2018) we know that $\mathcal{W}_2(q_0, q^*) \leq 3\left(\frac{d}{m_1} + \mathcal{D}^2\right)$. So we can choose $K$ as the following,

$$K \leq \frac{2\kappa_1}{\eta} \log\left(36\left(\frac{d}{m_1} + \mathcal{D}^2\right)\right).$$

Next, we choose a step size $\eta \leq \frac{\epsilon\kappa_1^{-1}}{\sqrt{479232/5(d/m_1 + \mathcal{D}^2)}}$ to ensure the second term is controlled below $\epsilon/3$. Since $1 - e^{-\eta/2\kappa_1} \geq \eta/4\kappa_1$ and definition of $\mathcal{E}_K$,

$$4\frac{\eta^2 \sqrt{\frac{8\mathcal{E}_K}{5}}}{1 - e^{-\eta/2\kappa}} \leq 4\frac{\eta^2 \sqrt{\frac{8\mathcal{E}_K}{5}}}{\eta/4\kappa_1} \leq 16\kappa_1 \left(\eta\sqrt{\frac{8\mathcal{E}_K}{5}}\right) \leq \epsilon/3.$$

Finally by choosing the step size satisfied that,

$$\eta \leq \frac{\epsilon^2}{2880\kappa_1 u \left(\frac{\Delta^2 d}{4} + \sigma^2\right)},$$

the third term can be bounded as:

$$\frac{20u^2\eta^2\left((M^2+1)\frac{\Delta^2 d}{4}+\sigma^2\right)+4u^2\Delta^2 d}{\eta^2\sqrt{\frac{8\mathcal{E}_K}{5}}+\sqrt{1-e^{-\eta/2\kappa_1}}\sqrt{5u^2\eta^2\left((M^2+1)\frac{\Delta^2 d}{4}+\sigma^2\right)}}$$

$$\leq \frac{20u^2\eta^2\left((M^2+1)\frac{\Delta^2 d}{4}+\sigma^2\right)+4u^2\Delta^2 d}{\sqrt{1-e^{-\eta/2\kappa_1}}\sqrt{5u^2\eta^2\left((M^2+1)\frac{\Delta^2 d}{4}+\sigma^2\right)}} \leq \frac{20u^2\eta^2\left((M^2+1)\frac{\Delta^2 d}{4}+\sigma^2\right)+4u^2\Delta^2 d}{\sqrt{\eta/4\kappa_1}\sqrt{5u^2\eta^2\left((M^2+1)\frac{\Delta^2 d}{4}+\sigma^2\right)}}$$

$$\leq 4\sqrt{20\kappa_1 u^2\eta\left((M^2+1)\frac{\Delta^2 d}{4}+\sigma^2\right)}+\frac{8u^2\Delta^2 d\sqrt{\kappa_1}}{\eta^{3/2}\sqrt{5u^2\eta^2\left((M^2+1)\frac{\Delta^2 d}{4}+\sigma^2\right)}}$$

$$\leq \epsilon/3 + \frac{8u^2\Delta^2 d\sqrt{\kappa_1}}{\eta^{3/2}\sqrt{5u^2\eta^2\left((M^2+1)\frac{\Delta^2 d}{4}+\sigma^2\right)}}.$$

This completes the proof.

### E.6  Proof of Theorem 6

In this section, we analyze the convergence of VC SGHMCLP-L when the target distribution is strongly log-concave. This proof is similar to the proof of Theorem 5, however, the variance-corrected quantization function gives us a bound for the difference between the quantized value and the full-precision value. This bound can scale with the learning rate. This fact leads to the advantage of variance-corrected quantization over naive stochastic rounding. Recall the VC SGHMCLP-L update rule is the following,

$$\mathbf{v}_{k+1} = Q^{vc}\left(v_k e^{-\gamma\eta} - u\gamma^{-1}\left(1-e^{-\gamma\eta}\right)Q_G\left(\widetilde{\nabla U}(\mathbf{x}_k)\right), Var_v, \Delta\right)$$

$$\mathbf{x}_{k+1} = Q^{vc}\left(\mathbf{x}_k + \gamma^{-1}\left(1-e^{-\gamma\eta}\right)v_k + u\gamma^{-2}\left(\gamma\eta+e^{-\gamma\eta}-1\right)Q_G(\widetilde{\nabla U}(\mathbf{x}_k)), Var_x, \Delta\right). \quad (54)$$

If we let $\alpha_k^{\mathbf{x}}$ and $\alpha_k^{\mathbf{v}}$ denote the quantization error,

$$\alpha_k^{\mathbf{v}} = Q^{vc}\left(v_k e^{-\gamma\eta} - u\gamma^{-1}\left(1-e^{-\gamma\eta}\right)Q_G\left(\widetilde{\nabla U}(\mathbf{x}_k)\right), Var_v, \Delta\right) - \left(\mathbf{v}_k e^{-\gamma\eta} - u\gamma^{-1}(1-e^{\gamma\eta})Q_G(\widetilde{\nabla U}(\mathbf{x}_k)) + \xi_k^{\mathbf{v}}\right)$$

$$\alpha_k^{\mathbf{x}} = Q^{vc}\left(\mathbf{x}_k + \gamma^{-1}\left(1-e^{-\gamma\eta}\right)v_k + u\gamma^{-2}\left(\gamma\eta+e^{-\gamma\eta}-1\right)Q_G(\widetilde{\nabla U}(\mathbf{x}_k)), Var_x, \Delta\right)$$
$$- \left(\mathbf{x}_k + \gamma^{-1}(1-e^{-\gamma\eta})v_k + u\gamma^{-2}(\gamma\eta+e^{-\gamma\eta}-1)Q_G(\widetilde{\nabla U}(\mathbf{x}_k)) + \xi_k^{\mathbf{x}}\right),$$

we can rewrite the update rule as:

$$\mathbf{v}_{k+1} = \mathbf{v}_k e^{-\gamma\eta} - u\gamma^{-1}(1-e^{\gamma\eta})Q_G(\widetilde{\nabla U}(\mathbf{x}_k)) + \xi_k^{\mathbf{v}} + \alpha_k^{\mathbf{v}}$$

$$\mathbf{x}_{k+1} = \mathbf{x}_k + \gamma^{-1}(1-e^{-\gamma\eta})v_k + u\gamma^{-2}(\gamma\eta+e^{-\gamma\eta}-1)Q_G(\widetilde{\nabla U}(\mathbf{x}_k)) + \xi_k^{\mathbf{x}} + \alpha_k^{\mathbf{x}}.$$

Next, we first derive a uniform bound of $\mathbb{E}\left[\|\alpha_k^{\mathbf{v}}\|^2\right]$. In this section and the following section, we further assume the norm of quantized stochastic gradients are bounded.

**Assumption 5.** *For any $x \in \mathbb{R}^d$, there exists a constant $\mathcal{G}$ and the quantized stochastic gradients at $x$ satisfies the following*

$$\mathbb{E}\left[\left\|Q_G(\widetilde{\nabla U}(x))\right\|^2\right] \leq \mathcal{G}^2.$$

By the definition of the variance corrected quantization function $Q^{vc}$, when $Var_v > \rho_0 = \frac{\Delta^2}{4}$, if we let $\psi_k$ denote $v_k e^{-\gamma\eta} - u\gamma^{-1}\left(1 - e^{-\gamma\eta}\right) Q_G\left(\widetilde{\nabla U}(\mathbf{x}_k)\right)$,

$$
\mathbb{E}\left[\|\alpha_k^{\mathbf{V}}\|^2 \Big| \psi_k\right]
$$
$$
=\mathbb{E}\left[\left\|\left(v_k e^{-\gamma\eta} - u\gamma^{-1}\left(1 - e^{-\gamma\eta}\right) Q_G(\widetilde{\nabla U}(\mathbf{x}_k))\right) + \sqrt{Var_v}\xi_k \right.\right.
$$
$$
\left.\left. -Q^d\left(v_k e^{-\gamma\eta} - u\gamma^{-1}\left(1 - e^{-\gamma\eta}\right) Q_G(\widetilde{\nabla U}(\mathbf{x}_k)) + \sqrt{Var_v - \rho_0}\xi_k\right) - \mathrm{sign}(r)c\right\|^2 \Big| \psi_k\right]
$$

Let

$$
b = Q^d\left(v_k e^{-\gamma\eta} - u\gamma^{-1}\left(1 - e^{-\gamma\eta}\right) Q_G(\widetilde{\nabla U}(\mathbf{x}_k)) + \sqrt{Var_v - \rho_0}\xi_k\right)
$$
$$
-\left(v_k e^{-\gamma\eta} - u\gamma^{-1}\left(1 - e^{-\gamma\eta}\right) Q_G(\widetilde{\nabla U}(\mathbf{x}_k)) + \sqrt{Var_v - \rho_0}\xi_k\right),
$$

then

$$
\mathbb{E}\left[\|\alpha_k^{\mathbf{V}}\|^2 \Big| \psi_k\right]
$$
$$
=\mathbb{E}\left[\left\|\left(v_k e^{-\gamma\eta} - u\gamma^{-1}\left(1 - e^{-\gamma\eta}\right) Q_G(\widetilde{\nabla U}(\mathbf{x}_k))\right) + \sqrt{Var_v}\xi_k \right.\right.
$$
$$
\left.\left. -\left(v_k e^{-\gamma\eta} - u\gamma^{-1}\left(1 - e^{-\gamma\eta}\right) Q_G(\widetilde{\nabla U}(\mathbf{x}_k)) + \sqrt{Var_v - \rho_0}\xi_k\right) - b - \mathrm{sign}(r)c\right\|^2 \Big| \psi_k\right]
$$
$$
=\mathbb{E}\left[\left\|\sqrt{Var_v}\xi_k - \sqrt{Var_v - \rho_0}\xi_k - b - \mathrm{sign}(r)c\right\|^2 \Big| \psi_k\right]
$$
$$
\leq\mathbb{E}\left[\left\|\sqrt{Var_v}\xi_k - \sqrt{Var_v - \rho_0}\xi_k\right\|^2\right] + \mathbb{E}\left[\|b + \mathrm{sign}(r)c\|^2 \Big| \psi_k\right]
$$
$$
\leq 2Var_v d - \rho_0 d + \rho_0 d
$$
$$
\leq 4\gamma u d\eta. \tag{55}
$$

When $Var_v < \frac{\Delta_W^2}{4}$,

$$
\mathbb{E}[\|\alpha_k^{\mathbf{V}}\|^2]
$$
$$
= \mathbb{E}\left[\left\|\left(\mathbf{v}_k e^{-\gamma\eta} - u\gamma^{-1}\left(1 - e^{-\gamma\eta}\right) Q_G(\widetilde{\nabla U}(\mathbf{x}_k))\right) - \mathbf{v}_{k+1} + \sqrt{Var_v}\xi_k\right\|^2\right]
$$
$$
= \mathbb{E}\left[\left\|\left(\mathbf{v}_k e^{-\gamma\eta} - u\gamma^{-1}\left(1 - e^{-\gamma\eta}\right) Q_G(\widetilde{\nabla U}(\mathbf{x}_k))\right) - \mathbf{v}_{k+1}\right\|^2\right] + \mathbb{E}\left[\left\|\sqrt{Var_v}\xi_k\right\|^2\right]
$$
$$
\leq \max\left(2\mathbb{E}\left[\left\|\left(\mathbf{v}_k e^{-\gamma\eta} - u\gamma^{-1}\left(1 - e^{-\gamma\eta}\right) Q_G(\widetilde{\nabla U}(\mathbf{x}_k))\right) - Q^s\left(\mathbf{v}_k e^{-\gamma\eta} - u\gamma^{-1}\left(1 - e^{-\gamma\eta}\right) Q_G(\widetilde{\nabla U}(\mathbf{x}_k))\right)\right\|^2\right], 2Var_v d\right).
$$
$$
\tag{56}
$$

Using the bound equation (6) in Li & De Sa (2019) gives us,

$$
\mathbb{E}\left[\left\|\left(\mathbf{v}_k e^{-\gamma\eta} - u\gamma^{-1}\left(1 - e^{-\gamma\eta}\right) Q_G(\widetilde{\nabla U}(\mathbf{x}_k))\right) - Q^s\left(\mathbf{v}_k e^{-\gamma\eta} - u\gamma^{-1}\left(1 - e^{-\gamma\eta}\right) Q_G(\widetilde{\nabla U}(\mathbf{x}_k))\right)\right\|^2\right]
$$
$$
\leq \Delta\left(1 - e^{-\gamma\eta}\right) \mathbb{E}\left[\left\|v_k - u\gamma^{-1} Q_G(\widetilde{\nabla U}(\mathbf{x}_k))\right\|_1\right]
$$
$$
\leq \Delta\left(1 - e^{-\gamma\eta}\right) \sqrt{d}\left(\mathbb{E}\left[\|v_k\|\right] + \mathbb{E}\left[\left\|Q_G(\widetilde{\nabla U}(\mathbf{x}_k))\right\|\right]\right).
$$

Now we need to derive a uniform bound of $\mathbb{E}\left[\|v_k\|\right]$, by the update rule, we know that,

$$\mathbb{E}\left[\|\mathbf{v}_{k+1}\|^2\right] = \mathbb{E}\left[\left\|\mathbf{v}_k e^{-\gamma\eta} - u\gamma^{-1}(1 - e^{\gamma\eta})Q_G(\widetilde{\nabla U}(\mathbf{x}_k)) + \xi_k^{\mathbf{v}} + \alpha_k^{\mathbf{v}}\right\|^2\right]$$

$$\leq (1 + \gamma\eta/2)\,(1 - \gamma\eta/2)^2 \mathbb{E}\left[\|v_k\|^2\right] + \left(\frac{2}{\gamma\eta} + 1\right) u^2\eta^2 \mathbb{E}\left[\left\|Q_G(\widetilde{\nabla U})\right\|^2\right] + 2\gamma u d\eta + \mathbb{E}\left[\|\alpha_k^{\mathbf{v}}\|^2\right]$$

$$\leq (1 - \gamma\eta/2)\,\mathbb{E}\left[\|v_k\|^2\right] + 3u^2\eta/\gamma\mathcal{G}^2 + 2\gamma u d\eta + \mathbb{E}\left[\|\alpha_k^{\mathbf{v}}\|^2\right].$$

When $\mathbb{E}\left[\|\alpha_k^{\mathbf{v}}\|^2\right] \leq 2Var_v d < 4\gamma u d\eta$, the inequality can be further written as:

$$\mathbb{E}\left[\|\mathbf{v}_{k+1}\|^2\right] \leq (1 - \gamma\eta/2)\,\mathbb{E}\left[\|v_k\|^2\right] + 3u^2\eta/\gamma\mathcal{G}^2 + 6\gamma u d\eta$$

$$\leq \mathbb{E}\left[\|\mathbf{v}_0\|^2\right] + \frac{6u^2\eta\mathcal{G}^2}{\gamma^2\eta} + \frac{12\gamma u d\eta}{\gamma\eta}$$

$$\leq \mathbb{E}\left[\|\mathbf{v}_0\|^2\right] + \frac{6u^2\eta\mathcal{G}^2}{\gamma^2} + 12ud.$$

If $\mathbb{E}\left[\|\alpha_k^{\mathbf{v}}\|^2\right] \leq 2\mathbb{E}\left[\left\|\left(\mathbf{v}_k e^{-\gamma\eta} - u\gamma^{-1}\left(1 - e^{-\gamma\eta}\right)Q_G(\widetilde{\nabla U}(\mathbf{x}_k))\right) - Q^s\left(\mathbf{v}_k e^{-\gamma\eta} - u\gamma^{-1}\left(1 - e^{-\gamma\eta}\right)Q_G(\widetilde{\nabla U}(\mathbf{x}_k))\right)\right\|^2\right]$,
the inequality can be written as:

$$\mathbb{E}\left[\|\mathbf{v}_{k+1}\|^2\right] \leq (1 - \gamma\eta/2)\,\mathbb{E}\left[\|v_k\|^2\right] + 3u^2\eta/\gamma\mathcal{G}^2 + 2\gamma u d\eta + 2\Delta\left(1 - e^{-\gamma\eta}\right)\sqrt{d}\left(\mathbb{E}\left[\|v_k\|\right] + \mathbb{E}\left[\left\|Q_G(\widetilde{\nabla U}(\mathbf{x}_k))\right\|\right]\right)$$

$$\leq (1 - \gamma\eta/2)\,\mathbb{E}\left[\|v_k\|^2\right] + 3u^2\eta/\gamma\mathcal{G}^2 + 2\gamma u d\eta + 2\Delta\gamma\eta\sqrt{d}\left(\sqrt{\mathbb{E}\left[\|v_k\|^2\right]} + \mathcal{G}\right)$$

$$\leq \left(\sqrt{1 - \gamma\eta/2}\sqrt{\mathbb{E}\left[\|v_k\|^2\right]} + \frac{\Delta\gamma\eta\sqrt{d}}{\sqrt{1 - \gamma\eta/2}}\right)^2 + 3u^2\eta/\gamma\mathcal{G}^2 + 2\gamma u d\eta + 2\Delta\gamma\eta\sqrt{d}\mathcal{G}.$$

Thus,

$$\mathbb{E}\left[\|v_k\|\right] \leq \sqrt{\mathbb{E}\left[\|\mathbf{v}_0\|^2\right]} + \frac{\Delta\gamma\eta\sqrt{d}}{\left(1 - \sqrt{1 - \gamma\eta/2}\right)\sqrt{1 - \gamma\eta/2}} + \frac{3u^2\eta/\gamma\mathcal{G}^2 + 2\gamma u d\eta + 2\Delta\gamma\eta\sqrt{d}\mathcal{G}}{\frac{\Delta\gamma\eta\sqrt{d}}{\sqrt{1-\gamma\eta/2}} + \sqrt{\gamma\eta/2\left(3u^2\eta/\gamma\mathcal{G}^2 + 2\gamma u d\eta + 2\Delta\gamma\eta\sqrt{d}\mathcal{G}\right)}}$$

$$\leq \sqrt{\mathbb{E}\left[\|\mathbf{v}_0\|^2\right]} + \frac{\Delta\gamma\eta\sqrt{d}}{1 - \gamma\eta/2} + \sqrt{6u^2/\gamma^2\mathcal{G}^2 + 4ud + 4\Delta\sqrt{d}\mathcal{G}}$$

$$\leq \sqrt{\mathbb{E}\left[\|\mathbf{v}_0\|^2\right]} + \Delta\sqrt{d} + \sqrt{6u^2/\gamma^2\mathcal{G}^2 + 4ud + 4\Delta\sqrt{d}\mathcal{G}}.$$

Finally, we can have:

$$\mathbb{E}\left[\|v_k\|\right] \leq \max\left\{\sqrt{\mathbb{E}\left[\|\mathbf{v}_0\|^2\right]} + \Delta\sqrt{d} + \sqrt{6u^2/\gamma^2\mathcal{G}^2 + 4ud + 4\Delta\sqrt{d}\mathcal{G}},\right.$$

$$\left.\sqrt{\mathbb{E}\left[\|\mathbf{v}_0\|^2\right]} + \sqrt{\frac{6u^2\eta\mathcal{G}^2}{\gamma^2} + \sqrt{12ud}}\right\} =: A'.$$

Thus, we can have,

$$\mathbb{E}\left[\left\|\left(\mathbf{v}_k e^{-\gamma\eta} - u\gamma^{-1}\left(1 - e^{-\gamma\eta}\right)Q_G(\widetilde{\nabla U}(\mathbf{x}_k))\right) - Q^s\left(\mathbf{v}_k e^{-\gamma\eta} - u\gamma^{-1}\left(1 - e^{-\gamma\eta}\right)Q_G(\widetilde{\nabla U}(\mathbf{x}_k))\right)\right\|^2\right]$$

$$\leq \Delta\gamma\eta\sqrt{d}\left(A' + \mathcal{G}\right),$$

and we can bound the $\mathbb{E}\left[\|\alpha_k^{\mathbf{y}}\|^2\right]$ as,

$$\mathbb{E}\left[\|\alpha_k^{\mathbf{y}}\|^2\right] \leq \max\left\{\Delta\gamma\eta\sqrt{d}\left(A' + \mathcal{G}\right), 4\gamma u d\eta\right\}$$
$$= \gamma\eta\max\left\{\Delta\sqrt{d}\left(A' + \mathcal{G}\right), 4ud\right\}$$
$$=: \gamma\eta A. \tag{57}$$

Now we bound the $\mathbb{E}\left[\|\alpha_k^{\mathbf{x}}\|^2\right]$. When $Var_x \geq \rho_0$, as the same analysis in (55) we can show,

$$\mathbb{E}\left[\|\alpha_k^{\mathbf{x}}\|^2\right] \leq 2Var_x d \leq 4ud\eta^2.$$

If $Var_x < \rho_0$, and let $\mu_x = \mathbf{x}_k + \gamma^{-1}\left(1 - e^{-\gamma\eta}\right)v_k + u\gamma^{-2}\left(\gamma\eta + e^{-\gamma\eta} - 1\right)Q_G(\widetilde{\nabla U}(\mathbf{x}_k))$, by the same analysis in (56) we can have:

$$\mathbb{E}\left[\|\alpha_k^{\mathbf{x}}\|^2\right]$$
$$\leq \max\left\{2\mathbb{E}\left[\|\mu_x - Q^s\left(\mu_x\right)\|^2\right], 2Var_x d\right\}.$$

Again using the bound equation (6) in Li & De Sa (2019) gives us,

$$\mathbb{E}\left[\|\mu_x - Q^s(\mu_x)\|^2\right] \leq \Delta\mathbb{E}\left[\left\|\gamma^{-1}\left(1 - e^{-\gamma\eta}\right)v_k + u\gamma^{-2}\left(\gamma\eta + e^{-\gamma\eta} - 1\right)Q_G(\widetilde{\nabla U}(\mathbf{x}_k))\right\|_1\right]$$
$$\leq \Delta\eta\mathbb{E}\left[\|v_k\|_1\right] + \frac{u\eta^2}{2}\mathbb{E}\left[\left\|Q_G(\widetilde{\nabla U}(\mathbf{x}_k))\right\|_1\right]$$
$$\leq \Delta\eta\sqrt{d}\mathbb{E}\left[\|v_k\|\right] + \frac{u\eta^2}{2}\sqrt{d}\mathbb{E}\left[\left\|Q_G(\widetilde{\nabla U}(\mathbf{x}_k))\right\|\right]$$
$$\leq \Delta\eta\sqrt{d}A' + \frac{u\eta^2}{2}\sqrt{d}\mathcal{G}.$$

Thus, we can have,

$$\mathbb{E}\left[\|\alpha_k^{\mathbf{x}}\|^2\right] \leq \max\left\{2\Delta\eta\sqrt{d}A' + u\eta^2\sqrt{d}\mathcal{G}, 4ud\eta^2\right\}$$
$$\leq \eta\max\left\{2\Delta\sqrt{d}A' + u\eta\sqrt{d}\mathcal{G}, 4ud\eta\right\}$$
$$=: \eta B. \tag{58}$$

Then follow the same analysis of (53), we can show

$$\mathcal{W}_2(p_K, p^*) \leq 4e^{-K\eta/2\kappa_1}\mathcal{W}_2(q_0, q^*) + \frac{4\eta^2\sqrt{\frac{8\mathcal{E}_K}{5}}}{1 - e^{-\eta/2\kappa_1}} \tag{59}$$

$$+ \frac{20u^2\eta^2\left(\frac{\Delta^2 d}{4} + \sigma^2\right) + 8u^2\eta\left(\gamma A + B\right)}{\eta^2\sqrt{\frac{8\mathcal{E}_K}{5}} + \sqrt{1 - e^{-\eta/\kappa_1}}\sqrt{5u^2\eta^2\left(\frac{\Delta^2 d}{4} + \sigma^2\right) + 2u^2\eta\left(\gamma A + B\right)}}. \tag{60}$$

Now we let the first term less than $\epsilon/3$, from the Lemma 13 in (Cheng et al., 2018) we know that $\mathcal{W}_2(q_0, q^*) \leq 3\left(\frac{d}{m_1} + \mathcal{D}^2\right)$. So we can choose $K$ as the following,

$$K \leq \frac{2\kappa_1}{\eta}\log\left(36\left(\frac{d}{m_1} + \mathcal{D}^2\right)\right).$$

Next, we choose a step size $\eta \leq \frac{\epsilon \kappa_1^{-1}}{\sqrt{479232/5(d/m_1 + \mathcal{D}^2)}}$ to ensure the second term is controlled below $\epsilon/3$. Since $1 - e^{-\eta/2\kappa_1} \geq \eta/4\kappa_1$ and definition of $\mathcal{E}_K$,

$$4\frac{\eta^2\sqrt{\frac{8\mathcal{E}_K}{5}}}{1 - e^{-\eta/2\kappa_1}} \leq 4\frac{\eta^2\sqrt{\frac{8\mathcal{E}_K}{5}}}{\eta/4\kappa_1} \leq 16\kappa_1\left(\eta\sqrt{\frac{8\mathcal{E}_K}{5}}\right) \leq \epsilon/3.$$

Finally choosing the step size satisfied that,

$$\eta \leq \frac{\epsilon^2}{2880\kappa_1 u\left(\frac{\Delta^2 d}{4} + \sigma^2\right)},$$

the third term can be bounded as:

$$\frac{20u^2\eta^2\left(\frac{\Delta^2 d}{4} + \sigma^2\right) + 8u^2\eta\left(\gamma A + B\right)}{\eta^2\sqrt{\frac{8\mathcal{E}_K}{5}} + \sqrt{1 - e^{-\eta/\kappa_1}}\sqrt{5u^2\eta^2\left(\frac{\Delta^2 d}{4} + \sigma^2\right) + 2u^2\eta\left(\gamma A + B\right)}}$$

$$\leq \frac{20u^2\eta^2\left(\frac{\Delta^2 d}{4} + \sigma^2\right) + 8u^2\eta\left(\gamma A + B\right)}{\sqrt{1 - e^{-\eta/\kappa_1}}\sqrt{5u^2\eta^2\left(\frac{\Delta^2 d}{4} + \sigma^2\right) + 2u^2\eta\left(\gamma A + B\right)}} \leq \frac{20u^2\eta^2\left(\frac{\Delta^2 d}{4} + \sigma^2\right) + 8u^2\eta\left(\gamma A + B\right)}{\sqrt{\eta/4\kappa_1}\sqrt{5u^2\eta^2\left(\frac{\Delta^2 d}{4} + \sigma^2\right) + 2u^2\eta\left(\gamma A + B\right)}}$$

$$\leq 4\sqrt{20u^2\kappa_1\eta\left(\frac{\Delta^2 d}{4} + \sigma^2\right) + 8\kappa_1 u^2\left(\gamma A + B\right)}$$

$$\leq \epsilon/3 + 8\sqrt{2\kappa_1 u^2\left(\gamma A + B\right)}.$$

This completes the proof.

### E.7 Proof of Thoerem 7

In this section we generalize the convergence analysis of LPSGLDLP-F in Zhang et al. (2022) to non-log-concave target distribution. We prove a more general version of theorem 7 following the same proof outlines in Raginsky et al. (2017). We further introduce an assumption about the initial distribution $p_0$.

**Assumption 6.** *The probability $p_0$ of the initial hypothesis $\mathbf{x}_0$ has a bounded and strictly positive density and satisfies the following:*

$$\kappa_0 := \log \int_{\mathbb{R}^d} e^{\|x\|^2} p_0(x) dx < \infty.$$

Note that the for initial distribution $\mathbf{x}_0 = 0$, the value $\kappa_0 = 0$ is bounded and the assumption is satisfied. Recall the Overdamped Langevin dynamics is

$$d\mathbf{x}_t = -\nabla U(\mathbf{x}_t)dt + \sqrt{2}dB_t. \tag{61}$$

We further define the value of the energy function and the gradient at point 0 at the following:

$$|U(0)| = G_0, \quad \|\nabla U(0)\| = G_1.$$

In order to analyze the convergence of SGLD for non-log-concave distribution, we need to introduce extra assumptions.

Then the solution of the Langevin dynamics should satisfies

$$\mathbf{x}_t = \mathbf{x}_0 - \int_0^t \nabla U(\mathbf{x}_s)ds + \sqrt{2}\int_0^t dB_s. \tag{62}$$

To analyze the LPSGLDLP-F in (1), we define a continuous interpolation of the low-precision sample as:

$$\hat{x}_t = \hat{x}_0 - \int_0^t G_s ds + \sqrt{2}\int_0^t dB_s. \tag{63}$$

where $G_s = \sum_{k=0}^{K} \tilde{g}(\hat{x}_k)\mathbf{1}_{s\in[k\eta,(k+1)\eta)}$. The Wasserstein distance can be bounded as

$$\mathcal{W}_2(p_K, p^*) \leq \mathcal{W}_2(p_K, \hat{p}_{K\eta}) + \mathcal{W}_2(\hat{p}_{K\eta}, p^*). \tag{64}$$

Now, we are ready to introduce the contraction rate $\lambda^*$ of the overdamped Langevin dynamics (13). By borrowing the proposition 9 of (Raginsky et al., 2017), we can have:

**Lemma 14** (Proposittion 9 in Raginsky et al. (2017)). *Suppose Assumptions 1 and 2 hold. Then*

$$
\begin{aligned}
\mathcal{W}_2(\hat{p}_{K\eta}, p^*) &\leq \sqrt{2C_{LS}\left(\log\|p_0\|_\infty + \frac{d}{2}\log\frac{3\pi}{m} + \left(\frac{M\kappa_0}{3} + B\sqrt{\kappa_0} + G_0 + \frac{b}{2}\log 3\right)\right)}e^{-K\eta/C_{LS}} \\
&\leq \widetilde{C_3}e^{-K\eta/C_{LS}},
\end{aligned}
\tag{65}
$$

*with constant*

$$C_{LS} \leq \frac{2m_2^2 + 8M^2}{m_2^2 M} + \frac{1}{\lambda^*}\left(\frac{6Md}{m_2} + 2\right).$$

*where the $\lambda^* = e^{-\tilde{\mathcal{O}}(d)}$ denotes the contraction rate of overdamped Langevin dynamics.*

Here, $\lambda^*$ acts as a contraction rate of the Markov process initiated by (13), with an exponential dependency on the dimension $d$ being inescapable in the worst-case scenario proved in Appendix B in (Raginsky et al., 2017).

The first term of equation 64 can be bounded via the weighted CKP inequality

$$\mathcal{W}_2(p_K, \hat{p}_{K\eta}) \leq C_{\hat{p}_{K\eta}}\left[\sqrt{D_{KL}\left(p_K\|\hat{p}_{K\eta}\right)} + \left(\frac{D_{KL}\left(p_K\|\hat{p}_{K\eta}\right)}{2}\right)^{1/4}\right],$$

where the constant $C_{\hat{p}_{K\eta}} = 2\inf_{\lambda>0}\left(\frac{1}{\lambda}\left(\frac{3}{2} + \log\int_{\mathbb{R}^d}e^{\lambda\|\omega\|^2}\hat{P}_{K\eta}(d\omega)\right)\right)$. By Lemma 4 in Raginsky et al. (2017) and assuming $K\eta > 1$, we can wrtie:

$$\mathcal{W}_2^2(p_K, \hat{p}_{K\eta}) \leq \left(12 + 8\left(\kappa_0 + 2b + 2d\right)K\eta\right)\left(D_{KL}\left(p_K\|\hat{p}_{K\eta}\right) + \sqrt{D_{KL}\left(p_K\|\hat{p}_{K\eta}\right)}\right).$$

Now we bound the term $D_{KL}\left(p_K\|\hat{p}_{K\eta}\right)$. The Radon-Nikodym derivative of the $\hat{P}_{K\eta}$ w.r.t $p_K$ is the following

$$\frac{d\hat{p}_{K\eta}}{dp_K} = exp\left\{\frac{1}{2}\int_0^t(\nabla U(\mathbf{x}_s) - G_s)d\mathbf{B}s - \frac{1}{4}\int_0^T\|\nabla U(\mathbf{x}_s) - G_s\|ds\right\}.$$

Thus, we have:

$$
\begin{aligned}
D_{KL}(p_K||\hat{p}_{K\eta}) &= \mathbb{E}_{p_K}\left[\log\left(\frac{d\hat{p}_{K\eta}}{dp_K}\right)\right] \\
&= \frac{1}{4}\int_0^{K\eta}\mathbb{E}\left[\|\nabla U(\mathbf{x}_s) - G_s\|^2\right]ds \\
&= \frac{1}{4}\sum_{k=0}^{K-1}\int_{k\eta}^{(k+1)\eta}\mathbb{E}\left[\|\nabla U(\mathbf{x}_s) - \tilde{g}(\mathbf{x}_k)\|^2\right]ds \\
&\leq \frac{1}{2}\sum_{k=0}^{K-1}\int_{k\eta}^{(k+1)\eta}\mathbb{E}\left[\|\nabla U(\mathbf{x}_s) - \nabla U(\mathbf{x}_k)\|^2\right] \\
&\quad + \frac{1}{2}\sum_{k=0}^{K-1}\int_{k\eta}^{(k+1)\eta}\mathbb{E}\left[\|\nabla U(\mathbf{x}_k) - \tilde{g}(\mathbf{x}_k)\|^2\right] \\
&\leq \frac{M^2}{2}\sum_{k=0}^{K-1}\int_{k\eta}^{(k+1)\eta}\mathbb{E}\left[\|\mathbf{x}_s - \mathbf{x}_k\|^2\right] \\
&\quad + \frac{1}{2}\sum_{k=0}^{K-1}\int_{k\eta}^{(k+1)\eta}\mathbb{E}\left[\|\nabla U(\mathbf{x}_k) - \tilde{g}(\mathbf{x}_k)\|^2\right].
\end{aligned}
\tag{66}
$$

We now bound the first term in the RHS of the equation (66), from the update rule in (63) we know:

$$
\begin{aligned}
\mathbf{x}_s - \mathbf{x}_k &= -(s - k\eta)\tilde{g}(\mathbf{x}_k) + \sqrt{2}\left(B_s - B_{k\eta}\right) \\
&= -(s - k\eta)\nabla U(\mathbf{x}_k) + (s - k\eta)\left(\nabla U(\mathbf{x}_k) - \tilde{g}(\mathbf{x}_k)\right) + \sqrt{2}\left(B_s - B_{k\eta}\right),
\end{aligned}
$$

thus,

$$
\begin{aligned}
\mathbb{E}\left[\|\mathbf{x}_s - \mathbf{x}_k\|^2\right] &\leq 3\eta^2\mathbb{E}\left[\|\nabla U(\mathbf{x}_k)\|^2\right] + 3\eta^2\mathbb{E}\left[\|\nabla U(\mathbf{x}_k) - \tilde{g}(\mathbf{x}_k)\|^2\right] + 6\eta d \\
&\leq 3\eta^2\left(M\mathbb{E}\left[\|\mathbf{x}_k\|\right] + G\right)^2 + 3\eta^2\left((M^2+1)\frac{\Delta^2 d}{4} + \sigma^2\right) + 6\eta d.
\end{aligned}
\tag{67}
$$

Similarly, we need a uniform bound of $\mathbb{E}\left[\|\mathbf{x}_k\|^2\right]$.

**Lemma 15.** *Under assumptions 1, 2 and 3, if we set the step size $\eta \in \left(0, 1 \wedge \frac{m_2}{2M^2}\right)$, then for all $k \geq 0$, the $\mathbb{E}\left[\|\mathrm{v}\mathbf{x}_k\|^2\right]$ can be bounded as*

$$
\mathbb{E}\left[\|\mathbf{x}_k\|^2\right] \leq \mathcal{E} + \frac{2\left(M^2+1\right)\Delta^2 d}{4m_2},
$$

*provided $\mathcal{E} = \mathbb{E}\left[\|\mathbf{x}_0\|^2\right] + \frac{M}{m_2}\left(2b + 2\eta G^2 + 2d\right).$*

The proof of Lemma 15 can be found in Appendix F.4. Using this bound, we can further bound $\mathbb{E}\left[\|\mathbf{x}_s - \mathbf{x}_s\|^2\right]$ as:

$$
\begin{aligned}
\mathbb{E}\left[\|\mathbf{x}_s - \mathbf{x}_s\|^2\right] &\leq 6\eta^2 M^2\left(\mathcal{E} + \frac{2\left(M^2+1\right)}{m_2}\frac{\Delta^2 d}{4}\right) + 6\eta^2 G^2 + 3\eta^2\left((M^2+1)\frac{\Delta^2 d}{4} + \sigma^2\right) + 6\eta d \\
&\leq 6\eta^2 M^2\mathcal{E} + 6\eta^2 G^2 + 6\eta d + \left(\frac{12\eta^2 M^2\left(M^2+1\right)}{m_2} + 3(M^2+1)\right)\eta^2\frac{\Delta^2 d}{4} + 3\eta^2\sigma^2, \\
&=: \overline{\mathcal{E}}\eta + C\eta^2\frac{\Delta^2 d}{4} + 3\eta^2\sigma^2
\end{aligned}
$$

where the costant $\mathcal{E}$ and $C$ are defined as:

$$\overline{\mathcal{E}} = 6M^2\mathcal{E} + 6G^2 + 6d$$
$$C = \frac{12\eta^2 M^2 \left(M^2 + 1\right)}{m_2} + 3(M^2 + 1).$$

Thus the divergence can be bounded as:

$$D_{KL}(p_K||\hat{p}_{K\eta}) \leq \frac{M^2}{2} \left(\overline{\mathcal{E}} + C\eta\frac{\Delta^2 d}{4} + 3\eta\sigma^2\right) K\eta^2 + \frac{1}{2} \left((M^2 + 1)\frac{\Delta^2 d}{4} + \sigma^2\right) K\eta$$
$$= \frac{M^2}{2}\overline{\mathcal{E}}K\eta^2 + \left(\frac{M^2}{2}C\eta^2 + \frac{1}{2}(M^2 + 1)\right) \frac{\Delta^2 d}{4}K\eta + \frac{3M^2\eta^2 + 1}{2}\sigma^2 K\eta$$
$$= \frac{M^2}{2}\overline{\mathcal{E}}K\eta^2 + \left(\frac{M^2}{2}C + \frac{1}{2}(M^2 + 1)\right) \frac{\Delta^2 d}{4}K\eta + \frac{3M^2 + 1}{2}\sigma^2 K\eta$$
$$=: C_0 K\eta^2 + C_1 \frac{\Delta^2 d}{4}K\eta + C_2\sigma^2 K\eta.$$

We are ready to bound the Wasserstein distance,

$$\mathcal{W}_2^2(p_K, \hat{p}_{K\eta}) \leq (12 + 8\left(\kappa_0 + 2b + 2d\right)) \left((C_0 + \sqrt{C_0})\sqrt{\eta} + \left(C_1 + \sqrt{C_1}\right)A + \left(C_2 + \sqrt{C_2}\right)B\right)(K\eta)^2$$
$$=: \left(\widetilde{C_0}^2\sqrt{\eta} + \widetilde{C_1}^2 A + \widetilde{C_2}^2 B\right)(K\eta)^2,$$

where the constants are defined as:

$$A = \max\left\{\frac{\Delta^2 d}{4}, \sqrt{\frac{\Delta^2 d}{4}}\right\}$$
$$B = \max\left\{\sigma^2, \sqrt{\sigma^2}\right\}$$
$$\widetilde{C_0}^2 = (12 + 8\left(\kappa_0 + 2b + 2d\right)) \left(C_0 + \sqrt{C_0}\right)$$
$$\widetilde{C_1}^2 = (12 + 8\left(\kappa_0 + 2b + 2d\right)) \left(C_1 + \sqrt{C_1}\right)$$
$$\widetilde{C_2}^2 = (12 + 8\left(\kappa_0 + 2b + 2d\right)) \left(C_2 + \sqrt{C_2}\right).$$

From Proposition 9 in the paper Raginsky et al. (2017), we know that

$$\mathcal{W}_2(\hat{p}_{K\eta}, p^*) \leq \sqrt{2C_{LS}\left(\log\|p_0\|_\infty + \frac{d}{2}\log\frac{3\pi}{m} + \left(\frac{M\kappa_0}{3} + B\sqrt{\kappa_0} + G_0 + \frac{b}{2}\log 3\right)\right)}e^{-K\eta/C_{LS}}$$
$$=: \widetilde{C_3}e^{-K\eta/C_{LS}}$$

Finally, we can have

$$\mathcal{W}_2(p_K, p^*) \leq \left(\widetilde{C_0}\eta^{1/4} + \widetilde{C_1}\sqrt{A} + \widetilde{C_2}\sqrt{B}\right)K\eta + \widetilde{C_3}e^{-K\eta/C_{LS}}. \tag{68}$$

To bound the Wasserstein distance, we need to set

$$\widetilde{C_0}K\eta^{5/4} = \frac{\epsilon}{2} \quad \text{and} \quad \widetilde{C_3}e^{-K\eta/C_{LS}} = \frac{\epsilon}{2}. \tag{69}$$

Solving the (69), we can have

$$K\eta = C_{LS}\log\left(\frac{2\widetilde{C_3}}{\epsilon}\right) \quad \text{and} \quad \eta = \frac{\epsilon^4}{16\widetilde{C_0}^4 (K\eta)^4}.$$

Combining these two we can have

$$\eta = \frac{\epsilon^4}{16\widetilde{C_0}^4 C_{LS}^4 \log^4\left(\frac{2\widetilde{C_3}}{\epsilon}\right)} \quad \text{and} \quad K = \frac{16\widetilde{C_0}^4 C_{LS}^5 \log^5\left(\frac{2\widetilde{C_3}}{\epsilon}\right)}{\epsilon^4}.$$

Plugging $K$ and $\eta$ into (68) completes the proof.

### E.8 Proof of Theorem 8

In this section we generalize the convergence analysis of SGLDLP-L in Zhang et al. (2022) to non-log-concave target distribution. Following the same proof outlines in Raginsky et al. (2017). Recall the SGLDLP-L update rule (2) is the following,

$$\mathbf{x}_{k+1} = Q_W(\mathbf{x}_k - \eta\widetilde{\nabla U}(\mathbf{x}_k) + \sqrt{2\eta}\xi_{k+1})$$
$$=: \mathbf{x}_k - \eta\widetilde{\nabla U}(\mathbf{x}_k) + \sqrt{2\eta}\xi_{k+1} + \alpha_k,$$

where $\alpha_k$ is defined as:

$$\alpha_k = Q_W(\mathbf{x}_k - \eta\widetilde{\nabla U}(\mathbf{x}_k) + \sqrt{2\eta}\xi_{k+1}) - \mathbf{x}_k - \eta\widetilde{\nabla U}(\mathbf{x}_k) + \sqrt{2\eta}\xi_{k+1}.$$

Thus, we can define a continuous interpolation of the SGLDLP-L as:

$$\mathbf{x}_t = \mathbf{x}_0 - \int_0^t G_s ds + \sqrt{2}\int_0^t dB(s) + \int_0^t \alpha(s)ds,$$

where $G_s = \sum_{k=0}^{\infty} Q_G(\widetilde{\nabla U}(\mathbf{x}_k))\mathbf{1}_{s\in(k\eta,(k+1)\eta)}$ and $\alpha(s) = \sum_{k=0}^{\infty} \alpha_k/\eta\mathbf{1}_{s\in(k\eta,(k+1)\eta)}$. By taking the difference of the interpolation with the discrete estimation of Langevin process in equation (62), we can derive the Radon-Nikodym derivative of the $\hat{p}_{K\eta}$ w.r.t $p_K$ as:

$$\frac{d\hat{p}_{K\eta}}{dp_K} = exp\left\{\frac{1}{2}\int_0^t (\nabla U(\mathbf{x}_s) - G_s - \alpha(s))d\mathbf{B}s - \frac{1}{4}\int_0^T \|\nabla U(\mathbf{x}_s) - G_s - \alpha(s)\|^2 ds\right\}.$$

Thus, the divergence can be computed as:

$$\begin{aligned}
D_{KL}(p_K\|\hat{p}_{K\eta}) =&\frac{1}{4}\int_0^{K\eta} \mathbb{E}\left[\|\nabla U(\mathbf{x}_s) - G_s - \alpha(s)\|^2\right] ds \\
=&\frac{1}{4}\sum_{k=0}^{K-1}\int_{k\eta}^{(k+1)\eta} \mathbb{E}\left[\left\|\nabla U(\mathbf{x}_s) - Q_G(\widetilde{\nabla U}(\mathbf{x}_k)) - \alpha_k/\eta\right\|^2\right] ds \\
=&\frac{1}{4}\sum_{k=0}^{K-1}\int_{k\eta}^{(k+1)\eta} \mathbb{E}\left[\left\|\nabla U(\mathbf{x}_s) - Q_G(\widetilde{\nabla U}(\mathbf{x}_k))\right\|^2\right] ds + \frac{1}{4}\sum_{k=0}^{K-1}\int_{k\eta}^{(k+1)\eta} \mathbb{E}\left[\|\alpha_k/\eta\|^2\right] ds \\
=&\frac{1}{4}\sum_{k=0}^{K-1}\int_{k\eta}^{(k+1)\eta} \mathbb{E}\left[\|\nabla U(\mathbf{x}_s) - \nabla U(\mathbf{x}_k)\|^2\right] ds + \frac{1}{4}\sum_{k=0}^{K-1}\int_{k\eta}^{(k+1)\eta} \mathbb{E}\left[\left\|\nabla U(\mathbf{x}_k) - Q_G(\widetilde{\nabla U}(\mathbf{x}_k))\right\|^2\right] ds \\
&+ \frac{1}{4}\sum_{k=0}^{K-1}\int_{k\eta}^{(k+1)\eta} \mathbb{E}\left[\|\alpha_k/\eta\|^2\right] ds \\
\leq&\frac{M^2}{4}\sum_{k=0}^{K-1}\int_{k\eta}^{(k+1)\eta} \mathbb{E}\left[\|\mathbf{x}_s - \mathbf{x}_k\|^2\right] ds + \frac{1}{4}\sum_{k=0}^{K-1}\int_{k\eta}^{(k+1)\eta} \mathbb{E}\left[\left\|\nabla U(\mathbf{x}_k) - Q_G(\widetilde{\nabla U}(\mathbf{x}_k))\right\|^2\right] ds \\
&+ \frac{1}{4}\sum_{k=0}^{K-1}\int_{k\eta}^{(k+1)\eta} \mathbb{E}\left[\|\alpha_k/\eta\|^2\right] ds.
\end{aligned} \tag{70}$$

From the same analysis in (25), we know that

$$\mathbb{E}\left[\|\mathbf{x}_s - \mathbf{x}_k\|^2\right] \le 3\eta^2 \mathbb{E}\left[\|\nabla U(\mathbf{x}_k)\|^2\right] + 3\eta^2 \mathbb{E}\left[\left\|\nabla U(\mathbf{x}_k) - Q_G(\widetilde{\nabla U}(\mathbf{x}_k))\right\|^2\right] + 6\eta d$$

$$\le 3\eta^2 \left(M\mathbb{E}\left[\|\mathbf{x}_k\|^2\right] + G\right)^2 + 3\eta^2 \left(\frac{\Delta^2 d}{4} + \sigma^2\right) + 6\eta d.$$

Again, we need to derive a uniform bound of $\mathbb{E}\left[\|\mathbf{x}_k\|^2\right]$,

$$\mathbb{E}\left[\|\mathbf{x}_{k+1}\|^2\right] = \mathbb{E}\left[\left\|\mathbf{x}_k - \eta Q_G(\widetilde{\nabla U}(\mathbf{x}_k))\right\|^2\right] + 2\mathbb{E}\left[\|\xi_{k+1}\|^2\right] + \mathbb{E}\left[\|\alpha_k\|^2\right]$$

$$= \mathbb{E}\left[\left\|\mathbf{x}_k - \eta\nabla U(\mathbf{x}_k) + \eta\nabla U(\mathbf{x}_k) - \eta Q_G(\widetilde{\nabla U}(\mathbf{x}_k))\right\|^2\right] + 2\eta d + \mathbb{E}\left[\|\alpha_k\|^2\right]$$

$$= \mathbb{E}\left[\left\|\mathbf{x}_k - \eta\nabla U(\mathbf{x}_k) + \eta\nabla U(\mathbf{x}_k) - \eta Q_G(\widetilde{\nabla U}(\mathbf{x}_k))\right\|^2\right] + \mathbb{E}\left[\|\alpha_k\|^2\right] + 2\eta d$$

$$= \mathbb{E}\left[\|\mathbf{x}_k - \eta\nabla U(\mathbf{x}_k)\|^2\right] + \eta^2 \mathbb{E}\left[\left\|\nabla U(\mathbf{x}_k) - Q_G(\widetilde{\nabla U}(\mathbf{x}_k))\right\|^2\right] + \mathbb{E}\left[\|\alpha_k\|^2\right] + 2\eta d.$$

By plugging in the inequality we derived before:

$$\mathbb{E}\left[\|\mathbf{x}_k - \eta\nabla U(\mathbf{x}_k)\|^2\right] \le \left(1 - 2\eta m_2 + 2\eta^2 M^2\right) \mathbb{E}\left[\|\mathbf{x}_k\|^2\right] + 2\eta b + 2\eta^2 G^2.$$

we can have:

$$\mathbb{E}\left[\|\mathbf{x}_{k+1}\|^2\right] \le \left(1 - 2\eta m_2 + 2\eta^2 M^2\right) \mathbb{E}\left[\|\mathbf{x}_k\|^2\right] + 2\eta b + 2\eta^2 G^2 + \frac{\eta^2 \Delta^2 d}{4} + \eta^2 \sigma^2 + \mathbb{E}\left[\|\alpha_k\|^2\right] + 2\eta d. \quad (71)$$

Thus for any $\eta \in (0, 1 \wedge \frac{m_2}{2M^2})$ and $1 - 2\eta m_2 + 2\eta^2 M^2 > 0$, we can bound $\mathbb{E}\left[\|\mathbf{x}_k\|^2\right]$ for any $k > 0$ as:

$$\mathbb{E}\left[\|\mathbf{x}_k\|^2\right] \le \mathbb{E}\left[\|\mathbf{x}_0\|^2\right] + \frac{1}{2(m_2 - \eta M^2)}\left(2b + 2G^2 + \frac{\Delta^2 d}{4} + \sigma^2 + 2d\right) + \frac{\mathbb{E}\left[\|\alpha_k\|^2\right]}{2\eta(m_2 - \eta M^2)}$$

$$\le \mathbb{E}\left[\|\mathbf{x}_0\|^2\right] + \frac{1}{m_2}\left(2b + 2G^2 + \frac{\Delta^2 d}{4} + \sigma^2 + 2d\right) + \frac{\mathbb{E}\left[\|\alpha_k\|^2\right]}{\eta m_2}$$

$$\le \mathcal{E} + \frac{\Delta^2 d}{4 m_2} + \frac{\mathbb{E}\left[\|\alpha_k\|^2\right]}{\eta m_2},$$

where the constant $\mathcal{E}$ is defined as:

$$\mathcal{E} = \mathbb{E}\left[\|\mathbf{x}_0\|^2\right] + \frac{1}{m_2}\left(2b + 2G^2 + \sigma^2 + 2d\right).$$

Thus, we can have,

$$\mathbb{E}\left[\|\mathbf{x}_s - \mathbf{x}_k\|^2\right] \le 6\eta^2\left(\mathcal{E} + \frac{\Delta^2 d}{4 m_2} + \frac{\mathbb{E}\left[\|\alpha_k\|^2\right]}{\eta m_2}\right) + 6\eta^2 G^2 + 3\eta^2\left(\frac{\Delta^2 d}{4} + \sigma^2\right) + 6\eta d$$

$$\le \overline{\mathcal{E}}\eta + 3\eta^2\sigma^2 + \frac{6 + 3m_2}{4m_2}\eta^2\Delta^2 d + \frac{6\eta\mathbb{E}\left[\|\alpha_k\|^2\right]}{m_2}.$$

Plugging this into the equation (70), we can have,

$$
\begin{aligned}
D_{KL}(p_K||\hat{p}_{K\eta}) \leq & \frac{M\overline{\mathcal{E}}}{4}K\eta^2 + \frac{3M\sigma^2 K\eta^3}{4} + \frac{(6+3m_2)M\Delta^2 d}{16m_2}K\eta^3 + \frac{6M\mathbb{E}\left[\|\alpha_k\|^2\right]K\eta^2}{4m_2} + \frac{1}{4}\left(\frac{\Delta^2 d}{4} + \sigma^2\right)K\eta + \frac{K\mathbb{E}\left[\|\alpha_k\|^2\right]}{4\eta} \\
\leq & \frac{M\overline{\mathcal{E}}}{4}K\eta^2 + \frac{3M+1}{4}\sigma^2 K\eta + \frac{((6+3m_2)M+m_2)d}{16m_2}\Delta^2 K\eta + \left(\frac{6M\eta}{4m_2} + \frac{1}{4\eta}\right)K\mathbb{E}\left[\|\alpha_k\|^2\right].
\end{aligned}
$$

By the fact that $\mathbb{E}\left[\|\alpha_k\|^2\right] \leq \frac{\Delta^2 d}{4}$, we can further bound the divergence as:

$$
\begin{aligned}
D_{KL}(p_K||\hat{p}_{K\eta}) \leq & \frac{M\overline{\mathcal{E}}}{4}K\eta^2 + \frac{3M+1}{4}\sigma^2 K\eta + \left(\frac{((12+3m_2)M+m_2)d}{16m_2} + \frac{d}{16\eta}\right)\Delta^2 K \\
=: & C_0 K\eta^2 + C_1\sigma^2 K\eta + C_2\Delta^2 K,
\end{aligned}
$$

where the constants are defined as:

$$
\begin{aligned}
C_0 &= \frac{M\overline{\mathcal{E}}}{4} \\
C_1 &= \frac{3M+1}{4} \\
C_2 &= \left(\frac{((12+3m_2)M+m_2)d}{16m_2} + \frac{d}{16\eta}\right).
\end{aligned}
$$

We are ready to bound the Wasserstein distance,

$$
\begin{aligned}
\mathcal{W}_2^2(p_K,\hat{p}_{K\eta}) &\leq (12+8(\kappa_0+2b+2d))\left[\left(C_0+\sqrt{C_0}+\left(C_1+\sqrt{C_1}\right)A\right)(K\eta)^2+\left(C_2+\sqrt{C_2}\right)\Delta K^2\eta\right] \\
&=: \left(\widetilde{C_0}^2\sqrt{\eta}+\widetilde{C_1}^2 A\right)(K\eta)^2+\widetilde{C_2}^2\Delta K^2\eta,
\end{aligned}
$$

where the constants are defined as:

$$
\begin{aligned}
A &= \max\left\{\sigma^2,\sqrt{\sigma^2}\right\} \\
\widetilde{C_0}^2 &= (12+8(\kappa_0+2b+2d))\left(C_0+\sqrt{C_0}\right) \\
\widetilde{C_1}^2 &= (12+8(\kappa_0+2b+2d))\left(C_1+\sqrt{C_1}\right) \\
\widetilde{C_2}^2 &= (12+8(\kappa_0+2b+2d))\left(C_2+\sqrt{C_2}\right).
\end{aligned}
$$

From Proposition 9 in the paper Raginsky et al. (2017), we know that

$$
\begin{aligned}
\mathcal{W}_2(\hat{p}_{K\eta},p^*) &\leq \sqrt{2C_{LS}\left(\log\|p_0\|_\infty + \frac{d}{2}\log\frac{3\pi}{m} + \left(\frac{M\kappa_0}{3} + B\sqrt{\kappa_0} + G_0 + \frac{b}{2}\log 3\right)\right)}e^{-K\eta/C_{LS}} \\
&=: \widetilde{C_3}e^{-K\eta/C_{LS}}
\end{aligned}
$$

Finally, we can have

$$
\mathcal{W}_2(p_K,p^*) \leq \left(\widetilde{C_0}\eta^{1/4}+\widetilde{C_1}\sqrt{A}\right)K\eta+\widetilde{C_2}\sqrt{\Delta}\sqrt{K^2\eta}+\widetilde{C_3}e^{-K\eta/C_{LS}}. \tag{72}
$$

To bound the 2-Wasserstein distance, we need to set

$$
\widetilde{C_0}K\eta^{5/4} \leq \frac{\epsilon}{2} \quad \text{and} \quad \widetilde{C_3}e^{-K\eta/C_{LS}} = \frac{\epsilon}{2}. \tag{73}
$$

Solving the (73), we can have

$$K\eta = C_{LS} \log\left(\frac{2\widetilde{C_3}}{\epsilon}\right) \quad \text{and} \quad \eta \leq \frac{\epsilon^4}{16\widetilde{C_0}^4 (K\eta)^4}.$$

Combining these two we can have

$$\eta \leq \frac{\epsilon^4}{16\widetilde{C_0}^4 C_{LS}^4 \log^4\left(\frac{2\widetilde{C_3}}{\epsilon}\right)} \quad \text{and} \quad K \geq \frac{16\widetilde{C_0}^4 C_{LS}^5 \log^5\left(\frac{2\widetilde{C_3}}{\epsilon}\right)}{\epsilon^4}.$$

Plugging $K$ and $\eta$ into (72) completes the proof.

### E.9   Proof of Theorem 9

In this section we generalize the convergence analysis of VC SGLDLP-F in Zhang et al. (2022) to non-log-concave target distribution. The proof is similar to the proof of Theorem 8, but the variance corrected-quantization function The variance-corrected quantization technique establishes a scalable bound for the discrepancy between quantized and full-precision values, contingent on the learning rate. This enables variance-corrected quantization advantage over simple stochastic rounding.

Recall that the update of VC SGLDLP-L is

$$\mathbf{x}_{k+1} = Q^{vc}\left(\mathbf{x}_k - \eta Q_G(\widetilde{\nabla U}(\mathbf{x}_k)), 2\eta, \Delta\right)$$
$$= \mathbf{x}_k - \eta Q_G(\widetilde{\nabla U}(\mathbf{x}_k)) + \sqrt{2\eta}\xi_k + \alpha_k,$$

where $\alpha_k$ is defined as

$$\alpha_k = Q^{vc}\left(\mathbf{x}_k - \eta Q_G(\widetilde{\nabla U}(\mathbf{x}_k)), 2\eta, \Delta\right) - \mathbf{x}_k - \eta Q_G(\widetilde{\nabla U}(\mathbf{x}_k)) + \sqrt{2\eta}\xi_k.$$

From analysis in Zhang et al. (2022), we know that

$$\mathbb{E}\left[\|\alpha_k\|^2\right] \leq \max\left(2\Delta\eta G, 5\eta d\right)$$
$$=: \eta A.$$

Combining the analysis in section E.8, we can show,

$$
\begin{aligned}
D_{KL}(p_K\|\hat{p}_{K\eta}) \leq& \frac{M\overline{\mathcal{E}}}{4}K\eta^2 + \frac{3M+1}{4}\sigma^2 K\eta + \frac{((6+3m_2)M + m_2)d}{16m_2}\Delta^2 K\eta + \left(\frac{6M\eta}{4m_2} + \frac{1}{4\eta}\right)K\mathbb{E}\left[\|\alpha_k\|^2\right] \\
\leq& \frac{M\overline{\mathcal{E}}}{4}K\eta^2 + \frac{3M+1}{4}\sigma^2 K\eta + \frac{((6+3m_2)M + m_2)d}{16m_2}\Delta^2 K\eta + \left(\frac{6M\eta}{4m_2} + \frac{1}{4\eta}\right)K\eta A \\
\leq& \frac{M\overline{\mathcal{E}}}{4}K\eta^2 + \frac{3M+1}{4}\sigma^2 K\eta + \frac{((6+3m_2)M + m_2)d}{16m_2}\Delta^2 K\eta + \frac{6M+m_2}{m_2}KA \\
=:& C_0 K\eta^2 + C_1 K\eta\sigma^2 + C_2 K\eta\Delta^2 + C_3 KA,
\end{aligned}
$$

where the constant $C_0$, $C_1$, $C_2$ and $C_3$ are defined as:

$$
\begin{aligned}
C_0 &= \frac{M\overline{\mathcal{E}}}{4} \\
C_1 &= \frac{3M+1}{4} \\
C_2 &= \frac{((6+3m_2)M + m_2)d}{16m_2} \\
C_3 &= \frac{6M+m_2}{m_2}
\end{aligned}
$$

We are ready to bound the Wasserstein distance,

$$
\begin{aligned}
\mathcal{W}_2^2(p_K, \hat{p}_{K\eta}) \leq{}& (12 + 8\left(\kappa_0 + 2b + 2d\right))\left[\left(\left(C_0 + \sqrt{C_0}\right)\eta + \left(C_1 + \sqrt{C_1}\right)\widetilde{A}\right)(K\eta)^2 + \left(C_2 + \sqrt{C_2}\right)\Delta(K\eta)^2\right. \\
&\left. + \left(C_3 + \sqrt{C_3}\right)\mathcal{A}K^2\eta\right] \\
=:{}& \left(\widetilde{C_0}^2\eta + \widetilde{C_1}^2\widetilde{A} + \widetilde{C_2}^2\Delta\right)(K\eta)^2 + \widetilde{C_3}^2\mathcal{A}K^2\eta,
\end{aligned}
$$

where the constants are defined as:

$$
\begin{aligned}
\widetilde{A} &= \max\left\{\sigma^2, \sqrt{\sigma^2}\right\} \\
\mathcal{A} &= \max\left\{A, \sqrt{A}\right\} \\
\widetilde{C_0}^2 &= (12 + 8\left(\kappa_0 + 2b + 2d\right))\left(C_0 + \sqrt{C_0}\right) \\
\widetilde{C_1}^2 &= (12 + 8\left(\kappa_0 + 2b + 2d\right))\left(C_1 + \sqrt{C_1}\right) \\
\widetilde{C_2}^2 &= (12 + 8\left(\kappa_0 + 2b + 2d\right))\left(C_2 + \sqrt{C_2}\right) \\
\widetilde{C_3}^2 &= (12 + 8\left(\kappa_0 + 2b + 2d\right))\left(C_3 + \sqrt{C_3}\right).
\end{aligned}
$$

From Proposition 9 in the paper Raginsky et al. (2017), we know that

$$
\begin{aligned}
\mathcal{W}_2(\hat{p}_{K\eta}, p^*) &\leq \sqrt{2C_{LS}\left(\log\|p_0\|_\infty + \frac{d}{2}\log\frac{3\pi}{m} + \left(\frac{M\kappa_0}{3} + B\sqrt{\kappa_0} + G_0 + \frac{b}{2}\log 3\right)\right)}e^{-K\eta/C_{LS}} \\
&=: \widetilde{C_4}e^{-K\eta/C_{LS}}
\end{aligned}
$$

Finally, we can have

$$
\mathcal{W}_2(p_K, p^*) \leq \left(\widetilde{C_0}\sqrt{\eta} + \widetilde{C_1}\sqrt{A} + \widetilde{C_2}\sqrt{\Delta}\right)K\eta + \widetilde{C_3}\sqrt{\mathcal{A}}\sqrt{K^2\eta} + \widetilde{C_4}e^{-K\eta/C_{LS}}. \tag{74}
$$

Too bound the 2-Wasserstein distance, we need to set

$$
\widetilde{C_0}K\eta^{5/4} = \frac{\epsilon}{2} \quad \text{and} \quad \widetilde{C_3}e^{-K\eta/C_{LS}} = \frac{\epsilon}{2}. \tag{75}
$$

Solving the (75), we can have

$$
K\eta = C_{LS}\log\left(\frac{2\widetilde{C_3}}{\epsilon}\right) \quad \text{and} \quad \eta = \frac{\epsilon^4}{16\widetilde{C_0}^4(K\eta)^4}.
$$

Combining these two we can have

$$
\eta = \frac{\epsilon^4}{16\widetilde{C_0}^4 C_{LS}^4\log^4\left(\frac{2\widetilde{C_3}}{\epsilon}\right)} \quad \text{and} \quad K = \frac{16\widetilde{C_0}^4 C_{LS}^5\log^5\left(\frac{2\widetilde{C_3}}{\epsilon}\right)}{\epsilon^4}.
$$

Plugging $K$ and $\eta$ into (74) completes the proof.

## F   Techinical Proofs

### F.1   Proof of Lemma 12

*Proof.* By the definition of $\xi$ in (E.4)

$$
\begin{aligned}
\|\mathbb{E}\xi\|^2 &= \|\mathbb{E}\tilde{g}(\mathbf{x}) - \mathbb{E}\nabla U(\mathbf{x})\|^2 \\
&= \|\mathbb{E}\nabla U(Q_w(\mathbf{x})) - \mathbb{E}\nabla U(\mathbf{x})\|^2 \\
&\leq \mathbb{E}\left[\|\nabla U(Q_w(\mathbf{x})) - \nabla U(\mathbf{x})\|^2\right] \\
&\leq M^2\mathbb{E}\left[\|Q_w(\mathbf{x}) - \nabla U(\mathbf{x})\|^2\right] \\
&\leq M\frac{\Delta^2 d}{4}.
\end{aligned}
$$

We also know that from the definition that

$$
\begin{aligned}
\mathbb{E}\|\xi\|^2 &= \mathbb{E}\|\tilde{g}(\mathbf{x}) - \nabla U(\mathbf{x})\|^2 \\
&= \mathbb{E}\left\|Q_G(\nabla\tilde{U}(Q_W(\mathbf{x}))) - \nabla\tilde{U}(Q_W(\mathbf{x})) + \nabla\tilde{U}(Q_W(\mathbf{x})) - \nabla U(Q_W(\mathbf{x})) + \nabla U(Q_W(\mathbf{x})) - \nabla U(\mathbf{x})\right\|^2 \\
&= \mathbb{E}\left\|Q_G(\nabla\tilde{U}(Q_W(\mathbf{x}))) - \nabla\tilde{U}(Q_W(\mathbf{x}))\right\|^2 + \mathbb{E}\left\|\nabla\tilde{U}(Q_W(\mathbf{x})) - \nabla U(Q_W(\mathbf{x}))\right\|^2 + \mathbb{E}\|\nabla U(Q_W(\mathbf{x})) - \nabla U(\mathbf{x})\|^2 \\
&\leq \frac{\Delta^2 d}{4} + \sigma^2 + M^2\mathbb{E}\|Q_W(\mathbf{x}) - \mathbf{x}\|^2 \\
&\leq (M^2 + 1)\frac{\Delta^2 d}{4} + \sigma^2,
\end{aligned}
$$

where in the first inequality, we apply Assumptions 1 and 3.

$\square$

### F.2   Proof of Lemma 13

*Proof.* Let $\Gamma_1$ be the set of all couplings between $\widetilde{\Phi}_\eta q_0$ and $q^*$ and $\Gamma_2$ be the set of all couplings between $\widehat{\Phi}_\eta q_0$ adn $q^*$. Let $r_1$ be the optimal coupling between $\widetilde{\Phi}_\eta q_0$ and $q^*$, i.e.

$$
\mathbb{E}_{(\theta,\phi)\sim r_1}[\|\theta - \phi\|^2] = \mathcal{W}_2^2(\widetilde{\Phi}_\eta q_0, q^*).
$$

Let $\left(\begin{bmatrix}\tilde{x} \\ \tilde{\omega}\end{bmatrix}, \begin{bmatrix}x^* \\ \omega^*\end{bmatrix}\right) \sim r_1$. We define the random variable $\begin{bmatrix}x \\ \omega\end{bmatrix}$ as

$$
\begin{bmatrix}x \\ \omega\end{bmatrix} = \begin{bmatrix}\tilde{x} \\ \tilde{\omega}\end{bmatrix} + u\begin{bmatrix}\left(\int_0^\eta\left(\int_0^r e^{-\gamma(s-r)}ds\right)dr\right)\xi \\ \left(\int_0^\eta\left(\int_0^r e^{-\gamma(s-r)}ds\right)dr + \int_0^\eta e^{-\gamma(s-\eta)}ds\right)\xi\end{bmatrix}.
$$

By equation (44), $\left(\begin{bmatrix}x \\ \omega\end{bmatrix}, \begin{bmatrix}x^* \\ \omega^*\end{bmatrix}\right)$ define a valid coupling between $\Phi_\eta q_0$ and $q^*$. Now we can analyze the Wasserstein distance between $\Phi_\eta q_0$ and $q^*$.

$$\mathcal{W}_2^2(\widehat{\Phi}_\eta q_0, q^*) \le \mathbb{E}_{r_1}\left[\left\|\begin{bmatrix}\tilde{x}\\\tilde{\omega}\end{bmatrix} + u\begin{bmatrix}\left(\int_0^\eta\left(\int_0^r e^{-\gamma(s-r)}ds\right)dr\right)\xi\\\left(\int_0^\eta\left(\int_0^r e^{-\gamma(s-r)}ds\right)dr + \int_0^\delta e^{-\gamma(s-\eta)}ds\right)\xi\end{bmatrix} - \begin{bmatrix}x^*\\\omega^*\end{bmatrix}\right\|^2\right] \tag{76}$$

$$\le \mathbb{E}_{r_1}\left[\left\|\begin{bmatrix}\tilde{x}-x^*\\\tilde{\omega}-\omega^*\end{bmatrix} + u\begin{bmatrix}\left(\int_0^\eta\left(\int_0^r e^{-\gamma(s-r)}ds\right)dr\right)\mathbb{E}\xi\\\left(\int_0^\eta\left(\int_0^r e^{-\gamma(s-r)}ds\right)dr + \int_0^\delta e^{-\gamma(s-\eta)}ds\right)\mathbb{E}\xi\end{bmatrix}\right\|^2\right]$$

$$+ \mathbb{E}_{r_1}\left[\left\|u\begin{bmatrix}\left(\int_0^\eta\left(\int_0^r e^{-\gamma(s-r)}ds\right)dr\right)(\xi-\mathbb{E}\xi)\\\left(\int_0^\eta\left(\int_0^r e^{-\gamma(s-r)}ds\right)dr + \int_0^\eta e^{-\gamma(s-\eta)}ds\right)(\xi-\mathbb{E}\xi)\end{bmatrix}\right\|^2\right]$$

$$\le \left(\mathcal{W}_2(\widetilde{\Phi}_\eta q_0, q^*) + 2u\sqrt{\eta^4/4+\eta^2}\,\|\mathbb{E}\xi\|\right)^2 + 4u^2(\eta^4/4+\eta^2)\mathbb{E}_{r_1}\left[\|\xi-\mathbb{E}\xi\|^2\right]$$

$$\le \left(\mathcal{W}_2(\widetilde{\Phi}_\eta q_0, q^*) + \sqrt{5}/2u\eta\sqrt{d}M\Delta\right)^2 + 5u^2\eta^2\left((M^2+1)\frac{\Delta^2 d}{4}+\sigma^2\right). \tag{77}$$

$\square$

## F.3 Proof of Lemma 10

*Proof.* In order to get the upper bound of $\|\mathbf{x}_k\|$ and $\|\mathbf{v}_k\|$, we bound the Lyapunov function $\mathcal{E}(\mathbf{x}_k, \mathbf{v}_k)$. By the smooth Assumption 1, we know

$$U(\mathbf{x}_{k+1}) - U(x^*) \le U(\mathbf{x}_k) + \langle\nabla U(\mathbf{x}_k), \mathbf{x}_{k+1} - \mathbf{x}_k\rangle + M^2/2\,\|\mathbf{x}_{k+1}-\mathbf{x}_k\|^2 - U(x^*).$$

Recall the definition of the Lyapunov function

$$\mathcal{E}(\mathbf{x}_{k+1}, \mathbf{v}_{k+1}) = \|\mathbf{x}_{k+1}\|^2 + \|\mathbf{x}_{k+1} + 2\mathbf{v}_{k+1}/\gamma\|^2 + 8u\left(U(\mathbf{x}_{k+1}) - U(x^*)\right)/\gamma^2.$$

For the first two terms we have

$$\|\mathbf{x}_{k+1}\|^2 = \|\mathbf{x}_k\|^2 + 2\langle\mathbf{x}_k, \mathbf{x}_{k+1}-\mathbf{x}_k\rangle + \|\mathbf{x}_{k+1}-\mathbf{x}_k\|^2$$

$$\|\mathbf{x}_{k+1} + 2\mathbf{v}_{k+1}/\gamma\|^2 = \|\mathbf{x}_k + 2\mathbf{v}_k/\gamma\|^2 + 2\langle\mathbf{x}_k + 2\mathbf{v}_k/\gamma, \mathbf{x}_{k+1} - \mathbf{x}_k + 2(\mathbf{v}_{k+1}-\mathbf{v}_k)/\gamma\rangle$$

$$+ \|\mathbf{x}_{k+1} - \mathbf{x}_k + 2(\mathbf{v}_{k+1}-\mathbf{v}_k)/\gamma\|^2.$$

This implies the following:

$$\mathbb{E}\left[\mathcal{E}(\mathbf{x}_{k+1}, \mathbf{v}_{k+1})\right] \le \mathbb{E}\left[\mathcal{E}(\mathbf{x}_k, \mathbf{v}_k)\right] + 4\mathbb{E}\left[\langle\mathbf{x}_k, \mathbf{x}_{k+1}-\mathbf{x}_k\rangle\right] + \frac{4}{\gamma}\mathbb{E}\left[\langle\mathbf{x}_k, \mathbf{v}_{k+1}-\mathbf{v}_k\rangle\right] + \frac{4}{\gamma}\mathbb{E}\left(\langle\mathbf{v}_k, \mathbf{x}_{k+1}-\mathbf{x}_k\rangle\right)$$

$$\tag{78}$$

$$+ \frac{8}{\gamma^2}\mathbb{E}\left[\langle\mathbf{v}_k, \mathbf{v}_{k+1}-\mathbf{v}_k\rangle\right] + \frac{8u}{\gamma^2}\mathbb{E}\left[\langle\nabla U(\mathbf{x}_k), \mathbf{x}_{k+1}-\mathbf{x}_k\rangle + M/2\,\|\mathbf{x}_{k+1}-\mathbf{x}_k\|^2\right]$$

$$+ \mathbb{E}\left[\|\mathbf{x}_{k+1}-\mathbf{x}_k\|^2\right] + \mathbb{E}\left[\|\mathbf{x}_{k+1} - \mathbf{x}_k + 2(\mathbf{v}_{k+1}-\mathbf{v}_k)/\gamma\|^2\right].$$

By the update rule in (5), we know that

$$\mathbb{E}\left[\langle\mathbf{x}_k, \mathbf{x}_{k+1}-\mathbf{x}_k\rangle\right] = \frac{1-e^{-\gamma\eta}}{\gamma}\mathbb{E}\left[\langle\mathbf{x}_k, \mathbf{v}_k\rangle\right] + \frac{u(\gamma\eta+e^{-\gamma\eta}-1)}{\gamma^2}\mathbb{E}\left[\langle\mathbf{x}_k, \tilde{g}(\mathbf{x}_k)\rangle\right],$$

$$\mathbb{E}\left[\langle\mathbf{x}_k, \mathbf{v}_{k+1}-\mathbf{v}_k\rangle\right] = -(1-e^{-\gamma\eta})\mathbb{E}\left[\langle\mathbf{x}_k, \mathbf{v}_k\rangle\right] - \frac{u(1-e^{-\gamma\eta})}{\gamma}\mathbb{E}\left[\langle\mathbf{x}_k, \tilde{g}(\mathbf{x}_k)\rangle\right],$$

$$\mathbb{E}\left[\langle\mathbf{v}_k, \mathbf{x}_{k+1}-\mathbf{x}_k\rangle\right] = \frac{1-e^{-\gamma\eta}}{\gamma}\mathbb{E}\left[\|\mathbf{v}_k\|^2\right] + \frac{u(\gamma\eta+e^{-\gamma\eta}-1)}{\gamma^2}\mathbb{E}\left[\langle\mathbf{v}_k, \tilde{g}(\mathbf{x}_k)\rangle\right],$$

$$\mathbb{E}\left[\langle\mathbf{v}_k, \mathbf{v}_{k+1}-\mathbf{v}_k\rangle\right] = -(1-e^{-\gamma\eta})\mathbb{E}\left[\|\mathbf{v}_k\|^2\right] - \frac{u(1-e^{-\gamma\eta})}{\gamma}\mathbb{E}\left[\langle\mathbf{v}_k, \tilde{g}(\mathbf{x}_k)\rangle\right].$$

Plug into the (78) yields:

$$\mathbb{E}\left[\mathcal{E}(\mathbf{x}_{k+1}, \mathbf{v}_{k+1})\right] \leq \mathbb{E}\left[\mathcal{E}(\mathbf{x}_k, \mathbf{v}_k)\right] - \frac{4u(2 - \gamma\eta - 2e^{-\gamma\eta})}{\gamma^2}\mathbb{E}\left[\langle\mathbf{x}_k, \tilde{g}(\mathbf{x}_k)\rangle\right] - \frac{4(1 - e^{-\gamma\eta})}{\gamma^2}\mathbb{E}\left[\|\mathbf{v}_k\|^2\right]$$

$$+ \frac{4u(\gamma\eta + e^{-\gamma\eta} - 1)}{\gamma^3}\mathbb{E}\left[\langle\mathbf{v}_k, \tilde{g}(\mathbf{x}_k)\rangle\right] + \frac{8u(1 - e^{-\gamma\eta})}{\gamma^3}\mathbb{E}\left[\langle\mathbf{v}_k, \nabla U(\mathbf{x}_k) - \tilde{g}(\mathbf{x}_k)\rangle\right]$$

$$+ \frac{8u^2(\gamma\eta + e^{-\gamma\eta} - 1)}{\gamma^4}\mathbb{E}\left[\langle\nabla U(\mathbf{x}_k), \tilde{g}(\mathbf{x}_k)\rangle\right] + \left(\frac{4Mu}{\gamma^2} + 3\right)\mathbb{E}\left[\|\mathbf{x}_{k+1} - \mathbf{x}_k\|^2\right]$$

$$+ \frac{8}{\gamma^2}\mathbb{E}\left[\|\mathbf{v}_{k+1} - \mathbf{v}_k\|^2\right]. \tag{79}$$

By Assumption 2, we know that $\langle\mathbf{x}_k, \nabla U(\mathbf{x}_k)\rangle \geq m_2 \|\mathbf{x}_k\|^2 - b$. We then assume $\eta \leq 1/(8\gamma)$ and use the inequality $-x \leq e^{-x} - 1 \leq x^2/2 - x$ for any $x \geq 0$, it follows that

$$-\frac{4u(2 - \gamma\eta - 2e^{-\gamma\eta})}{\gamma^2}\mathbb{E}\left[\langle\mathbf{x}_k, \tilde{g}(\mathbf{x}_k)\rangle\right]$$

$$= -\frac{4u(2 - \gamma\eta - 2e^{-\gamma\eta})}{\gamma^2}\left(\mathbb{E}\left[\langle\mathbf{x}_k, \nabla U(\mathbf{x}_k)\rangle\right] + \mathbb{E}\left[\langle\mathbf{x}_k, \tilde{g}(\mathbf{x}_k) - \nabla U(\mathbf{x}_k)\rangle\right]\right)$$

$$\leq -\frac{4u(2 - \gamma\eta - 2e^{-\gamma\eta})}{\gamma^2}\left(m_2\mathbb{E}\left[\|\mathbf{x}_k\|^2\right] - b\right) + \frac{4u(2 - \gamma\eta - 2e^{-\gamma\eta})}{\gamma^2}\left(\frac{1}{8}\mathbb{E}\left[\|\mathbf{x}_k\|^2\right] + 2\mathbb{E}\left[\|\tilde{g}(\mathbf{x}_k) - \nabla U(\mathbf{x}_k)\|^2\right]\right)$$

$$\leq -\frac{3m_2u\eta}{\gamma}\mathbb{E}\left[\|\mathbf{x}_k\|^2\right] + \frac{4u\eta b}{\gamma} + \frac{8u\eta}{\gamma}\mathbb{E}\left[\|\tilde{g}(\mathbf{x}_k) - \nabla U(\mathbf{x}_k)\|^2\right],$$

where the first inequality is because of the Young's inequaltiy and Assumption 1 and the last inequality is based on the inequality that $\gamma\eta - (\gamma\eta)^2 \leq 2 - \gamma\eta - 2e^{-\gamma\eta} \leq \gamma\eta$. Again by Young's inequality and the update rule in (5) we have:

$$\mathbb{E}\left[\|\mathbf{x}_{k+1} - \mathbf{x}_k\|^2\right] \leq 2\eta^2\mathbb{E}\left[\|\mathbf{v}_k\|^2\right] + u^2\eta^4/2\mathbb{E}\left[\|\tilde{g}(\mathbf{x}_k)\|^2\right] + \mathbb{E}\left[\|\xi_k^x\|^2\right]$$

$$\mathbb{E}\left[\|\mathbf{v}_{k+1} - \mathbf{v}_k\|^2\right] \leq 2\gamma^2\eta^2\mathbb{E}\left[\|\mathbf{v}_k\|^2\right] + 2u^2\eta^2\mathbb{E}\left[\|\tilde{g}(\mathbf{x}_k)\|^2\right] + \mathbb{E}\left[\|\xi_k^v\|^2\right].$$

It is easy to verify the fact that $\mathbb{E}\left[\|\xi_k^v\|^2\right] \leq 2\gamma ud\eta$ and $\mathbb{E}\left[\|\xi_k^x\|^2\right] \leq 2ud\eta^2$. Thus,

$$\mathbb{E}\left[\mathcal{E}(\mathbf{x}_{k+1}, \mathbf{v}_{k+1})\right]$$

$$\leq \mathbb{E}\left[\mathcal{E}(\mathbf{x}_k, \mathbf{v}_k)\right] - \frac{3um\eta^2}{\gamma}\mathbb{E}\left[\|\mathbf{x}_k\|^2\right] - \frac{3(1 - e^{-\gamma\eta}) - \eta^2(8Mu + u\gamma + 22\gamma^2)}{\gamma^2}\mathbb{E}\left[\|\mathbf{v}_k\|^2\right]$$

$$+ \frac{36u^2\eta^2 + 2\gamma u\eta^2 + \left(4Mu + 3\gamma^2\right)\eta^4}{2\gamma^2}\mathbb{E}\left[\|\tilde{g}(\mathbf{x}_k)\|^2\right] + \frac{2u^2\eta^2}{\gamma^2}\mathbb{E}\left[\|\nabla U(\mathbf{x}_k)\|^2\right]$$

$$+ \frac{8u\eta(\gamma^2 + 2u)}{\gamma^3}\mathbb{E}\left[\|\nabla U(\mathbf{x}_k) - \tilde{g}(\mathbf{x}_k)\|^2\right] + \frac{(8Mu + 6\gamma^2)ud\eta^2 + 4(4d + b)u\gamma\eta}{\eta^2}.$$

If we set

$$\eta \leq \min\left\{\frac{\gamma}{4\left(8Mu + u\gamma + 22\gamma^2\right)}, \sqrt{\frac{4u^2}{4Mu + 3\gamma^2}}, \frac{6\gamma bu}{\left(4Mu + 3\gamma^2\right)d}\right\},$$

we can obtain the following,

$$\mathbb{E}\left[\mathcal{E}(\mathbf{x}_{k+1}, \mathbf{v}_{k+1})\right] \leq \mathbb{E}\left[\mathcal{E}(\mathbf{x}_k, \mathbf{v}_k)\right] - \frac{3um_2\eta}{\gamma}\mathbb{E}\left[\|\mathbf{x}_k\|^2\right] - \frac{2\eta}{\gamma}\mathbb{E}\left[\|\mathbf{v}_k\|^2\right] + \frac{(20u + \gamma)u\eta^2}{\gamma^2}\mathbb{E}\left[\|\tilde{g}(\mathbf{x}_k)\|^2\right]$$

$$+ \frac{2u^2\eta^2}{\gamma^2}\mathbb{E}\left[\|\nabla U(\mathbf{x}_k)\|^2\right] + \frac{8u\eta\left(\gamma^2 + 2u\right)}{\gamma^3}\mathbb{E}\left[\|\nabla U(\mathbf{x}_k) - \tilde{g}(\mathbf{x}_k)\|^2\right] + \frac{16(d + b)u\eta}{\gamma}. \tag{80}$$

Furthermore we can bound $\mathbb{E}\left[\|\tilde{g}(\mathbf{x}_k)\|^2\right]$ by the following analysis:

$$
\begin{aligned}
\mathbb{E}\left[\|\tilde{g}(\mathbf{x}_k)\|^2\right] &\leq 2\mathbb{E}\left[\|\tilde{g}(\mathbf{x}_k) - \nabla U(\mathbf{x}_k)\|^2\right] + 2\mathbb{E}\left[\|\nabla U(\mathbf{x}_k)\|^2\right] \\
&\leq 2\left((M^2+1)\frac{\Delta^2 d}{4} + \sigma^2\right) + 4M^2\mathbb{E}\left[\|\mathbf{x}_k\|^2\right] + 4G^2,
\end{aligned}
\tag{81}
$$

where $G^2$ is the bound of the gradient at 0, i.e. $\|\nabla U(0)\|^2 \leq G^2$. Thus we can have:

$$
\mathbb{E}\left[\mathcal{E}(\mathbf{x}_{k+1}, \mathbf{v}_{k+1})\right] \leq \mathbb{E}\left[\mathcal{E}(\mathbf{x}_k, \mathbf{v}_k)\right] - \frac{3um_2\eta}{\gamma}\mathbb{E}\left[\|\mathbf{x}_k\|^2\right] - \frac{2\eta}{\gamma}\mathbb{E}\left[\|\mathbf{v}_k\|^2\right] + \frac{(21u+\gamma)4M^2u\eta^2}{\gamma^2}\mathbb{E}\left[\|\mathbf{x}_k\|^2\right]
\tag{82}
$$

$$
+ \left(\frac{2(20u+\gamma)u\eta^2}{\gamma^2} + \frac{8u\eta\left(\gamma^2+2u\right)}{\gamma^3}\right)\left((M^2+1)\frac{\Delta^2 d}{4} + \sigma^2\right)
\tag{83}
$$

$$
+ \frac{(21u+\gamma)4u\eta^2}{\gamma^2}G^2 + \frac{16(d+b)u\eta}{\gamma}.
\tag{84}
$$

If we set the stepsize

$$
\eta \leq \min\left\{\frac{\gamma m_2}{12(21u+\gamma)M^2}, \frac{8(\gamma^2+2u)}{(20u+\gamma)\gamma}\right\},
$$

then we have:

$$
\begin{aligned}
\mathbb{E}\left[\mathcal{E}(\mathbf{x}_{k+1}, \mathbf{v}_{k+1})\right] \leq {}& \mathbb{E}\left[\mathcal{E}(\mathbf{x}_k, \mathbf{v}_k)\right] - \frac{8um_2\eta}{3\gamma}\mathbb{E}\left[\|\mathbf{x}_k\|^2\right] - \frac{2\eta}{\gamma}\mathbb{E}\left[\|\mathbf{v}_k\|^2\right] \\
& + \left(\frac{16u\eta\left(\gamma^2+2u\right)}{\gamma^3}\right)\left((M^2+1)\frac{\Delta^2 d}{4} + \sigma^2\right) \\
& + \frac{(21u+\gamma)4u\eta^2}{\gamma^2}G^2 + \frac{16(d+b)u\eta}{\gamma}.
\end{aligned}
$$

Furthermore by Young's inequality and Assumption 1, we can bound the Lyapunov function by the following:

$$
\mathcal{E}(x,v) \leq 5/2\|x\|^2 + \frac{12}{\gamma^2} + \frac{2uM}{\gamma^2}\left(3\|x\|^2 + 6\|x^*\|^2\right).
$$

Then if $\gamma^2 \leq 4Mu$, we have

$$
\mathcal{E}(x,v) \leq \frac{16uM}{\gamma^2}\|x\|^2 + \frac{12}{\gamma^2}\|v\|^2 + \frac{12uM}{\gamma^2}\|x^*\|^2.
\tag{85}
$$

Thus,

$$
\begin{aligned}
\mathbb{E}\left[\mathcal{E}(\mathbf{x}_{k+1}, \mathbf{v}_{k+1})\right] \leq {}& \left(1 - \frac{\gamma m_2\eta}{6M}\right)\mathbb{E}\left[\mathcal{E}(\mathbf{x}_k, \mathbf{v}_k)\right] + \left(\frac{16u\eta\left(\gamma^2+2u\right)}{\gamma^3}\right)\left((M^2+1)\frac{\Delta^2 d}{4} + \sigma^2\right) \\
& + \frac{(21u+\gamma)4u\eta^2}{\gamma^2}G^2 + \frac{16(d+b)u\eta}{\gamma}.
\end{aligned}
$$

Finally we show that

$$
\begin{aligned}
\sup_{k\geq 0}\mathbb{E}\left[\mathcal{E}(\mathbf{x}_k, \mathbf{v}_k)\right] \leq {}& \mathbb{E}\left[\mathcal{E}(x_0, v_0)\right] + \frac{6M}{\gamma m_2\eta}\left(\frac{16u\eta\left(\gamma^2+2u\right)}{\gamma^3}\right)\left((M^2+1)\frac{\Delta^2 d}{4} + \sigma^2\right) \\
& + \frac{6M}{\gamma m_2\eta}\frac{(21u+\gamma)4u\eta^2}{\gamma^2}G^2 + \frac{6M}{\gamma m_2\eta}\frac{16(d+b)u\eta}{\gamma} \\
\leq {}& \mathbb{E}\left[\mathcal{E}(x_0, v_0)\right] + \frac{96u\left(\gamma^2+2u\right)}{m_2\gamma^4}\left((M^2+1)\frac{\Delta^2 d}{4} + \sigma^2\right) + \frac{24(21u+\gamma)uM}{m_2\gamma^3}G^2 + \frac{96(d+b)uM}{m_2\gamma^2} \\
\leq {}& \overline{\mathcal{E}} + C_0\left((M^2+1)\frac{\Delta^2 d}{4} + \sigma^2\right),
\end{aligned}
\tag{86}
$$

where $\overline{\mathcal{E}} = \mathbb{E}\left[\mathcal{E}(x_0, v_0)\right] + \frac{24(21u+\gamma)uM}{m_2\gamma^3}G^2 + \frac{96(d+b)uM}{m_2\gamma^2}$ and $C_0 = \frac{96u(\gamma^2+2u)}{m_2\gamma^4}$. Moreover by the definition of Laypunov function, we know $\mathcal{E}(x, v) \geq \max\{\|x\|^2, 2\|v/\gamma\|^2\}$. This further implies that

$$\mathbb{E}\left[\|\mathbf{x}_k\|^2\right] \leq \overline{\mathcal{E}} + C_0\left((M^2 + 1)\frac{\Delta^2 d}{4} + \sigma^2\right)$$

$$\mathbb{E}\left[\|\mathbf{v}_k\|^2\right] \leq \gamma^2\overline{\mathcal{E}}/2 + \gamma^2 C_0/2\left((M^2 + 1)\frac{\Delta^2 d}{4} + \sigma^2\right).$$

Combining with equation (81) we can bound $\mathbb{E}\left[\|\tilde{g}(\mathbf{x}_k)\|^2\right]$ as:

$$\mathbb{E}\left[\|\tilde{g}(\mathbf{x}_k)\|^2\right] \leq 2\left((M^2 + 1)\frac{\Delta^2 d}{4} + \sigma^2\right) + 4M^2\overline{\mathcal{E}} + 4G^2. \tag{87}$$

$\square$

### F.4  Proof of Lemma 15

*Proof.* By the update rule in (1), we have:

$$\begin{aligned}
\mathbb{E}\left[\|\mathbf{x}_{k+1}\|^2\right] =& \mathbb{E}\left[\|\mathbf{x}_k - \eta\tilde{g}(\mathbf{x}_k)\|^2\right] + \sqrt{8\eta}\mathbb{E}\left[\langle\mathbf{x}_k - \eta\tilde{g}(\mathbf{x}_k), \xi_{k+1}\rangle\right] + 2\eta\mathbb{E}\left[\|\xi_{k+1}\|^2\right] \\
=& \mathbb{E}\left[\|\mathbf{x}_k - \eta\tilde{g}(\mathbf{x}_k)\|^2\right] + 2\eta d \\
=& \mathbb{E}\left[\|\mathbf{x}_k - \eta\nabla U(\mathbf{x}_k) - \eta\left(\tilde{g}(\mathbf{x}_k) - \nabla U(Q_W(\mathbf{x}_k))\right) - \eta\left(\nabla U(Q_W(\mathbf{x}_k)) - \nabla U(\mathbf{x}_k)\right)\|^2\right] + 2\eta d \\
=& \mathbb{E}\left[\|\mathbf{x}_k - \eta\nabla U(\mathbf{x}_k) - \eta\left(\nabla U(Q_W(\mathbf{x}_k)) - \nabla U(\mathbf{x}_k)\right)\|^2\right] + \eta^2\mathbb{E}\left[\|\tilde{g}(\mathbf{x}_k) - \nabla U(Q_W(\mathbf{x}_k))\|^2\right] + 2\eta d \\
=& \left(\mathbb{E}\left[\|\mathbf{x}_k - \eta\nabla U(\mathbf{x}_k)\|\right] + \eta\mathbb{E}\left[\|\nabla U(Q_W(\mathbf{x}_k)) - \nabla U(\mathbf{x}_k)\|\right]\right)^2 + \eta^2\left(\frac{\Delta^2 d}{4} + \sigma^2\right) + 2\eta d.
\end{aligned}$$

We know the fact that:

$$\begin{aligned}
\mathbb{E}\left[\|\mathbf{x}_k - \eta\nabla U(\mathbf{x}_k)\|^2\right] &= \mathbb{E}\left[\|\mathbf{x}_k\|^2\right] - 2\eta\mathbb{E}\left[\langle\mathbf{x}_k, \nabla U(\mathbf{x}_k)\rangle\right] + \eta^2\mathbb{E}\left[\|\nabla U(\mathbf{x}_k)\|^2\right] \\
&= \mathbb{E}\left[\|\mathbf{x}_k\|^2\right] + 2\eta\left(b - m_2\mathbb{E}\left[\|\mathbf{x}_k\|^2\right]\right) + 2\eta^2\left(M^2\mathbb{E}\left[\|\mathbf{x}_k\|^2\right] + G^2\right) \\
&= \left(1 - 2\eta m_2 + 2\eta^2 M^2\right)\mathbb{E}\left[\|\mathbf{x}_k\|^2\right] + 2\eta b + 2\eta^2 G^2.
\end{aligned}$$

For any $\eta \in \left(0, 1 \wedge \frac{m_2}{2M^2}\right)$, if $0 < 1 - 2\eta m_2 + 2\eta^2 M^2 < 1$ and set $c = \frac{\eta m_2 - \eta^2 M^2}{1 - 2\eta m + 2\eta^2 M^2}$, then we have:

$$\mathbb{E}\left[\|\mathbf{x}_{k+1}\|^2\right] \leq (1 + c)\mathbb{E}\left[\|\mathbf{x}_k - \eta\nabla U(\mathbf{x}_k)\|^2\right] + \left(1 + \frac{1}{c}\right)\eta^2\mathbb{E}\left[\|\nabla U(Q_W(\mathbf{x}_k)) - \nabla U(\mathbf{x}_k)\|^2\right] + \eta^2\left(\frac{\Delta^2 d}{4} + \sigma^2\right) + 2\eta d \tag{88}$$

$$\leq \left(1 - \eta m_2 + \eta^2 M^2\right)\mathbb{E}\left[\|\mathbf{x}_k\|^2\right] + \frac{1 - \eta m_2 + \eta^2 M}{\eta m_2 - \eta^2 M}\frac{M^2\eta^2\Delta^2 d}{4} + \frac{1 - \eta m_2 + \eta^2 M}{1 - 2\eta m_2 + 2\eta^2 M^2}\left(2\eta b + 2\eta^2 G^2\right) \tag{89}$$

$$+ \eta^2\left(\frac{\Delta^2 d}{4} + \sigma^2\right) + 2\eta d. \tag{90}$$

For any $k > 0$ we can bound the recursive equations as:

$$
\begin{aligned}
\mathbb{E}\left[\|\mathbf{x}_k\|^2\right] &\leq \mathbb{E}\left[\|x_0\|^2\right] + \frac{1 - \eta m_2 + \eta^2 M^2}{\eta^2 (m_2 - \eta M^2)^2} \frac{M^2 \eta^2 \Delta^2 d}{4} + \frac{1 - \eta m_2 + \eta^2 M^2}{\eta(1 - 2\eta m_2 + 2\eta^2 M^2)(m_2 - \eta M^2)} \left(2\eta b + 2\eta^2 G^2\right) \\
&\quad + \frac{1}{\eta(m_2 - \eta M)} \left(\eta^2 \frac{\Delta^2 d}{4} + 2\eta d\right) \\
&= \mathbb{E}\left[\|x_0\|^2\right] + \frac{1 - \eta m_2 + \eta^2 M^2}{(m_2 - \eta M^2)^2} \frac{M^2 \Delta^2 d}{4} + \frac{1 - \eta m_2 + \eta^2 M^2}{(1 - 2\eta m_2 + 2\eta^2 M^2)(m_2 - \eta M^2)} \left(2b + 2\eta G^2\right) \\
&\quad + \frac{1}{m_2 - \eta M^2} \left(\eta \frac{\Delta^2 d}{4} + \eta \sigma^2 + 2d\right) \\
&\leq \mathbb{E}\left[\|x_0\|^2\right] + \frac{2M^2}{m_2} \frac{\Delta^2 d}{4} + \frac{2}{m_2}\left(2b + 2\eta G^2\right) + \frac{2}{m_2}\left(\eta \frac{\Delta^2 d}{4} + \eta \sigma^2 + 2d\right).
\end{aligned}
$$

Now if we let $\mathcal{E} = \mathbb{E}\left[\|x_0\|^2\right] + \frac{M}{m_2}\left(2b + 2\eta G^2 + 2d\right)$, then we can write:

$$
\mathbb{E}\left[\|\mathbf{x}_k\|^2\right] \leq \mathcal{E} + \frac{2\left(M^2 + 1\right)}{m_2} \frac{\Delta^2 d}{4} + \frac{2\sigma^2}{m_2}.
$$

$\square$

## F.5  Proof of Lemma 11

*Proof.* From the same analysis in (80), if we set

$$
\eta \leq \min\left\{\frac{\gamma}{4\left(8Mu + u\gamma + 22\gamma^2\right)}, \sqrt{\frac{4u^2}{4Mu + 3\gamma^2}}, \frac{6\gamma bu}{\left(4Mu + 3\gamma^2\right) d}\right\},
$$

we can obtain the following,

$$
\begin{aligned}
\mathbb{E}\left[\mathcal{E}(\mathbf{x}_{k+1}, \mathbf{v}_{k+1})\right] &\leq \mathbb{E}\left[\mathcal{E}(\mathbf{x}_k, \mathbf{v}_k)\right] - \frac{3um_2\eta}{\gamma} \mathbb{E}\left[\|\mathbf{x}_k\|^2\right] - \frac{2\eta}{\gamma} \mathbb{E}\left[\|\mathbf{v}_k\|^2\right] + \frac{(20u + \gamma)u\eta^2}{\gamma^2} \mathbb{E}\left[\left\|Q_G(\nabla \tilde{U}(\mathbf{x}_k))\right\|^2\right] \\
&\quad + \frac{2u^2\eta^2}{\gamma^2} \mathbb{E}\left[\|\nabla U(\mathbf{x}_k)\|^2\right] + \frac{8u\eta\left(\gamma^2 + 2u\right)}{\gamma^3} \mathbb{E}\left[\left\|\nabla U(\mathbf{x}_k) - Q_G(\nabla \tilde{U}(\mathbf{x}_k))\right\|^2\right] + \frac{16(d + b)u\eta}{\gamma}.
\end{aligned}
\tag{91}
$$

By assumption 1, we can bound $\mathbb{E}\left[\left\|Q_G(\nabla \tilde{U}(\mathbf{x}_k))\right\|^2\right]$ by the following,

$$
\begin{aligned}
\mathbb{E}\left[\|Q_G(\nabla U(\mathbf{x}_k))\|^2\right] &= \mathbb{E}\left[\left\|Q_G(\nabla \tilde{U}(\mathbf{x}_k)) - \nabla U(\mathbf{x}_k) + \nabla U(\mathbf{x}_k) - \nabla U(0) + \nabla U(0)\right\|^2\right] \\
&\leq \mathbb{E}\left[\left\|Q_G(\nabla \tilde{U}(\mathbf{x}_k)) - \nabla U(\mathbf{x}_k)\right\|^2\right] + 2\mathbb{E}\left[\|\nabla U(\mathbf{x}_k) - \nabla U(0)\|^2\right] + 2\mathbb{E}\left[\|\nabla U(0)\|^2\right] \\
&\leq \left(\frac{\Delta^2 d}{4} + \sigma^2\right) + 2M^2 \mathbb{E}\left[\|\mathbf{x}_k\|^2\right] + 2G^2.
\end{aligned}
$$

Plugging this bound into equation (91), we can have:

$$
\begin{aligned}
\mathbb{E}\left[\mathcal{E}\left(\mathbf{x}_{k+1}, \mathbf{v}_{k+1}\right)\right] &\leq \mathbb{E}\left[\mathcal{E}(\mathbf{x}_k, \mathbf{v}_k)\right] - \frac{3u m_2 \eta}{\gamma} \mathbb{E}\left[\|\mathbf{x}_k\|^2\right] - \frac{2\eta}{\gamma} \mathbb{E}\left[\|\mathbf{v}_k\|^2\right] + \frac{2(20u+\gamma)u\eta^2 M^2}{\gamma^2} \mathbb{E}\left[\|\mathbf{x}_k\|^2\right] \\
&\quad + \frac{(20u+\gamma)u\eta^2}{\gamma^2}\left(\frac{\Delta^2 d}{4} + \sigma^2 + 2G^2\right) + \frac{2u^2\eta^2}{\gamma^2}\left(2M^2\mathbb{E}\left[\|\mathbf{x}_k\|^2\right] + 2G^2\right) \\
&\quad + \frac{8u\eta\left(\gamma^2+2u\right)}{\gamma^3}\left(\frac{\Delta^2 d}{4} + \sigma^2\right) + \frac{16(d+b)u\eta}{\gamma} \\
&\leq \mathbb{E}\left[\mathcal{E}(\mathbf{x}_k, \mathbf{v}_k)\right] - \frac{3u m_2 \eta}{\gamma} \mathbb{E}\left[\|\mathbf{x}_k\|^2\right] - \frac{2\eta}{\gamma} \mathbb{E}\left[\|\mathbf{v}_k\|^2\right] + \frac{2(22u+\gamma)u\eta^2 M^2}{\gamma^2} \mathbb{E}\left[\|\mathbf{x}_k\|^2\right] \\
&\quad + \frac{(20u+\gamma)\gamma u\eta^2 + 8\left(\gamma^2+2u\right)u\eta}{\gamma^3}\left(\frac{\Delta^2 d}{4} + \sigma^2\right) + \frac{2(22u+\gamma)u\eta^2 M^2}{\gamma^2}G^2 + \frac{16(d+b)u\eta}{\gamma} \\
&\leq \mathbb{E}\left[\mathcal{E}(\mathbf{x}_k, \mathbf{v}_k)\right] - \frac{3u m_2 \eta}{\gamma} \mathbb{E}\left[\|\mathbf{x}_k\|^2\right] - \frac{2\eta}{\gamma} \mathbb{E}\left[\|\mathbf{v}_k\|^2\right] + \frac{2(22u+\gamma)u\eta^2 M^2}{\gamma^2} \mathbb{E}\left[\|\mathbf{x}_k\|^2\right] \\
&\quad + \frac{\left(36u+9\gamma^2\right)u\eta}{\gamma^3}\left(\frac{\Delta^2 d}{4} + \sigma^2\right) + \frac{2(22u+\gamma)u\eta^2 M^2}{\gamma^2}G^2 + \frac{16(d+b)u\eta}{\gamma}.
\end{aligned}
$$

If we set the step size $\eta \leq \frac{\gamma m_2}{6(22u+\gamma)M^2}$, we can have:

$$
\mathbb{E}\left[\mathcal{E}\left(\mathbf{x}_{k+1}, \mathbf{v}_{k+1}\right)\right] \leq \mathbb{E}\left[\mathcal{E}(\mathbf{x}_k, \mathbf{v}_k)\right] - \frac{8u m_2 \eta}{3\gamma} \mathbb{E}\left[\|\mathbf{x}_k\|^2\right] - \frac{2\eta}{\gamma} \mathbb{E}\left[\|\mathbf{v}_k\|^2\right] \tag{92}
$$

$$
+ \frac{\left(36u+9\gamma^2\right)u\eta}{\gamma^3}\left(\frac{\Delta^2 d}{4} + \sigma^2\right) + \frac{2(22u+\gamma)u\eta^2 M^2}{\gamma^2}G^2 + \frac{16(d+b)u\eta}{\gamma}. \tag{93}
$$

Again from the same analysis in (85), if $\gamma^2 \leq 4Mu$, we have

$$
\mathcal{E}(x, v) \leq \frac{16uM}{\gamma^2}\|x\|^2 + \frac{12}{\gamma^2}\|v\|^2 + \frac{12uM}{\gamma^2}\|x^*\|^2.
$$

Thus,

$$
\begin{aligned}
\mathbb{E}\left[\mathcal{E}(\mathbf{x}_{k+1}, \mathbf{v}_{k+1})\right] &\leq \left(1 - \frac{\gamma m_2 \eta}{6M}\right)\mathbb{E}\left[\mathcal{E}(\mathbf{x}_k, \mathbf{v}_k)\right] + \frac{\left(36u+9\gamma^2\right)u\eta}{\gamma^3}\left(\frac{\Delta^2 d}{4} + \sigma^2\right) \\
&\quad + \frac{2(22u+\gamma)u\eta^2 M^2}{\gamma^2}G^2 + \frac{16(d+b)u\eta}{\gamma}.
\end{aligned}
$$

Finally, we show that for any $k > 0$,

$$
\begin{aligned}
\mathbb{E}\left[\mathcal{E}(\mathbf{x}_k, \mathbf{v}_k)\right] &\leq \mathbb{E}\left[\mathcal{E}(x_0, v_0)\right] + \frac{6M}{\gamma m_2 \eta}\frac{\left(36u+9\gamma^2\right)u\eta}{\gamma^3}\left(\frac{\Delta^2 d}{4} + \sigma^2\right) \\
&\quad + \frac{6M}{\gamma m_2 \eta}\frac{2(22u+\gamma)u\eta^2 M^2}{\gamma^2}G^2 + \frac{6M}{\gamma m_2 \eta}\frac{16(d+b)u\eta}{\gamma} \\
&\leq \mathbb{E}\left[\mathcal{E}(x_0, v_0)\right] + \frac{54\left(4u+\gamma^2\right)u}{m_2\gamma^4}\left(\frac{\Delta^2 d}{4} + \sigma^2\right) + \frac{12(22u+\gamma)uM^3}{m_2\gamma^3}G^2 + \frac{96(d+b)uM}{m_2\gamma^2} \\
&=: \mathcal{E} + C\Delta^2 d.
\end{aligned}
$$

Finally by the fact that $\mathbb{E}\left[\|\mathbf{x}_k\|^2\right] \leq \mathbb{E}\left[\mathcal{E}(\mathbf{x}_k, \mathbf{v}_k)\right]$ and $\mathbb{E}\left[\|\mathbf{v}_k\|^2\right] \leq \gamma^2\mathbb{E}\left[\mathcal{E}(\mathbf{x}_k, \mathbf{v}_k)\right]/2$ we can get our claim in Lemma 11.

$\square$

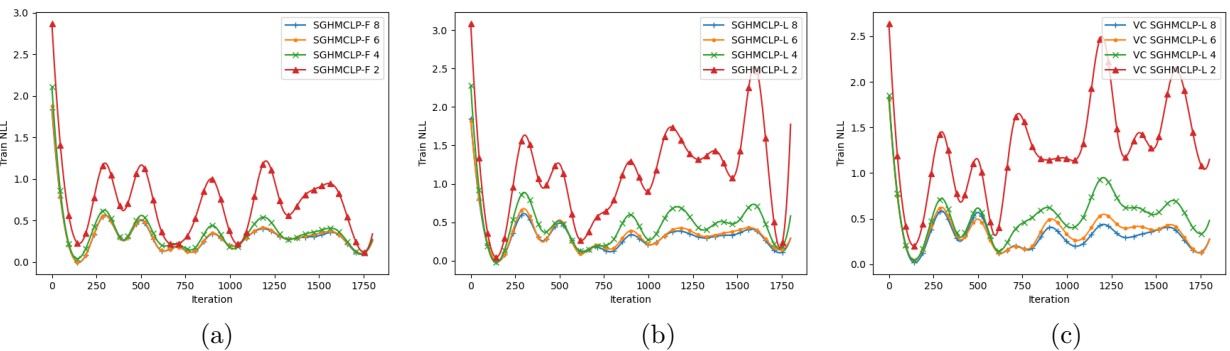

Figure 8: Train NLL of low-precision SGHMC on logistic model with MNIST in terms of different numbers of fractional bits. (a): Methods with Full-precision gradient accumulators. (b): Methods with Low-precision gradients accumulators. (c): Variance corrected quantization.

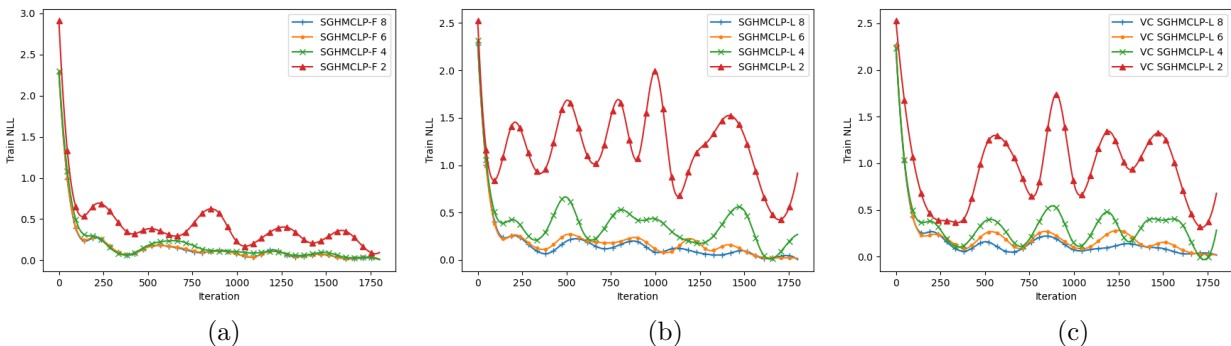

Figure 9: Train NLL of low-precision SGHMC on MLP with MNIST in terms of different numbers of fractional bits. (a): Methods with full-precision gradient accumulators. (b): Methods with low-precision gradient accumulators. (c): Variance corrected quantization.

## G  Additional experiment results

In this section, we provide additional experiment results.

### G.1  Logistic model

In this section, we present the low-precision SGHMC with logistic models on the MNIST dataset. The results are shown in Figure 8. We can see that SGHMCLP-F is robust to the quantization error, even though only 2 bits are used to represent the fractional part the SGHMCLP-F can converge to a good point.

### G.2  Multi-layer perception

We present the low-precision SGHMC with MLP on MNIST dataset in Figure 9. We observe similar results as the low-precision SGHMC with the logistic model.

## H  Generalization of Theorem 4 under Relaxed Variance Assumption 7

All theorems presented in the paper assume bounded variance for the stochastic variance (i.e., Assumption 3). Another more flexible yet commonly used variance assumption (e.g., Raginsky et al., 2017; Gao et al.,

2022) allows the variance scales with $\|\mathbf{x}\|^2$. In the rest of this supplementary material (Sections G-O), we generalize all the theorems in the main text to such a variance assumption, i.e. Assumption 7. Note that most of the arguments used in the proofs under Assumption 3 still hold, necessitating only slight modifications. Hence for the sake of readability, we only present the key changes due to the differences in the variance assumptions.

**Assumption 7** (Bouned Variance). *There exists a constant $\tilde{B} \geq 0$, such that $\left\|\widetilde{\nabla U}(0)\right\|^2 \leq \tilde{B}$. And there exists a constant $\delta \in [0, 1)$ such that:*

$$\mathbb{E}\left\|\widetilde{\nabla U}(\mathbf{x}) - \nabla U(\mathbf{x})\right\|^2 \leq 2\delta\left(M^2\|\mathbf{x}\|^2 + \tilde{B}^2\right), \quad \text{for any } \mathbf{x} \in \mathbb{R}^d.$$

Let Assumption 7 be true, we can then derive the following lemma.

**Lemma 16.** *For any $\mathbf{x} \in \mathbb{R}^d$, the random noise $\xi$ of the low-precision gradients defined in (40) satisfies:*

$$\|\mathbb{E}\xi\|^2 \leq M^2\frac{\Delta^2 d}{4}, \tag{94}$$

$$\mathbb{E}[\|\xi\|^2] \leq \left(2M^2 + 1\right)\frac{\Delta^2 d}{4} + 2\delta M^2\mathbb{E}\|\mathbf{x}\|^2 + 2\delta\tilde{B}^2. \tag{95}$$

Now, we are ready to present the Theorem 4 after revision.

**Theorem 17.** *Suppose Assumptions 1, 4, and 7 hold and the minimum satisfies $\|\mathbf{x}^*\|^2 < \mathcal{D}^2$. Furthermore, let $p^*$ denote the target distribution of $\mathbf{x}$ and $\mathbf{v}$. Given any sufficiently small $\epsilon$, if we set the step size to be*

$$\eta = \min\left\{\frac{\epsilon\kappa_1^{-1}}{\sqrt{479232/5(d/m_1 + \mathcal{D}^2)}}, \frac{\epsilon^2}{72\left(20u^2\delta M^2\kappa_1(d/m_1 + \mathcal{D}^2) + 5u^2\left((2M^2 + 1)\frac{\Delta^2 d}{4}\right) + \delta\tilde{B}^2\right)}, \sqrt{\frac{e^{1/(4\kappa_1)} - 1}{10u\delta M^2}}\right\},$$

*then after $K$ steps starting with initial points $\mathbf{x}_0 = \mathbf{v}_0 = 0$, the output $(\mathbf{x}_K, \mathbf{v}_K)$ of the SGHMCLP-F in (5) satisfies*

$$\mathcal{W}_2(p(\mathbf{x}_K, \mathbf{v}_K), p^*) \leq \tilde{\mathcal{O}}(\epsilon + \Delta),$$

*for some $K$ satisfying*

$$K \leq \frac{\kappa_1}{\eta}\log\left(\frac{36\left(\frac{d}{m_1} + \mathcal{D}^2\right)}{\epsilon}\right) = \tilde{\mathcal{O}}\left(\sqrt{\delta}\epsilon^{-2}\log\left(\epsilon^{-1}\right)\Delta^2\right).$$

*Proof.* For strongly log-concave target distributions, the equation (77) in Lemma 10 needs to be updated as the following,

$$\mathcal{W}_2^2(\widehat{\Phi}_\eta q_0, q^*) \le \left(\mathcal{W}_2(\widetilde{\Phi}_\eta q_0, q^*) + \sqrt{5}/2u\eta\sqrt{d}M\Delta\right)^2 + 5u^2\eta^2\left((2M^2+1)\frac{\Delta^2 d}{4} + \delta M^2\mathbb{E}_{q_0}\|\mathbf{x}\|^2 + \delta\delta\tilde{B}^2\right)$$

$$\le \left(\mathcal{W}_2(\widetilde{\Phi}_\eta q_0, q^*) + \sqrt{5}/2u\eta\sqrt{d}M\Delta\right)^2 + 10u^2\eta^2\delta M^2\mathbb{E}_\zeta\|\mathbf{x}-\mathbf{x}'\|^2 + 10u^2\eta^2\delta M^2\mathbb{E}_{p^*}\|\mathbf{x}\|^2$$

$$+ 5u^2\eta^2\left((2M^2+1)\frac{\Delta^2 d}{4} + \delta\delta\tilde{B}^2\right) \tag{96}$$

$$= \left(\mathcal{W}_2(\widetilde{\Phi}_\eta q_0, q^*) + \sqrt{5}/2u\eta\sqrt{d}M\Delta\right)^2 + 10u^2\eta^2\delta M^2\mathcal{W}_2^2(\widetilde{\Phi}_\eta q_0, q^*) + 10u^2\eta^2\delta M^2\mathbb{E}_{p^*}\|\mathbf{x}\|^2$$

$$+ 5u^2\eta^2\left((2M^2+1)\frac{\Delta^2 d}{4} + \delta\delta\tilde{B}^2\right) \tag{97}$$

$$\le \left(\left(1 + 10u^2\eta^2\delta M^2\right)\mathcal{W}_2(\widetilde{\Phi}_\eta q_0, q^*) + \sqrt{5}/2u\eta\sqrt{d}M\Delta\right)^2 + 10u^2\eta^2\delta M^2\mathbb{E}_{p^*}\|\mathbf{x}\|^2$$

$$+ 5u^2\eta^2\left((2M^2+1)\frac{\Delta^2 d}{4} + \delta\tilde{B}^2\right). \tag{98}$$

In the second inequality, $\zeta$ is an optimal coupling between $q_0$ and $q_*$. $\mathbf{x}/\mathbf{x}'$ are from the distributions $q_0/q^*$. Then we choose $\eta \le \sqrt{\frac{e^{1/(4\kappa_1)}-1}{10u\delta M^2}}$, and following the same argument for deriving equation (46), we can have:

$$\mathcal{W}_2^2(\widehat{\Phi}_\eta q_i, q^*) \le \left(e^{-\eta/4\kappa_1}\mathcal{W}_2(q_i, q^*) + \eta^2\sqrt{\frac{8\mathcal{E}_K}{5}} + \sqrt{5}/2u\eta\sqrt{d}M\Delta\right)^2 + 10u^2\eta^2\delta M^2\mathbb{E}_{p^*}\|\mathbf{x}\|^2 \tag{99}$$

$$+ 5u^2\eta^2\left((2M^2+1)\frac{\Delta^2 d}{4} + \delta M^2 B^2\right). \tag{100}$$

Moreover, we know the fact that

$$\mathbb{E}_{p^*}\|\mathbf{x}\|^2 = \mathbb{E}_{p^*}\|\mathbf{x}-\mathbf{x}_0\|^2 \tag{101}$$

$$\le 2\mathbb{E}\|\mathbf{x}-\mathbf{x}^*\|^2 + 2\|\mathbf{x}^*-\mathbf{x}_0\|^2 \tag{102}$$

$$\le \frac{2d}{m} + 2\mathcal{D}^2 \tag{103}$$

Thus plug it into the equation (99), we can have

$$\mathcal{W}_2(\widehat{\Phi}_\eta q_i, q^*) \le \left(e^{-\eta/4\kappa_1}\mathcal{W}_2(q_i, q^*) + \eta^2\sqrt{\frac{8\mathcal{E}_K}{5}} + \sqrt{5}/2u\eta\sqrt{d}M\Delta\right)^2 + 10u^2\eta^2\delta M^2\left(\frac{2d}{m} + 2\mathcal{D}^2\right) \tag{104}$$

$$+ 5u^2\eta^2\left((2M^2+1)\frac{\Delta^2 d}{4} + \delta\tilde{B}^2\right). \tag{105}$$

Finally, by invoking Lemma 7 Dalalyan & Karagulyan (2019), we can have

$$\mathcal{W}_2(q_K, q^*) \le e^{-K\eta/4\kappa_1}\mathcal{W}_2^2(q_0, q^*) + \frac{\eta^2\sqrt{\frac{8\mathcal{E}_K}{5}} + \frac{u\eta M\Delta\sqrt{5d}}{2}}{1 - e^{-\eta/2\kappa_1}} \tag{106}$$

$$+ \frac{10u^2\eta^2\delta M^2\left(\frac{2d}{m} + 2\mathcal{D}^2\right) + 5u^2\eta^2\left((2M^2+1)\frac{\Delta^2 d}{4} + \delta\tilde{B}^2\right)}{\eta^2\sqrt{\frac{8\mathcal{E}_K}{5}} + \frac{u\eta M\Delta\sqrt{5d}}{2} + \sqrt{1 - e^{-\eta/2\kappa_1}}\sqrt{10u^2\eta^2\delta M^2\left(\frac{2d}{m} + 2\mathcal{D}^2\right) + 5u^2\eta^2\left((2M^2+1)\frac{\Delta^2 d}{4} + \delta\tilde{B}^2\right)}}. \tag{107}$$

Thus for some $K$ satisfied

$$K \leq \frac{2\kappa_1}{\eta} \log \left( 36 \left( \frac{d}{m_1} + \mathcal{D}^2 \right) \right),$$

we can get the following

$$\mathcal{W}_2(p_K, p^*) \leq \epsilon + \frac{16\kappa_1 u M \Delta \sqrt{5d}}{2}. \tag{108}$$

$\square$

## I  Generalization of Theorem 5 under Relaxed Variance Assumption 7

Now, we are ready to present the Theorem 5 after revision.

**Theorem 18.** *Suppose Assumptions 1, 4, and 7 hold and the minimum satisfies $\|\mathbf{x}^*\|^2 < \mathcal{D}^2$. Furthermore, let $p^*$ denote the target distribution of $\mathbf{x}$ and $\mathbf{v}$. Given any sufficiently small $\epsilon$, if we set the step size to be*

$$\eta = \min \left\{ \frac{\epsilon \kappa_1^{-1}}{\sqrt{663552/5(d/m_1 + \mathcal{D}^2)}}, \frac{\epsilon^2}{144 \left( 20u^2 \delta M^2 \kappa_1 (d/m_1 + \mathcal{D}^2) + 5u^2 \left( (2M^2 + 1)\frac{\Delta^2 d}{4} \right) + \delta \tilde{B}^2 \right)}, \sqrt{\frac{e^{1/(4\kappa_1)} - 1}{10u\delta M^2}} \right\},$$

*then after $K$ steps starting with initial points $\mathbf{x}_0 = \mathbf{v}_0 = 0$, the output $(\mathbf{x}_K, \mathbf{v}_K)$ of the SGHMCLP-L in (5) satisfies*

$$\mathcal{W}_2(p(\mathbf{x}_K, \mathbf{v}_K), p^*) \leq \tilde{\mathcal{O}} \left( \epsilon + \frac{\Delta}{\epsilon} \right),$$

*for some $K$ satisfying*

$$K \leq \frac{\kappa_1}{\eta} \log \left( \frac{36 \left( \frac{d}{m_1} + \mathcal{D}^2 \right)}{\epsilon} \right) = \tilde{\mathcal{O}} \left( \sqrt{\delta} \epsilon^{-2} \log \left( \epsilon^{-1} \right) \Delta^2 \right).$$

*Proof.* Now we revise the proof of Theorem 5. The revision is similar to the revision for Theorem 5.

By similar argument in (98), equation (52) need to be changed as the following:

$$\mathcal{W}_2^2 \left( \hat{\Phi}_\eta q_0, q^* \right) \leq \mathcal{W}_2^2(\widetilde{\Phi}_\eta q_0, q^*) + 5u^2\eta^2 \left( \frac{\Delta^2 d}{4} + \delta M^2 \mathbb{E}_{q_i} \|\mathbf{x}\|^2 + \tilde{B}^2 \right) + 2u^2 (A + B) \tag{109}$$

$$\leq \mathcal{W}_2^2(\widetilde{\Phi}_\eta q_0, q^*) + 10u^2\eta^2 \delta M^2 \mathbb{E}_\zeta \|\mathbf{x} - \mathbf{x}'\|^2 + 10u^2\eta^2 \delta M^2 \mathbb{E}_{p^*} \|\mathbf{x}\|^2 \tag{110}$$

$$+ 5u^2\eta^2 \tilde{B}^2 + 5u^2\eta^2 \frac{\Delta^2 d}{4} + 2u^2(A + B) \tag{111}$$

$$\leq \left( 1 + 10u^2\eta^2 \delta M^2 \right) \mathcal{W}_2^2(\widetilde{\Phi}_\eta q_0, q^*) + 10u^2\eta^2 \delta M^2 \left( \frac{2d}{m} + \mathcal{D}^2 \right) + 5u^2\eta^2 \tilde{B}^2 + 5u^2\eta^2 \frac{\Delta^2 d}{4} + 2u^2(A + B). \tag{112}$$

Then we choose $\eta \leq \sqrt{\frac{e^{1/(4\kappa_1)} - 1}{10u\delta M^2}}$, and following the same argument for (52), we can have

$$\mathcal{W}_2^2 \left( \hat{\Phi}_\eta q_i, q^* \right) \leq \left( e^{-\eta/4\kappa_1} \mathcal{W}_2 \left( q_i, q^* \right) + \eta^2 \sqrt{\frac{8\mathcal{E}_K}{5}} \right)^2 + 10u^2\eta^2 \delta M^2 \left( \frac{2d}{m} + \mathcal{D}^2 \right) \tag{113}$$

$$+ 5u^2\eta^2 \tilde{B}^2 + 5u^2\eta^2 \frac{\Delta^2 d}{4} + 2u^2(A + B). \tag{114}$$

Then the remaining analysis follows the proof in the original Theorem 5.

$\square$

## J  Generalization of Theorem 6 under Relaxed Variance Assumption 7

**Theorem 19.** *Let Assumption 1, 4 and 7 hold and the minimum satisfies $\|\mathbf{x}^*\|^2 < \mathcal{D}^2$. Furthermore, let $p^*$ denote the target distribution of $\mathbf{v}$ and $\mathbf{x}$. Given any sufficiently small $\epsilon$, if we set the step size $\eta$ to be*

$$\eta = \min\left\{ \frac{\epsilon\kappa_1^{-1}}{\sqrt{663552/5\left(\frac{d}{m_1}+\mathcal{D}^2\right)}}, \frac{\epsilon^2}{144\left(20u^2\delta M^2\kappa_1(d/m_1+\mathcal{D}^2)+5u^2\left((2M^2+1)\frac{\Delta^2 d}{4}\right)+\delta\tilde{B}^2\right)}, \sqrt{\frac{e^{1/(4\kappa_1)}-1}{10u\delta M^2}} \right\},$$

*then after $K$ steps starting with initial points $\mathbf{x}_0 = \mathbf{v}_0 = 0$, the output $(\mathbf{x}_K, \mathbf{v}_K)$ of the SGHMCLP-L in (6) satisfies*

$$\mathcal{W}_2(p(\mathbf{x}_K, \mathbf{v}_K), p^*) = \tilde{\mathcal{O}}\left(\epsilon + \sqrt{\Delta}\right), \tag{115}$$

*for some $K$ such that*

$$K \le \frac{\kappa_1}{\eta}\log\left(\frac{36\left(\frac{d}{m_1}+\mathcal{D}^2\right)}{\epsilon}\right) = \tilde{\mathcal{O}}\left(\epsilon^{-2}\log\left(\sqrt{\delta}\epsilon^{-1}\right)\Delta^2\right).$$

The revision closely mirrors the revised proof of Theorem 5, based on the assumption that Assumption 3 holds.

The only equation we need to revise is (60). Given the similar argument in (98), we can have

$$\mathcal{W}_2(p_K, p^*) \le 4e^{-K\eta/4\kappa_1}\mathcal{W}_2(q_0, q^*) + \frac{4\eta^2\sqrt{\frac{8\mathcal{E}_K}{5}}}{1-e^{-\eta/4\kappa_1}} \tag{116}$$

$$+ \frac{40u^2\eta^2\delta M^2\left(\frac{2d}{m_1}+2\mathcal{D}^2\right)+20u^2\eta^2\left(\frac{\Delta^2 d}{4}+\delta\tilde{B}^2\right)+8u^2\eta\left(\gamma A+B\right)}{\eta^2\sqrt{\frac{8\mathcal{E}_K}{5}}+\sqrt{1-e^{-\eta/\kappa_1}}\sqrt{10u^2\eta^2\delta M^2\left(\frac{2d}{m_1}+2\mathcal{D}^2\right)+5u^2\eta^2\left(\frac{\Delta^2 d}{4}+\delta\tilde{B}^2\right)+2u^2\eta\left(\gamma A+B\right)}}. \tag{117}$$

## K  Generalization of Theorem 1 under Relaxed Variance Assumption 7

This section updates Theorem 1's proof, replacing Assumption 3 with 7, and introduces the revised theorem.

**Theorem 20.** *Assuming 1, 2, 7, and 8 hold. Let $p^*$ denote the target distribution of $(\mathbf{x}, \mathbf{v})$. If $\gamma^2 \le 4Mu$ and setting the step size $\eta = \tilde{\mathcal{O}}\left(\frac{\mu^*\epsilon^2}{\log(1/\epsilon)}\right)$ satisfying*

$$\eta \le \min\left\{ \frac{\gamma}{4\left(8Mu+u\gamma+22\gamma^2\right)}, \sqrt{\frac{4u^2}{4Mu+3\gamma^2}}, \frac{6\gamma bu}{(4Mu+3\gamma^2)d}, \frac{1}{8\gamma}, \frac{\gamma m_2}{12(21u+\gamma)M^2}, \frac{8(\gamma^2+2u)}{(20u+\gamma)\gamma}, \frac{m_2\gamma^2}{12(\gamma^2+2u)M^2} \right\},$$

*then after $K$ steps starting at the initial point $\mathbf{x}_0 = \mathbf{v}_0 = 0$, the output $(\mathbf{x}_K, \mathbf{v}_K)$ of SGHMCLP-F in (5) satisfies*

$$\mathcal{W}_2(p(\mathbf{x}_K, \mathbf{v}_K), p^*) \le \tilde{\mathcal{O}}\left(\epsilon + \tilde{A}\sqrt{\log\left(\frac{1}{\epsilon}\right)}\right),$$

*for some $K$ satisfying*

$$K = \tilde{\mathcal{O}}\left(\frac{1}{\epsilon^2\mu^{*2}}\log^2\left(\frac{1}{\epsilon}\right)\right),$$

*where constants are defined as: $\tilde{A} = \max\left\{\sqrt{\Delta^2 d}+\delta^{1/4}, \sqrt[4]{\Delta^2 d}+\delta^{1/4}\right\}$, and constant $1/\mu^* = exp\left(\mathcal{O}(d)\right)$ denotes the contraction rate of underdamped Langevin dynamics (10).*

*Proof.* We first need to introduce a further assumption on the variance parameter $\delta$ as the following assumption.

**Assumption 8.** *Given Assumption 7, we further assume the $\delta$ satisfies the following condition:*

$$\delta \leq min \left\{ \frac{\gamma m_2}{12(21u + \gamma)M^2}, \frac{8(\gamma^2 + 2u)}{(20u + \gamma)\gamma}, \frac{m_2\gamma}{3(20u + \gamma)}, \frac{m_2\gamma^2}{12(\gamma^2 + 2u)M^2} \right\}.$$

We need to revise the (84) as the following:

$$\mathbb{E}\left[\mathcal{E}(\mathbf{x}_{k+1}, \mathbf{v}_{k+1})\right] \leq \mathbb{E}\left[\mathcal{E}(\mathbf{x}_k, \mathbf{v}_k)\right] - \frac{3um_2\eta}{\gamma}\mathbb{E}\left[\|\mathbf{x}_k\|^2\right] - \frac{2\eta}{\gamma}\mathbb{E}\left[\|\mathbf{v}_k\|^2\right] + \frac{(21u + \gamma)4M^2u\eta^2}{\gamma^2}\mathbb{E}\left[\|\mathbf{x}_k\|^2\right] \quad (118)$$

$$+ \left( \frac{2(20u + \gamma)u\eta^2}{\gamma^2} + \frac{8u\eta\left(\gamma^2 + 2u\right)}{\gamma^3} \right) \left( (M^2 + 1)\frac{\Delta^2 d}{4} + \delta M^2\mathbb{E}\|\mathbf{x}_k\|^2 + \delta\tilde{B}^2 \right) \quad (119)$$

$$+ \frac{(21u + \gamma)4u\eta^2}{\gamma^2}G^2 + \frac{16(d + b)u\eta}{\gamma}. \quad (120)$$

If we choose $\eta$ satisfy the following condition

$$\delta \leq \eta \leq \min \left\{ \frac{\gamma m_2}{12(21u + \gamma)M^2}, \frac{8(\gamma^2 + 2u)}{(20u + \gamma)\gamma}, \frac{m_2\gamma}{3(20u + \gamma)}, \frac{m_2\gamma^2}{12(\gamma^2 + 2u)M^2} \right\}, \quad (121)$$

then we can have

$$\mathbb{E}\left[\mathcal{E}(\mathbf{x}_{k+1}, \mathbf{v}_{k+1})\right] \leq \mathbb{E}\left[\mathcal{E}(\mathbf{x}_k, \mathbf{v}_k)\right] - \frac{4um_2\eta}{3\gamma}\mathbb{E}\left[\|\mathbf{x}_k\|^2\right] - \frac{2\eta}{\gamma}\mathbb{E}\left[\|\mathbf{v}_k\|^2\right] \quad (122)$$

$$+ \left( \frac{16u\eta\left(\gamma^2 + 2u\right)}{\gamma^3} \right) \left( (M^2 + 1)\frac{\Delta^2 d}{4} + \delta\tilde{B}^2 \right) \quad (123)$$

$$+ \frac{(21u + \gamma)4u\eta^2}{\gamma^2}G^2 + \frac{16(d + b)u\eta}{\gamma}. \quad (124)$$

The remainder of the analysis is unchanged.

$\square$

## L Generalization of Theorem 2 under Relaxed Variance Assumption 7

This section updates Theorem 2's proof, replacing Assumption 3 with 7, and introduces the revised theorem.

**Theorem 21.** *Assuming 1, 2, 7 and 8 hold. Let $p^*$ denote the target distribution of $(\mathbf{x}, \mathbf{v})$. If $\gamma^2 \leq 4Mu$ and setting the step size $\eta = \tilde{\mathcal{O}}\left(\frac{\mu^*\epsilon^2}{\log(1/\epsilon)}\right)$ satisfying*

$$\eta \leq \min \left\{ \frac{\gamma}{4\left(8Mu + u\gamma + 22\gamma^2\right)}, \sqrt{\frac{4u^2}{4Mu + 3\gamma^2}}, \frac{6\gamma bu}{\left(4Mu + 3\gamma^2\right)d}, \frac{1}{8\gamma}, \frac{\gamma m_2}{12(21u + \gamma)M^2}, \frac{8(\gamma^2 + 2u)}{(20u + \gamma)\gamma}, \frac{m_2\gamma^2}{12(\gamma^2 + 2u)M^2} \right\},$$

*then after $K$ steps starting at the initial point $\mathbf{x}_0 = \mathbf{v}_0 = 0$, the output $(\mathbf{x}_K, \mathbf{v}_K)$ of SGHMCLP-L in (6) satisfies*

$$\mathcal{W}_2(p(\mathbf{x}_K, \mathbf{v}_K), p^*) = \tilde{\mathcal{O}}\left( \epsilon + \delta^{1/4}\sqrt{\log\left(\frac{1}{\epsilon}\right)} + \frac{\log^{3/2}\left(\frac{1}{\epsilon}\right)}{\epsilon^2}\sqrt{\Delta} \right), \quad (125)$$

*for some $K$ satisfying*

$$K = \tilde{\mathcal{O}}\left( \frac{1}{\epsilon^2\mu^{*2}}\log^2\left(\frac{1}{\epsilon}\right) \right).$$

*Proof.* We need to revise the equation (93) as the following:

$$\mathbb{E}\left[\mathcal{E}\left(\mathbf{x}_{k+1}, \mathbf{v}_{k+1}\right)\right] \leq \mathbb{E}\left[\mathcal{E}(\mathbf{x}_k, \mathbf{v}_k)\right] - \frac{8um_2\eta}{3\gamma}\mathbb{E}\left[\|\mathbf{x}_k\|^2\right] - \frac{2\eta}{\gamma}\mathbb{E}\left[\|\mathbf{v}_k\|^2\right] \tag{126}$$

$$+ \frac{\left(36u + 9\gamma^2\right)u\eta}{\gamma^3}\left(\frac{\Delta^2 d}{4} + \delta M^2\mathbb{E}\|\mathbf{x}_k\|^2 + \delta\tilde{B}^2\right) + \frac{2(22u + \gamma)u\eta^2 M^2}{\gamma^2}G^2 + \frac{16\left(d + b\right)u\eta}{\gamma}. \tag{127}$$

If we further choose $\eta \leq \frac{4\gamma^2 m_2}{3(36u+9\gamma^2)M^2}$ and assume $\delta \leq \eta$, then we can have

$$\mathbb{E}\left[\mathcal{E}\left(\mathbf{x}_{k+1}, \mathbf{v}_{k+1}\right)\right] \leq \mathbb{E}\left[\mathcal{E}(\mathbf{x}_k, \mathbf{v}_k)\right] - \frac{4um_2\eta}{3\gamma}\mathbb{E}\left[\|\mathbf{x}_k\|^2\right] - \frac{2\eta}{\gamma}\mathbb{E}\left[\|\mathbf{v}_k\|^2\right] \tag{128}$$

$$+ \frac{\left(36u + 9\gamma^2\right)u\eta}{\gamma^3}\left(\frac{\Delta^2 d}{4} + \delta\tilde{B}^2\right) + \frac{2(22u + \gamma)u\eta^2 M^2}{\gamma^2}G^2 + \frac{16\left(d + b\right)u\eta}{\gamma}. \tag{129}$$

$$\square$$

## M  Generalization of Theorem 3 under Relaxed Variance Assumption 7

In this section, we present Theorem 3 after revision.

**Theorem 22.** *Assuming 1, 2, 7 and 8 hold. Let $p^*$ denote the target distribution of $(\mathbf{x}, \mathbf{v})$. If $\gamma^2 \leq 4Mu$ and setting the step size $\eta = \tilde{\mathcal{O}}\left(\frac{\mu^*\epsilon^2}{\log(1/\epsilon)}\right)$ satisfying*

$$\eta \leq \min\left\{\frac{\gamma}{4\left(8Mu + u\gamma + 22\gamma^2\right)}, \sqrt{\frac{4u^2}{4Mu + 3\gamma^2}}, \frac{6\gamma bu}{\left(4Mu + 3\gamma^2\right)d}, \frac{1}{8\gamma}, \frac{\gamma m_2}{12(21u + \gamma)M^2}, \frac{8(\gamma^2 + 2u)}{(20u + \gamma)\gamma}, \frac{m_2\gamma^2}{12(\gamma^2 + 2u)M^2}\right\},$$

*then after $K$ steps starting at the initial point $\mathbf{x}_0 = \mathbf{v}_0 = 0$, the output $(\mathbf{x}_K, \mathbf{v}_K)$ of SGHMCLP-L in (6) satisfies*

$$\mathcal{W}_2(p(\mathbf{x}_K, \mathbf{v}_K), p^*) = \tilde{\mathcal{O}}\left(\epsilon + \delta^{1/4}\sqrt{\log\left(\frac{1}{\epsilon}\right)} + \frac{\log\left(\frac{1}{\epsilon}\right)}{\epsilon}\sqrt{\Delta}\right), \tag{130}$$

*for some $K$ satisfying*

$$K = \tilde{\mathcal{O}}\left(\frac{1}{\epsilon^2\mu^{*2}}\log^2\left(\frac{1}{\epsilon}\right)\right).$$

The revision of Theorem 3 is the same as the revision for Theorem 2.

## N  Generalization of Theorem 7 under Relaxed Variance Assumption 7

In this proof, we revise the proof of Theorem 7 if the Assumption 3 is replaced by Assumption 7.

**Theorem 23.** *Suppose Assumptions 1, 2, and 7 hold. Let $p^*$ denote the target distribution of $\mathbf{x}$, $\widetilde{A}$ have the same definition in Theorem 1, and $1/\lambda^* = exp\left(\mathcal{O}(d)\right)$ be the concentration number of (13). After $K$ steps starting with initial point $\mathbf{x}_0 = 0$, if we set the stepsize to be $\eta = \tilde{\mathcal{O}}\left(\left(\frac{\epsilon}{\log(1/\epsilon)}\right)^4\right)$. The output $\mathbf{x}_K$ of SGLDLP-F in (1) satisfies*

$$\mathcal{W}_2(p(\mathbf{x}_K), p^*) \leq \tilde{\mathcal{O}}\left(\epsilon + \left(\sqrt{\Delta} + \delta^{1/4}\right)\log\left(\frac{1}{\epsilon}\right)\right), \tag{131}$$

*for some $K$ satisfied*

$$K = \tilde{\mathcal{O}}\left(\frac{1}{\epsilon^4\lambda^*}\log^5\left(\frac{1}{\epsilon}\right)\right).$$

*Proof.* We need to revise the equation (90) as the following:

$$\mathbb{E}\left[\|\mathbf{x}_{k+1}\|^2\right] \leq \left(1 - \eta m_2 + 2\eta^2 M^2\right)\mathbb{E}\left[\|\mathbf{x}_k\|^2\right] + \frac{1 - \eta m_2 + \eta^2 M}{\eta m_2 - \eta^2 M}\frac{M^2\eta^2\Delta^2 d}{4} + \frac{1 - \eta m_2 + \eta^2 M}{1 - 2\eta m_2 + 2\eta^2 M^2}\left(2\eta b + 2\eta^2 G^2\right) \tag{132}$$

$$+ \eta^2\left(\frac{\Delta^2 d}{4} + \delta\tilde{B}^2\right) + 2\eta d. \tag{133}$$

Then we can have the following:

$$\mathbb{E}\left[\|\mathbf{x}_k\|^2\right] \leq \mathcal{E} + \frac{2\left(M^2 + 1\right)}{m_2}\frac{\Delta^2 d}{4} + \frac{2\delta\tilde{B}^2}{m_2}.$$

Moreover, the equation (67) is also needed to revised as the accordingly:

$$\mathbb{E}\left[\|\mathbf{x}_s - \mathbf{x}_k\|^2\right] \leq 3\eta^2\left(M\mathbb{E}\left[\|\mathbf{x}_k\|\right] + G\right)^2 + 3\eta^2\left((M^2 + 1)\frac{\Delta^2 d}{4} + \delta M^2\mathbb{E}\|\mathbf{x}_k\|^2 + \delta\tilde{B}^2\right) + 6\eta d$$

$$\leq 3\eta^2\left(2M\mathbb{E}\left[\|\mathbf{x}_k\|\right] + G\right)^2 + 3\eta^2\left((M^2 + 1)\frac{\Delta^2 d}{4} + \delta\tilde{B}^2\right) + 6\eta d. \tag{134}$$

The remaining analysis should be the same as the proof of Theorem 7.

$$\square$$

## O  Generalization of Theorem 8 under Relaxed Variance Assumption 7

In this proof, we revise the proof of Theorem 8 if the Assumption 3 is replaced by Assumption 7.

**Theorem 24.** *Let Assumptions 1, 2 and 7 hold. Let $p^*$ denote the target distribution of $\mathbf{x}$ and $1/\lambda^* = exp\left(\mathcal{O}(d)\right)$ be the concentration number of (13). If we set the step size to be $\eta = \tilde{\mathcal{O}}\left(\left(\frac{\epsilon}{\log(1/\epsilon)}\right)^4\right)$, after $K$ steps starting at the initial point $\mathbf{x}_0 = 0$ the output $\mathbf{x}_K$ of the SGLDLP-L in (2) satisfies*

$$\mathcal{W}_2(p(\mathbf{x}_K), p^*) = \tilde{\mathcal{O}}\left(\epsilon + \delta^{1/4}\log\left(\frac{1}{\epsilon}\right) + \frac{\log^5\left(\frac{1}{\epsilon}\right)}{\epsilon^4}\sqrt{\Delta}\right), \tag{135}$$

*for some $K$ satisfied*

$$K = \tilde{\mathcal{O}}\left(\frac{1}{\epsilon^4\lambda^*}\log^5\left(\frac{1}{\epsilon}\right)\right).$$

We need to revise the (71)

$$\mathbb{E}\left[\|\mathbf{x}_{k+1}\|^2\right] \leq \left(1 - 2\eta m_2 + 2\eta^2 M^2\right)\mathbb{E}\left[\|\mathbf{x}_k\|^2\right] + 2\eta b + 2\eta^2 G^2 + \frac{\eta^2\Delta^2 d}{4} + \eta^2\left(2\delta M^2\mathbb{E}\|\mathbf{x}_k\|^2 + 2\delta\tilde{B}^2\right) \tag{136}$$

$$+ \mathbb{E}\left[\|\alpha_k\|^2\right] + 2\eta d. \tag{137}$$

$$\leq \left(1 - 2\eta m_2 + 4\eta^2 M^2\right)\mathbb{E}\left[\|\mathbf{x}_k\|^2\right] + 2\eta b + 2\eta^2 G^2 + \frac{\eta^2\Delta^2 d}{4} + \eta^2\delta\tilde{B}^2 + \mathbb{E}\left[\|\alpha_k\|^2\right] + 2\eta d. \tag{138}$$

## P    Generalization of Theorem 9 under Relaxed Variance Assumption 7

In this proof, we update Theorem 9 if the Assumption 3 is replaced by Assumption 7.

**Theorem 25.** *Let Assumptions 1, 2 and 7 hold. Let $p^*$ denote the target distribution of $\mathbf{x}$ and $1/\lambda^* = exp\left(\mathcal{O}(d)\right)$ be the concentration number of (13). If we set the step size to be $\eta = \tilde{\mathcal{O}}\left(\left(\frac{\epsilon}{\log(1/\epsilon)}\right)^4\right)$, after $K$ steps starting at the initial point $\mathbf{x}_0 = 0$ the output $\mathbf{x}_K$ of the SGLDLP-L in (2) satisfies*

$$\mathcal{W}_2(p(\mathbf{x}_K), p^*) = \tilde{\mathcal{O}}\left(\epsilon + \delta^{1/4}\log\left(\frac{1}{\epsilon}\right) + \frac{\log^3\left(\frac{1}{\epsilon}\right)}{\epsilon^2}\sqrt{\Delta}\right), \tag{139}$$

*for some $K$ satisfied*

$$K = \tilde{\mathcal{O}}\left(\frac{1}{\epsilon^4\lambda^*}\log^5\left(\frac{1}{\epsilon}\right)\right).$$

The key revision is the same as the Theorem 8.

