# OpenReview forum: "Enhancing Low-Precision Sampling via Stochastic Gradient Hamiltonian Monte Carlo"
_TMLR — Accepted by TMLR_

### Review · Reviewer_59sv · 2024-01-10

**Summary Of Contributions:**

This paper studied sampling with low-precision gradients, in particular it conducted a thorough analysis of Stochastic Gradient Hamiltonian Monte Carlo (SGHMC) with low-precision and full-precision gradient accumulators for both strongly log-concave and non-log-concave distributions. The results suggested that low-precision SGHMC has priority over stochastic gradient Langevin dynamics (SGLD). Numerical experiments are also performed, which support the theoretical claims.

**Audience:**

Yes

**Broader Impact Concerns:**

From my perspective, this paper does not have broader impact concerns.

**Claims And Evidence:**

Yes

**Requested Changes:**

From my perspective, there are quite a few places that the writing of the paper is not clear enough. I think the paper needs major revision before being accepted to TMLR.

- The introduction does not have a high-level discussion about the technical contributions. I believe that for the TMLR audiences, some are familiar with theoretical analysis of sampling algorithms and some are familiar with low-precision algorithms, but few are familiar with applying low-precision gradients to sampling algorithms. It will be very helpful if the authors can highlight how the analysis of sampling algorithms can still go through with low-precision gradients. It will also be good if the authors can highlight the technical contributions on this compared to Zhang et al. (2022) - since this paper is the first one that analyzed low-precision sampling algorithms, does the current paper mostly follow Zhang et al. (2022), or it has novelties when analyzing SGHMC? In addition, is it possible to put the result of Zhang et al. (2022) in the table for a more direct comparison?

- The settings of the problem can be better explained in many places.

- - In the table, it will be very helpful if the authors can add a column titled “Settings”, and briefly explain which settings these theorems apply to. Apparently, the results are written in different parameters, and it is worth highlighting the specific settings they concern.

- - I found the definitions of mu* and lambda* not clear enough. For mu*, Theorem 1 mentions it as the spectral gap of the underdamped Langevin dynamics and points it to Zou et al. (2019). It will be much better if this can be written down explicitly so that it is self-contained for readers to check. For lambda*, the caption of the table mentions is as the spectral gap of the overdamped Langevin dynamics, whereas Theorem 7 mentions it as the concentration number of (13). Do they share the same definition here?

- - The definition of “low-precision” can also be better explained. In the table it is denoted by a parameter Delta - quantization error, but which type of quantization error does it refer to? There are deterministic rounding, stochastic rounding, and variance correction rounding. It would be helpful to explain in the main body which rounding methods can be applied in the proposed theorems.

- It would be helpful to explain the relationship between the proofs of different theorems. Currently, Appendix D has 9 subsections going through the proofs of Theorem 1 to Theorem 9. The section is very long, but from reading the proofs share some similarities. It would be very helpful if the authors can explain at the beginning of the subsections the similarity and difference between these proofs.

- In practice, if we only have a low-precision computer say with 8-bit digit numbers, how to do the stochastic rounding? I can imagine that for deterministic rounding to the nearest neighbor, the last bit after precision point may be stored so that if it’s .1, it is rounded to the integer right above, and if it’s .0 it is rounded to the current integer, i.e., simply omit the digits after the precision point. But how in practice can we make an unbiased random variable with low-precision?

- I’m confused about Page 5 and 6 that use log(x)<=x^{1/e} and obtain subsequent bounds with powers expressed by e. Isn’t it the case that log(x)<=x^c for any constant c>0 in asymptotic bounds?

**Strengths And Weaknesses:**

In terms of strengths, the study of low-precision sampling algorithms is of general interest because low-precision is an important factor for training machine learning models when saving cost. It is nice to see that the authors are able to go through the analyses and achieve theoretical guarantees for SGHMC even with low-precision gradients. The proofs are solid and the theories applies to various settings. In addition, the paper also conducted extensive numerical experiments, which are convincing and provide evidence for the correctness of the theory results.

In terms of weaknesses, from my perspective the paper has significant space to improve in presentation because many places are unclear. See my comments in “requested changes”. In addition, technically the paper follows very closely to Zhang et al. (2022), especially the analysis related to low-precision gradients. More detailed comparisons between this paper and the current submission can be very helpful.

---

> ### Author Response · Authors · 2024-02-15
> **Rebuttal by Authors**
>
> Thanks for your supportive and insightful comments! Please find our responses to your questions below.
> # Q1. Technical contributions
> Thanks for the suggestion. We have revised the introduction to highlight our technical contributions.
>
> In short, existing works on low-precision sampling only consider strongly log-concave target distribution using overdamped Langevin dynamics. This work extends to non-log-concave target distributions, which are more reflective of real-world applications, as well as underdamped Langevin dynamics, known for their increased efficiency. To achieve these, we introduce an intermediate process to handle the quantizations of the state $x$, the momentum $v$, and stochastic gradients.
>
> # Q2. Differences between Zhang and Analysis with Low-precision gradients
> Zhang et.al (2022) mainly follows Theorem 4 of [1], adapting it to low-precision SGLD by incorporating the quantization error into the stochastic gradient noise.
> This approach, however, is not applicable to analyzing SGHMC. In contrast, our analysis follows the technical tools of Theorem 6 of [2], Theorem 3.3 of [3], and Theorem 3 of [4], diverging significantly from Zhang et al. (2022). Since we quantize the state $x$, the momentum $v$, and stochastic gradients, the analysis needs to handle all three types of errors introduced by quantization. To do so, we define an intermediate process $(v', x')$ such that $x'$ has the same distribution as the original process.
> The quantization error on $x$ is absorbed in the $v'$ process such that the intermediate process still follows the underdamped Langevin dynamic.
>
>
> Zhang et.al (2022)'s bounds only consider strongly log-concave target distributions. Our results show that low-precision SGHMC and SGLD achieved the same W2 upper bound in the same number of iterations, thus we cannot claim advantages of low-precision SGHMC over SGLD under strongly concave target distributions. And this result is not surprising according to [4]. In Table 1, we combinedly show that SGLD (results in Zhang et. al (2022)) and SGHMC  (our Theorems 4,5,6) share the same rate under strong log-concavity. On non-log-concave target distributions, we provide bounds for both low-precision SGHMC and SGLD, and found that low-precision SGHMC achieves a better bound.
>
> The above discussion has been incorporated into the introduction section of the paper.
> # Q3. Add a new column "setting" in the table
> We added a column "setting" in Table 1. Please see https://anonymous.4open.science/r/Low-precisionSGHMC-D6E7/tab_1.pdf or the updated version for the modified table.
> # Q4. Definition of $\mu^*$ and $\lambda^*$
> Sorry for the confusion.
>
> $\lambda^*$ and $\mu^*$ should have the same definitions in Table 1 and all theorems in the paper.
>
> $\lambda^*$ and $\mu^*$ denote the contraction rates for continuous-time overdamped Langevin dynamics and underdamped Langevin dynamics respectively. To be more specific, let $x_t$ follow the overdamped (or underdamped) Langevin dynamics initialized at ${x_0=0}$, $\pi_z$ be the invariant distribution, $p_t$ be the marginal distribution $x_t$, then $\lambda^*$ and $\mu^*$ satisfy that
> $$W^2(p_t, \pi_z) \leq C e^{-\lambda^* t/d}, \mbox{ or } W^2(p_t, \pi_z) \leq C e^{-\mu^* t/d},$$ for some constant $C$.
>
> The contraction rates $\mu^*$ and $\lambda^*$ are related to but are not directly the spectral gaps of the Langevin dynamics.
> According to [5], under Assumption A1-A2, one can choose $\lambda^*$ to be the uniform lower bound of the spectral gap (i.e., equation 2.5), and it is proved that
> $$
> {\lambda^*}^{-1} =\exp( \mathcal{O}(d)).
> $$
>
> Similarly, [3, 6] derives contraction results with a low bound of contraction rate
> $$
> {\mu^*}^{-1} = \exp(\mathcal{O}(d)).
> $$

---

> ### Author Response · Authors · 2024-02-15
> **Rebuttal by Authors (Cont.)**
>
> # Q5. Definition of "Low-precision"
> The low-precision training considered in this paper means training the model with gradient and parameters in a low-precision binary computer number format. In particular, we consider the fixed-point low-precision format, following previous works [8, 9] to represent parameters and gradients for all our theorems.
> Theorems 1, 2, 4, 5, 7, and 8 consider stochastic rounding, and Theorems 3, 6, and 9 consider the VC rounding for parameters and stochastic rounding for gradients. We avoid using deterministic rounding due to its bias [10, 11].
> The quantization error $\Delta$ is the gap between any two consecutive representable numbers in fixed-point low-precision format. The larger $\Delta$ is, the fewer numbers the fixed-point format can present. Once we have selected a low-precision level, the quantization error $\Delta$ is decided. For example, if we use 8 bits to represent a number where 1 bit is assigned for the sign, 2 bits for the integer part, and 5 bits for the fractional part, then the gap between two consecutive low-precision numbers is $2^{-5}$, i.e., $\Delta = 2^{-5}$.
> Thus, the quantization error $\Delta$ is not attached to the rounding type.
>
> Our theorems for different stochastic rounding and VC rounding are based on the same $\Delta$ to ensure a fair comparison of their convergence results.
>
> # Q6.  Similarities and differences between these proofs.
> Thanks for the suggestion. In the revision, we added paragraphs that give a sketch of the proofs in each section. We provide a big picture here.
>
> Theorem 1, Theorem 2, and Theorem 3 consider the same non-log-concave target distributions. The difference is that in the algorithm considered by Theorem 1, the gradients are biased because we quantize the parameter before taking the gradients and using those gradients to update the unquantized parameters. Thus in proof of Theorem 1, we need to derive the upper bound given biased gradients. Young's inequality plays an important role in the proof.
> For the algorithms in Theorem 2 and Theorem 3, the gradients are all unbiased. The main difference between the proofs of Theorem 2 and Theorem 3 is the quantization methods.
> The quantization method used in Theorem 3 gives us a bound for the difference between the quantized value and full-precision value. This bound can scale with the learning rate. This fact leads to the advantage of VC rounding (i.e., Theorem 3) over naive stochastic rounding (i.e., Theorem 2).
> Theorems 4, 5, and 6 consider the same algorithms as Theorems 1, 2, and 3 respectively but for strongly log-concave target distributions. The differences among Theorems 4, 5, and 6 are similar to the differences among Theorems 1, 2, and 3. Similar arguments apply to Theorems 7, 8, and 9 which study variants of low-precision SGLD algorithms under non-log-concave target distributions.
>
> # Q7. How to do stochastic rounding in practice
> In practice, to implement stochastic rounding based on the rule (19), the computer still needs a full-precision Unif(0,1) random number generator (note that we ignore the discretization gap between full precision values and real values), then compares this random number with the residual and rounds up if it is smaller otherwise rounds down. This full precision random number generator can be shared for all rounding steps, hence won't affect memory usage too much. For more details about the implementation of stochastic rounding, please refer to [7, 8, 11].
>
> # Q8. $\log(x)<=x^c$
> Yes. As you mentioned, $\log(x) \leq x^c$ is only an asymptotic bound and not always true for any $x > 0$. In our analysis $x = \Delta$ does not converge to zero, so we use the non-asymptotic inequality $log(x)\leq x^{1/e}$.

---

> ### Author Response · Authors · 2024-02-15
> **Rebuttal References**
>
> References:
>
> [1]: Dalalyan A S, Karagulyan A. User-friendly guarantees for the Langevin Monte Carlo with inaccurate gradient. Stochastic Processes and their Applications, 2019, 129(12): 5278-5311.
>
> [2]: Gao X, Gürbüzbalaban M, Zhu L. Global convergence of stochastic gradient Hamiltonian Monte Carlo for nonconvex stochastic optimization: nonasymptotic performance bounds and momentum-based acceleration[J]. Operations Research, 2022, 70(5): 2931-2947.
>
> [3]: Zou D, Xu P, Gu Q. Stochastic gradient Hamiltonian Monte Carlo methods with recursive variance reduction. Advances in Neural Information Processing Systems, 2019, 32,
>
> [4]: Cheng X, Chatterji N S, Bartlett P L, et al. Underdamped Langevin MCMC: A non-asymptotic analysis. Conference on learning theory. PMLR, 2018: 300-323.
>
> [5]: Maxim Raginsky, Alexander Rakhlin, and Matus Telgarsky. Non-convex learning via stochastic gradient Langevin dynamics: a nonasymptotic analysis. In Conference on Learning Theory, pp. 1674–1703. PMLR, 2017.
>
> [6]: Eberle A, Guillin A, Zimmer R. Couplings and quantitative contraction rates for Langevin dynamics. 2019.
>
> [7]: Mikaitis M. Stochastic rounding: Algorithms and hardware accelerator. 2021 International Joint Conference on Neural Networks (IJCNN). IEEE, 2021: 1-6.
>
> [8]: Gupta S, Agrawal A, Gopalakrishnan K, et al. Deep learning with limited numerical precision. International conference on machine learning. PMLR, 2015: 1737-1746.
>
> [9]: Lin D, Talathi S, Annapureddy S. Fixed point quantization of deep convolutional networks. International conference on machine learning. PMLR, 2016: 2849-2858.
>
> [10]: Li, H., De, S., Xu, Z., Studer, C., Samet, H., and Goldstein, T. Training quantized nets: A deeper understanding. Advances in neural information processing systems, 2017
>
> [11]: Yang, G., Zhang, T., Kirichenko, P., Bai, J., Wilson, A. G., and De Sa, C. Swalp: Stochastic weight averaging in low-precision training. International Conference on Machine
> Learning, 2019.
>
> [12]: Croci M, Fasi M, Higham N J, et al. Stochastic rounding: implementation, error analysis and applications[J]. Royal Society Open Science, 2022, 9(3): 211631.

---

> ### Author Response · Authors · 2024-04-05
> **Response from Authors**
>
> Dear Reviewer,
>
> Thank you for your valuable feedback. We have revised the manuscript in line with your suggestions. All the changes are highlighted in blue. We apologize for any delay in submission. The updated paper slightly exceeds the original length limit, and we awaited further guidelines to ensure compliance.
>
> We appreciate your understanding and patience. Your guidance has been a huge help in enhancing the quality of our work.

---

### Review · Reviewer_arQG · 2024-01-28

**Summary Of Contributions:**

The paper studies low-precision sampling algorithms, based on HMC with a stochastic gradient estimator. Theoretical asymptotic convergence guarantees in Wasserstein distance are provided. The complexity of the methods are obtained for a variety of methods. Comparison is performed also experimentally.

**Audience:**

Yes

**Broader Impact Concerns:**

No ethical concerns.

**Claims And Evidence:**

No

**Requested Changes:**

- In (1), the function $\tilde{U}$ and the sequence $\xi_k$ are not introduced.
- The quantization error $\Delta$ is not defined.
- The literature review needs  to be made more complete and the proposed methods need to be put in perspective with the existing work on federated learning and federated sampling.
- The mathematical notation should be better introduced.
- The comparison of the methods does not reflect the theory and is not thus informative. In addition, there are no plots comparing with the "regular" sampling algorithms without quantization.
- For the rest, please refer to the weaknesses section, above.

**Strengths And Weaknesses:**

### Strengths

The authors asymptotic convergence guarantees for all the methods, that allows to compare to each other.

The experimental setting is rather strong, as they apply their methods on simple Neural Networks.

### Weaknesses

- The problem formulation is not clear. The notation is not clear. See the **requested changes** section for specifics.
- It is hard to follow the storyline of the paper, without having the very specific background on low-precision sampling schemes.
- There is a line of work on federated sampling. In these algorithms, compression of the gradients and the iterates plays a crucial role, as communication complexity is the bottleneck. The suggested algorithms SGLDLP-(P,F), are very similar to these methods and the comparison with these methods should be made clearly. See Bibliography.
- In addition, in optimization, direct compression of the gradient and/or the iterate are by far not the most efficient methods in terms of communication(quantization) and iteration complexities. There are methods based on Markov compressors /momentum variance reduction etc. that are much faster and achieve convergence without quantization neighborhood $\Delta$.
- Table 1 is not informative. Using quantization as the compression mechanism, one can save only a fixed portion the memory. That means the memory only multiples by a constant. Thus, if the LP sampling algorithms achieve the same **asymptotic** complexity guarantees as the FP methods, it does not necessarily mean that they improve on them. In order for this comparison be meaningful, the complexities need to be explicit.
- Furthermore the Wasserstein error for all the low-precision methods, depend polynomially on the inverse precision error $\epsilon$, which can be large if the precision is low. This also raises questions about the theoretical efficiency of the methods.
- In addition to the previous point, the use of these methods can justified only if they are compared to the non-compressed LMC,HLMC etc. Again, compressing the gradients, one loses in the accuracy and/or iteration complexity. None of the plots show this comparison of the algorithms.
- The intuition of both Algorithm 1 and 2 is not presented. There is no explanation on why the methods are designed that way.


**Bibliography**
- M. Vono, V. Plassier, A. Durmus, A. Dieuleveut, and E. Moulines. QLSD: Quantised Langevin Stochastic Dynamics for Bayesian Federated Learning . 2022
- W. Deng, Y.-A. Ma, Z. Song, Q. Zhang, and G. Lin. On convergence of federated averaging langevin dynamics. arXiv preprint arXiv:2112.05120, 2021.
- L. Sun, A. Salim, and P. Richtárik. Federated learning with a sampling algorithm under isoperimetry, 2022. URL https://arxiv.org/abs/2206.00920.
- Karagulyan, Avetik, and Peter Richtárik. "ELF: Federated Langevin Algorithms with Primal, Dual and Bidirectional Compression." _arXiv preprint arXiv:2303.04622_ (2023).
- MARINA: Faster non-convex distributed learning with compression. E Gorbunov, KP Burlachenko, Z Li, P Richtárik. *International Conference on Machine Learning, 3788-3798*
- DASHA: Distributed nonconvex optimization with communication compression, optimal oracle complexity, and no client synchronization. A Tyurin, P Richtárik. *In International Conference on Learning Representations. 2023. (ICLR 2023)*

---

> ### Author Response · Authors · 2024-02-15
> **Rebuttal by Authors**
>
> We really appreciate the feedback. Regarding your questions,
> # Q1. Problem and notations not clear
> Sorry for the confusion. We do not introduce a new notation $\tilde{U}$ in the paper. Instead, we abuse the notation a little bit and directly define the term $\nabla
> \tilde{U}$ as an unbiased stochastic estimate of $\nabla U$. To avoid such confusion, we change the notation to $\widetilde{\nabla U}$.
> And $\xi_k \in \mathbf{R}^d$ is a random variable that follows the standard normal distribution. The quantization error $\Delta$ is the gap between any two consecutive representable numbers.
> In the context of fixed-point low-precision number format, if we use 8 bits to represent a number and 1 bit is assigned for the sign, 2 bits for the integer part, and 5 bits for the fractional part, then the gap between two consecutive low-precision numbers is $2^{-5}$, i.e., $\Delta = 2^{-5}$.
> # Q2. Storyline of the paper
> Thank you! We added some motivations, history, and application of low-precision training in the introduction section. We also include some survey papers (e.g. [1, 2]) from where readers without a background can find more detailed information on low-precision training.
>
> To help readers quickly comprehend the main point of low-precision training, we add the following discussion as well:
> The purpose of designing low-precision training is not to achieve low-order memory usage. Reducing the computation complexity and the memory usage to a feasible level (even only by a constant factor) such that it can be implemented and deployed in a wide range of hardware, is of critical importance for modern large-scale data science problems.
>
> # Q3. Similar Works on Federated Learning
>
> Thank you for the suggestion. We included suggested references in the revision.
> However, we would like to point out that the comparison between the convergence result of our work and the federated MCMC sampling works is unfair.
>
> Under the Federated Learning framework, the main bottleneck is the communication bandwidth but not the local memory usage, hence Federated Learning MCMC sampling methods only compress the gradients at the communication stage and the induced performance loss can be compensated by local training using full-precision algorithmic.
>
> For the problem we want to solve, in both the training and inference stages, low-precision algorithmic must be used. Therefore, the Federated Learning algorithms mentioned by the reviewer are not applicable. Under our setting, there is inevitably a loss in performance, i.e., the intrinsic gap between the best low-precision sampler and true posterior sampler.
> Our goal is to design a better learning algorithm to improve the computation efficiency while minimizing the performance loss.
>
> #  Q4. Table 1 and Complexity Improvement
>
> First of all, we want to clarify here all the algorithms presented in Table 1 are low-precision algorithms. The "Full-precision gradient accumulators'' mean that we keep a full-precision copy of parameters, and we still quantize the parameters before calculating the gradients. In other papers, this method is also called the "binary connected'' low-precision algorithm.
> On the other hand, the ``Low-precision gradient accumulators'' means that we do not keep a full-precision copy of the parameters. All the parameters and gradients are stored in a low-precision format in the memory.
> Please refer to equations (5) and (6) and their differences.
> Therefore, this table presents the results of two variations of the low-precision sampler, rather than comparing the performance between the full-precision sampler and low-precision sampler.
>
> We want to emphasize that the purpose of designing low-precision training is not to achieve low-order memory usage. Reducing the computation complexity and the memory usage to a feasible level (even only by a constant factor) such that it can be implemented and deployed in a wide range of hardware, is of critical importance for modern large-scale data science problems.
>
> Full precision algorithms are considered to be the gold standard, and practitioners should use them when hardware allows. In other words, the purpose of our paper is not to design low-precision algorithms that are better than full-precision algorithms. We aim to minimize the performance loss in sampling caused by precision constraints when the computational environment does not allow full precision algorithms.
>
> Also, the main comparison purpose of Table 1 is to compare SGLD and SGHMC in the context of low-precision algorithmic. The table shows that under the low-precision setting, SGHMC still converges faster than SGLD, and the convergence result has a better dependence on quantization error $\Delta$. This is because the update rule of SGHMC can be seen as the moving average of noisy gradients which averages out the stochastic quantization error.

---

> ### Author Response · Authors · 2024-02-15
> **Rebuttal by Authors (Cont.)**
>
> # Q5. Wasserstein bound not going to zero
> First of all, we want to point out that the 2-Wasserstein distance between the low-precision sampler and the target distribution cannot be zero, since the low-precision sampler follows some discrete distribution (i.e., its support is discrete representable values) and the target distribution is, in general, a continuous distribution. As $\Delta$ increases, there are fewer representable values for the low-precision sampler, and its approximation to the true target gets worse. In other words, it is expected that the upper bound of W2 distance grows with $\Delta$.
>
> As for our non-converging result w.r.t. $\epsilon$ for non-log-concave target distributions, we would like to point out that, even for full-precision sampling algorithms, the best non-asymptotic convergence result in the 2-Wasserstein distance (e.g., [3,4,5]) all contain a $\log(\epsilon^{-1})$ factor, and diverge as $\epsilon\rightarrow0$.
>
> It is our future work to improve current convergence results, and we want to emphasize the non-triviality. Even for the optimization problem (not the sampling problem we study in this paper), in the presence of saddle point and local minima, it is shown the problem is NP-hard. Please see the references [6,7,8].
>
> # Q6. Comparison with full-precision HMC and LMC
>
> That is a good suggestion, and we add the full-precision results to the plot as well. Please see figure 1.
> https://anonymous.4open.science/r/Low-precisionSGHMC-D6E7/fig_1.pdf
>
> From the figure, unsurprisingly the full-precision algorithms outperform their low-precision counterparts. But with long enough iterations the performance gap between SGHMC/SGLD and SGHMCLP-F/SGLDLP-F converges toward zero, especially under the blocking floating point low-precision representation format. Note that in plot b) of Figure 1, SGHMCLP-F is even better than full-precision SGLD in block floating low-precision representation format, supporting our argument that the underdamped Langevin dynamic is more suitable for low-precision training.
>
> # Q7. Intuition of both Algorithms 1 and 2
>
> We correct the typos in Algorithm 1, see via the link.
> https://anonymous.4open.science/r/Low-precisionSGHMC-D6E7/alg_1.pdf
>
> Algorithm 1 is a practical version of the three different types of low-precision SGHMC updates proposed in our main text, i.e. equations (5), (6), and (8). Additional components in the algorithm include
> (i) How to compute the stochastic gradient via forward/back-propagation (steps colored by red in the Algorithm box). Due to the low-precision nature of the algorithm, we also quantize all intermediate results along the propagation process by proper quantizers $Q_A$ and $Q_E$.
> (ii) Additional optional scale/re-scale step (colored by blue in the Algorithm box). The reason for adding this step is that: in practice, we found that the momentum term $v_k$ tends to be close to $0$. When $v_k$ is represented in the low-precision fixed-point format, the information carried by $v_k$ is lost since the low-precision fixed-point is loose around 0, and only 2 or 3 bits are used representing the momentum (i.e., the other bits are wasted). With this observation, we store a scaled-up momentum to fully utilize all bits, thus the information carried will be kept in an optimal way.
>
> Algorithm 2 is proposed by [9].
> The rationale behind Algorithm 2 is that:
> if we directly quantize the SGLD update result, i.e. the mean shift plus a Gaussian noise variable, it essentially introduces an additional quantization noise to the sample, leading to a larger sampling variance. Instead of quantizing the mean shift plus Gaussian noise, we can first quantize the mean shift, then plus a low-precision discrete random variable. In this way, we guarantee the sampler yields low-precision values and have the freedom to design the variance of the low-precision discrete random variable, such that overall sampling variance (i.e., variance due to stochastic round and low-precision discrete random variable) matches the idea sampling variance of full-precision Langevin update.

---

> ### Author Response · Authors · 2024-02-15
> **Rebuttal References**
>
> References:
>
> [1]: Lei Deng, Guoqi Li, Song Han, Luping Shi, and Yuan Xie. Model compression and hardware acceleration for neural networks: A comprehensive survey. Proceedings of the IEEE, 108(4):485–532, 2020.
>
> [2]: Tailin Liang, John Glossner, Lei Wang, Shaobo Shi, and Xiaotong Zhang. Pruning and quantization for deep neural network acceleration: A survey. Neurocomputing, 461:370–403, 2021.
>
> [3]: Maxim Raginsky, Alexander Rakhlin, and Matus Telgarsky. Non-convex learning via stochastic gradient langevin dynamics: a nonasymptotic analysis. In Conference on Learning Theory, pp. 1674–1703. PMLR, 2017.
>
> [4]: Difan Zou, Pan Xu, and Quanquan Gu. Stochastic gradient hamiltonian monte carlo methods with recursive variance reduction. Advances in Neural Information Processing Systems, 32, 2019.
>
> [5]: Xuefeng Gao, Mert Gürbüzbalaban, and Lingjiong Zhu. Global convergence of stochastic gradient hamiltonian monte carlo for nonconvex stochastic optimization: Nonasymptotic performance bounds and momentum-based acceleration. Operations Research, 70(5):2931–2947, 2022.
>
> [6]: Danilova M, Dvurechensky P, Gasnikov A, et al. Recent theoretical advances in non-convex optimization High-Dimensional Optimization and Probability: With a View Towards Data Science. Cham: Springer International Publishing, 2022: 79-163.
>
> [7]: Murty K G, Kabadi S N. Some NP-complete problems in quadratic and nonlinear programming[R]. 1985.
>
> [8]: Nesterov Y. Lectures on convex optimization[M]. Berlin: Springer, 2018.
>
> [9]: Zhang, Ruqi, Andrew Gordon Wilson, and Christopher De Sa. "Low-precision stochastic gradient Langevin dynamics." International Conference on Machine Learning. PMLR, 2022.

---

> ### Author Response · Authors · 2024-04-05
> **Response from Authors**
>
> Dear Reviewer,
>
> Thank you for your valuable feedback. We have revised the manuscript in line with your suggestions. All the changes are highlighted in blue. We apologize for any delay in submission. The updated paper slightly exceeds the original length limit, and we awaited further guidelines to ensure compliance.
>
> We appreciate your understanding and patience. Your guidance has been a huge help in enhancing the quality of our work.

---

### Review · Reviewer_Xnd4 · 2024-01-31

**Summary Of Contributions:**

The authors studied a collection of both full precision and low precision versions of stochastic gradient based samplers, provided various convergence guarantees in terms of the 2-Wasserstein metric.

**Audience:**

Yes

**Claims And Evidence:**

Yes

**Requested Changes:**

### On the Vacuous Nature of the Guarantees

Firstly, I would like the authors to discuss the connection between the theory and the applications here. In particular, if the theoretical results rely on spectral gap constants $\lambda^*, \mu^*$, which are known to have exponential dependence on dimension, then we would expect the results to be consistently vacuous for large dimensional problems. For example, in Theorem 1, we would require $(\mu^*)^{-2} = O( e^{O(d)} )$ number of steps to reach the convergence result. A quick Google search led me to find that ResNet-18 has about 11 million trainable parameters, which would require an astronomical number of steps to achieve this guarantee. In fact, even for 50 parameters, this value would be close to number of atoms in the universe at 10^50.

Given this enormous gap, I don't believe the application here is so relevant for the theory. Can the authors provide some discussion towards where this theory can be contributing towards if the dependence on dimension here is so critical?

### On the Definition of Spectral Gaps

Two key quantities used in all of the results are $\mu^*, \lambda^*$, which the authors have called the "spectral gap" constants, and occasionally "concentration parameter", "concentration number", or "concentration rate." I hope the author can first of all introduce a consistent name for these important constants.

More importantly, I believe it is absolutely critical to precisely define these constants, which is nowhere found within this article. In fact, it is not even found in the Zou et al. (2019) citation. Not only do these coefficients often end up having an exponential dependence on dimension, I believe the typical definition of spectral gap does not exist for the underdamped Langevin diffusion! Here's a quote from Villani's seminal work on Hypocoercivity:

"In many fields of applied mathematics, one is led to study dissipative evolution equations involving (i) a degenerate dissipative operator, and (ii) a conservative operator presenting certain symmetry properties, such that the combination of both operators implies convergence to a uniquely determined equilibrium state. Typically, the dissipative part is not coercive, in the sense that **it does not admit a spectral gap**; instead, it may possess a huge kernel, which is not stable under the action of the conservative part."

Instead, what I believe the authors are referring to is an exponential contraction rate of a Lyapunov function, which implies an exponential convergence in terms of the Wasserstein distance. I believe this is a **particularly important detail**, and it should not be swept under the rug. So here, I request the authors make the definitions precise, use a consistent naming convention, and provide a related discussion to the vacuous nature on dimension.

### Wasserstein Distance is Non-Vanishing

I found the statement in Theorem 1 somewhat peculiar, where the guarantee is stated to reach the goal of

$$ \mathcal{W}_2 \leq \tilde O \left( \epsilon + \tilde A \sqrt{ \log \frac{1}{\epsilon} } \right) . $$

However, this expression on the right hand side when treated as a function of $\epsilon$, cannot reach zero ($\log \frac{1}{\epsilon}$ increases as we decreases $\epsilon$). In my opinion, this is strange, because the point of a convergence guarantee is that we should be able to minimize the error due to our sampler being an approximation, so we should be able to arbitrarily improve our sample quality at the cost of additional computational power.

This also implies the stated result in Table 1 is misleading, as additional dependence on $\sigma$ is hidden, which cannot be driven to zero, unlike $\Delta$ as a quantization error. I believe this should be fixed in order for a correct representation of your results.

On a fundamental level, I am questioning whether or not Assumption 3 is the best assumption on stochastic gradients, which is likely the core reason that the authors are dealing with this issue. First of all, I don't think this is true for minibatch SGD with shifted quadratic loss, where the variance scales with $\|x\|^2$. But that aside, I believe it's fair to assume that the stochastic gradient is unbiased, e.g. $\mathbb{E} \nabla \tilde U = \mathbb{E} \nabla U$. In which case, we would have that the discretization SGLD (or the underdamped version) will converge to the **unbiased** Langevin dynamics as the step size $\eta\to 0$. This heuristically implies that the sampler should be able to approximate the Langevin diffusion arbitrarily well, and hence $\mathcal{W}_2$ should be driven to zero with a decreased step size.

Can the authors incorporate this condition into their proof, so that we can achieve a result that actually drives the Wasserstein distance to zero? I understand this may not be easy to do, if that's the case, can the authors instead provide a discussion on which part of the proof is difficult to adapt? I believe this is a rather unfortunate set of assumptions and results passed down, and should be fixed at some point.

**Strengths And Weaknesses:**

Strengths
 - There is a large collection of results.

Weaknesses
 - It is unclear to me that the applications are convincing motivation for the theory, given the exponential dependence on dimension are hidden.
 - Several mathematical details should be made precise, in particular with regards with spectral gaps.
 - The guarantees are given in a form where the achieved Wasserstein distance cannot be driven to zero in e.g. Theorem 1.

---

> ### Author Response · Authors · 2024-02-15
> **Rebuttal by Authors**
>
> Your insights are greatly appreciated. Regarding your questions,
> # Q1. Dependence on Dimension $d$ and definition of $\mu^*$ and $\lambda^*$
> Thanks for the question. To make things more clear,
>
> $\lambda^*$ and $\mu^*$ denote the contraction rates for continuous-time overdamped Langevin dynamics and underdamped Langevin dynamics respectively. To be more specific, let $x_t$ follow the overdamped (or underdamped) Langevin dynamics initialized at ${x_0=0}$, $\pi_z$ be the invariant distribution, $p_t$ be the marginal distribution $x_t$, then % by Proposition 9 of reference [5], the following is true
> $\lambda^*$ and $\mu^*$ satisfy that
> $$W^2(p_t, \pi_z) \leq C e^{-\lambda^* t/d}, \mbox{ or } W^2(p_t, \pi_z) \leq C e^{-\mu^* t/d},$$
> for some constant $C$.
>
> The contraction rates $\mu^*$ and $\lambda^*$ are related to but are not directly the spectral gaps of the Langevin dynamics.
> According to  reference [5], under Assumption A1-A2, one can choose $\lambda^*$ to be the uniform lower bound of the spectral gap (i.e., equation 2.5), and it is proved that
> $$
> {\lambda^*}^{-1} =\exp( \mathcal{O}(d)).
> $$
>
> Similarly, reference [3, 6] derives contraction results with a low bound of contraction rate
> $$
> {\mu^*}^{-1} = \exp(\mathcal{O}(d)).
> $$
>
> The dependence of $\mu^*$ on $d$ (i.e., the above equation) is merely the worst-case lower bound derived by prior works. This exponential dependency is not universally sharp. Therefore, one should view $\mu^*$ as the intrinsic sampling difficulty of the target distribution is Langevin dynamics, rather than an exponentially decaying function of $d$.
>
> We revised the statements in the paper to avoid any possible confusion.
>
> # Q2. Wasserstein Distance is Non-Vanishing, and Table 1 should not hide the term $\sigma^2$
> First of all, we want to point out that the 2-Wasserstein distance between the low-precision sample and the target distribution cannot be zero, since the low-precision sample follows some discrete distribution (i.e., its support is discrete representable values) and the target distribution is in general continuous distribution. As $\Delta$ increases, there are fewer representable values for the low-precision sampler, and its approximation to the true target gets worse. In other words, it is expected that the upper bound of W2 distance grows with $\Delta$.
>
> As for our non-converging result w.r.t. $\epsilon$ for non-log-concave target distributions, we would like to point out that, even for full-precision sample algorithms, the best non-asymptotic convergence result in the 2-Wasserstein distance (e.g., References [1,2,3]) all contain a $\log(\epsilon^{-1})$ factor, and diverge as $\epsilon\rightarrow0$.
>
> It is our future work to improve current convergence results, and we want to emphasize the non-triviality. Even for the optimization problem (not the sampling problem we study in this paper), in the presence of saddle point and local minima, it is shown the problem is NP-hard. Please see the references [4,5,6].
>
> The purpose of Table 1 is to compare HMC and SGLD under the low-precision setting, thus the focus is on the rate dependency on the quantization error $\Delta$. That is, to answer the question of which algorithm adapts well under low precision algorithmic. Therefore, we hide $\sigma^2$ in Table 1, such that the comparison w.r.t. $\Delta$ is explicit.
>
> # Q3. Assumptions not reasonable
> First, to avoid confusion (as pointed out by reviewer arQG), in the revision, we will use the notation $\widetilde{\nabla U}$ to replace  $\nabla \tilde{U}$ as the estimation of the true gradient $\nabla U$.
>
> Second, $\widetilde{\nabla U}$ is by definition unbiased (refer to Section 2.3), hence the unbiasedness is not stated as a formal assumption in the paper, and our proofs do depend on the unbiasedness of  $\widetilde{\nabla U}$.
>
> Following the suggestion, we can replace Assumption 3 with a weaker one:
> $$
> \mathbf{E}\|\widetilde{\nabla U}(x)-\nabla U(x)\|_2^2 \leq \delta ( M^2\|x\|_2^2 + B^2).
> $$
> Similar variance assumptions can be found in the following references [1, 3].
> This change won't alter our key results, and we will update our theorem statements and proofs accordingly in the camera-ready version.

---

> > ### Comment · Reviewer_Xnd4 · 2024-03-14
> > **Response**
> >
> > Apologies for my late response, I was very occupied by personal matters and really could not get to reading this. I will now respond to all the points above.
> >
> > ### Spectral Gap
> >
> > I see that you have fixed most of the terminology (albeit spectral gap still occurs a couple of times), however I want to emphasize it is **important to precisely define** what $\mu^*$ and $\lambda^*$ are. It is not acceptable to refer the definition to another paper, especially when it's not precisely defined in there either!
> >
> > I believe there are many things we can overlook here for submissions at TMLR, but I still maintain that we need all submissions to be **mathematically precise** to be considered acceptable.
> >
> > ### On Dimensional Dependence
> >
> > Actually, I disagree that the dimension dependence is not tight. In particular, we have a fairly good understanding of metastability in the low temperature regime. See for example: Menz and Schlichting (2012) linked here: https://arxiv.org/abs/1202.1510
> >
> > In particular, the results here suggest asymptotically, the bottleneck is on the order of $e^{O(\beta)}$ for some inverse temperature $\beta$. And if $\beta$ is simply proportional to dimension, which is currently speculated to be necessary to recover this proof, then we do effectively have an exponential dependence on dimension.
> >
> > My intention with bringing up this concern is not to discredit this work, but rather wanting the authors to provide a fair discussion in the paper. This way, the future readers will understand the existing issues with this approach of studying Langevin dynamics.
> >
> > ### Wasserstein Error Driven to Zero
> >
> > Let us ignore the quantization error for now. As long as the quantization regime is such that it is unbiased, i.e. $\mathbb{E} \text{ Stochastic Gradient} = \text{True Gradient}$, then we know asymptotically the limit of the finite time Markov chain converges to the true diffusion. See for example Theorem 17.25 and 17.28 from Kallenberg's Foundations of Modern Probability for a precise statement. The intuition is quite clear, basically there's a law of large numbers happening at a very small range of time. So all the noises from the stochastic gradient should be vanishing in the LLN.
> >
> > So what is the role that $\epsilon$ plays here? And why can't the noise be driven to zero? Once again, I'm not asking the authors to fix this result if it's difficult, I simply want the authors to have a proper discuss about this confusion. If a new reader to this field comes in, I can imagine this being an obvious confusion: why is that the convergence guarantee cannot be driven to zero given more computation power?
> >
> > ### Summary
> >
> > At this point, I would like the authors to make the requested changes, including adding a precise definition of $\lambda^*$ and $\mu^*$, and adding the appropriate discussions for the points raised above.

---

> > > ### Author Response · Authors · 2024-04-04
> > > **Response by Authors**
> > >
> > > Thanks for your comments and we added the revision regarding to your concern as the following.
> > >
> > > ## Spectral Gap
> > > To maintain conciseness in the main text, we introduce $\lambda^*$ and $\mu^*$ as the contraction rate of over/under-damped Langevin continuous process as follows.
> > > Definition:
> > > Let $\lambda^*$ and $\mu^*$ denote the contraction rates for continuous-time overdamped Langevin dynamics and underdamped Langevin dynamics respectively. In other words, let $x_t$ follow the overdamped (or underdamped) Langevin dynamics initialized at ${x_0=0}$, $\pi_z$ be the invariant distribution, $p_t$ be the marginal distribution $x_t$, then % by Proposition 9 of reference [5], the following is true $\lambda^*$ and $\mu^*$ satisfy
> > > $$\mathcal{W}_2^2(p_t, \pi_z) \leq C e^{-\lambda^* t/d}, \mbox{or} \mathcal{W}_2^2(p_t, \pi_z) \leq C e^{-\mu^* t/d},$$
> > > for some constant $C$.
> > >
> > > In the appendix, we add a more detailed discussion as follows:
> > >
> > > According to reference [10], under Assumptions 1 and 2, one can choose $\lambda^*$ to be the uniform lower bound of the contraction rate, i.e.,
> > >  $$\lambda^* :=  \inf \\{\frac{\int_{R^d}\|\nabla g\|^2 dp^*}{\int_{R^d} g^2 dp^*}:g\in \mathcal{C}^1(R^d) \cap L^2 (p^*), g=0, \int_{R^d} g dp^*=0 \\}, $$
> > > which satisfied
> > > $$
> > > \frac{1}{\lambda^*} = \frac{2}{m_2(d+b)} + \frac{4C(d+b)}{m_2}exp\left(\frac{2}{m_2}(M+B)(b+d)+(A+B)\right),
> > > $$
> > > where $A$, $B$ denote bounds such that $|U(0)| \leq A, \|\nabla \widetilde{U}(0)\| \leq B$. In other words, asymptotically w.r.t. the dimension, we have ${\lambda^*}^{-1} =\exp( \mathcal{O}(d))$.
> > > Similarly,  reference [5,11] derives a contraction rate as
> > > $$
> > >         \mu^* = \frac{2d}{768\gamma e^{\Lambda}}\min\left\\{\lambda Mue^{\Lambda}, \Lambda^{1/2}Mu, \gamma\Lambda^{1/2} \right\\},
> > > $$
> > > where the constants are defined as:
> > >     \begin{align*}
> > >         &\lambda = \frac{2m_2}{4M+u^{-1}\gamma^2} \\
> > >         &\Lambda = \frac{12(1+2\alpha+2\alpha^2)(d+\mathcal{A})Mu}{5\gamma^2\lambda(1-2\lambda)} \\
> > >         &\mathcal{A} = \frac{2m_2(U(x^*)+M\|x^*\|^2)}{4M+u^{-1}\gamma^2}+\frac{b}{2}.
> > >     \end{align*}
> > > Note that the above rate also satisfies ${\mu^*}^{-1} = \exp(\mathcal{O}(d)).$
> > > ## Dimensional Dependence
> > > In general, the contraction rates exponentially depend on the dimension $d$. For example, two popular approaches to analyze the Wasserstein distance convergence property is the couplings method (Reference [2,5]) and Bakry–Émery method based on which the exponential convergence of the kinetic Fokker–Planck equation is proved (Reference [1,3,4]).  Unfortunately, both rates lead to exponential dependency on dimension in general. It raises a crucial open question of whether restricted models can exhibit improved dimensional dependence. While overdamped Langevin diffusions have seen corresponding advancements, as evidenced in references [6, 7], analogous progress for underdamped Langevin diffusions remains underexplored.
> > >
> > > ## Wasserstein Error Driven to Zero
> > >
> > > We agree with the reviewers' opinion. In our proof, the stochastic error does not average out, and this is merely due to mathematical technicality. We added the following statement in our revision.
> > > The non-convergence of our Wasserstein upper bound is due to the accumulation of stochastic gradient noise and stochastic discretion error. Conceptually, these random errors may average out over iterations when the iteration number increases to infinity (i.e., the law of large numbers), as in the classical ergodic theory of the Markov chain (Theorem 17.25, 17.28 of Reference [12]). However, our mathematical tools lead to an upper bound that involves some weighted summation of the norm of these random errors over iterations rather than the summation of these random errors. Under strongly log-concave target distributions with no discretion error, this sum is bounded as $t\rightarrow \infty$ and proportional to the stepsize, allowing for a sufficiently small step size to zero the bound (Reference [8, 9]). However, for general cases, this sum grows to infinity. It is yet an open question to sharpen this type of analysis.

---

> > > ### Author Response · Authors · 2024-04-04
> > > **Reponse References**
> > >
> > > [1]: Bakry D, Émery M. Diffusions hypercontractives Séminaire de Probabilités XIX 1983/84: Proceedings. Berlin, Heidelberg: Springer Berlin Heidelberg, 2006: 177-206.
> > >
> > > [2]: Dalalyan A S, Riou-Durand L. On sampling from a log-concave density using kinetic Langevin diffusions. 2020
> > >
> > > [3]: Baudoin F. Wasserstein contraction properties for hypoelliptic diffusions. arXiv preprint arXiv:1602.04177, 2016.
> > >
> > > [4]: Baudoin, F.: Bakry–émery meet Villani. J. Funct. Anal. 273(7), 2275–2291, 2017.
> > >
> > > [5]: Eberle, A., Guillin, A., Zimmer, R.: Couplings and quantitative contraction rates for Langevin dynamics. Ann. Probab. 47(4), 1982–2010, 2019.
> > >
> > > [6]: Eberle A. Reflection couplings and contraction rates for diffusions. Probability theory and related fields, 2016, 166: 851-886.
> > >
> > > [7]: Zimmer R. Explicit contraction rates for a class of degenerate and infinite-dimensional diffusions. Stochastics and Partial Differential Equations: Analysis and Computations, 2017, 5: 368-399.
> > >
> > > [8]: Dalalyan A S, Karagulyan A. User-friendly guarantees for the Langevin Monte Carlo with inaccurate gradient. Stochastic Processes and their Applications, 2019, 129(12): 5278-5311.
> > >
> > > [9]: Cheng X, Chatterji N S, Bartlett P L, et al. Underdamped Langevin MCMC: A non-asymptotic analysis. Conference on learning theory. PMLR, 2018: 300-323.
> > >
> > > [10]: Raginsky M, Rakhlin A, Telgarsky M. Non-convex learning via stochastic gradient langevin dynamics: a nonasymptotic analysis. Conference on Learning Theory. PMLR, 2017: 1674-1703.
> > >
> > > [11]: Zou D, Xu P, Gu Q. Stochastic gradient Hamiltonian Monte Carlo methods with recursive variance reduction. Advances in Neural Information Processing Systems, 2019, 32.
> > >
> > > [12]: Kallenberg O, Kallenberg O. Foundations of modern probability. New York: springer, 1997.

---

> > > > ### Comment · Reviewer_Xnd4 · 2024-04-04
> > > > **Response**
> > > >
> > > > Thank you for the detailed response. I believe these are helpful towards me supporting the acceptance of this paper. However, I still would like to follow up on several recommended changes. It would be helpful if you highlight these changes in a different colour for the next revision.
> > > >
> > > > 1. Can you incorporate the definitions you listed above into the main text? I believe these are critical definitions.
> > > >
> > > > 2. There are still two references of $\mu^*$ as the ``spectral gap'' of the underdamped Langevin dynamics. In context of the previous discussion, this name is inappropriate, as the typical definition of spectral gap refers to the generator, and the gap doesn't exist.
> > > >
> > > > 3. Can you add the dimension dependence discussion into the main paper? In particular, at where the contraction rates are being discussed. The goal here is to avoid providing a misleading representation for this line of work of sampling from non-log-concave distributions.
> > > >
> > > > 4. I don't see the discussion on the non-convergence of Wasserstein distance added yet.

---

> > > > > ### Author Response · Authors · 2024-04-04
> > > > > **Response from Authors**
> > > > >
> > > > > Thank you for your valuable feedback, which has significantly enhanced our paper's clarity and presentation. For your ease of review, we have uploaded the updated manuscript to the following anonymous link, with all changes marked in blue for easy identification: https://anonymous.4open.science/r/Low-precisionSGHMC-717C/TMLR_Revision.pdf.
> > > > >
> > > > > The revised manuscript now exceeds the initial length requirement, and, due to a lack of specific guidance from the editors regarding revisions, we have not yet submitted it through the portal. Upon receiving further instructions, we are prepared to integrate the discussion on dimension dependence (now in Appendix C) into the main text as suggested.

---

> ### Author Response · Authors · 2024-02-15
> **Rebuttal References**
>
> References:
>
>
> [1]: Maxim Raginsky, Alexander Rakhlin, and Matus Telgarsky. Non-convex learning via stochastic gradient langevin dynamics: a nonasymptotic analysis. In Conference on Learning Theory, pp. 1674–1703. PMLR, 2017.
>
> [2]: Difan Zou, Pan Xu, and Quanquan Gu. Stochastic gradient hamiltonian monte carlo methods with recursive variance reduction. Advances in Neural Information Processing Systems, 32, 2019.
>
> [3]: Xuefeng Gao, Mert Gürbüzbalaban, and Lingjiong Zhu. Global convergence of stochastic gradient hamiltonian monte carlo for nonconvex stochastic optimization: Nonasymptotic performance bounds and momentum-based acceleration. Operations Research, 70(5):2931–2947, 2022.
>
> [4]: Danilova M, Dvurechensky P, Gasnikov A, et al. Recent theoretical advances in non-convex optimization High-Dimensional Optimization and Probability: With a View Towards Data Science. Cham: Springer International Publishing, 2022: 79-163.
>
> [5]: Murty K G, Kabadi S N. Some NP-complete problems in quadratic and nonlinear programming[R]. 1985.
>
> [6]: Nesterov Y. Lectures on convex optimization[M]. Berlin: Springer, 2018.
>
> [7]: Zhang, Ruqi, Andrew Gordon Wilson, and Christopher De Sa. "Low-precision stochastic gradient Langevin dynamics." International Conference on Machine Learning. PMLR, 2022.
>
> [8]: Eberle A, Guillin A, Zimmer R. Couplings and quantitative contraction rates for Langevin dynamics. 2019.

---

> ### Author Response · Authors · 2024-04-05
> **Response from Authors**
>
> Dear Reviewer,
>
> Thank you for your valuable feedback. We have revised the manuscript in line with your suggestions and apologize for any delay in submission. The updated paper slightly exceeds the original length limit, and we awaited further guidelines to ensure compliance.
>
> Following your suggestions, we have incorporated the discussion on dimension dependence into the main body of the paper.
>
> We appreciate your understanding and patience. Your guidance has been a huge help in enhancing the quality of our work.

---

> > ### Comment · Reviewer_Xnd4 · 2024-04-05
> > **Response**
> >
> > Thank you for the revision. I have now finished reading the revision and agree with the current state of the manuscript, and will support accepting this paper.

---

### Comment · Reviewer_hv7o · 2023-12-27

I'm sorry, this article is far from my research area, and I have been ill recently, so I haven't had the time to review it. Please find someone else to review it. I apologize for any inconvenience caused.

---

### Author Response · Authors · 2024-02-15
**Summary Response for Authors**

We thank all reviewers for their constructive comments and suggestions, which help us improve the quality and representation of the paper.
To summarize the rebuttal,
* we explicitly point out our technical contribution over existing results (i.e., Zhang et.al (2022)).
We consider not only strongly log-concave target distribution but also non-log-concave target distribution whereas Zhang et.al (2022) only consider strongly log-concave cases.
We introduce an intermediate process to handle the different quantization errors compared to the low-precision SGLD algorithm.
* We clarify the common confusion about the definition of $\lambda^*$ and $\mu^*$. $\lambda^*$ and $\mu^*$ denote the contraction rates of the continuous time overdamped equation (13) and underdamped Langevin dynamics equation (10), representing the intrinsic sampling difficulty via Langevin dynamics.
* We provide a clarification for our "low-precision" setting and the notation $\Delta$. Our Theorems consider the fixed-point low-precision format, and $\Delta$ is the gap between any two consecutive representable numbers.
* We included additional empirical comparisons between full-precision and low-precision SGHMC/SGLD. Please see the update in the following link:
https://anonymous.4open.science/r/Low-precisionSGHMC-D6E7/fig_1.pdf .
* Finally, we discussed the common concern of two reviewers that the derived Wasserstein distance bound does not converge to zero. We would like to point out that, even for full-precision sampling algorithms, the best non-asymptotic convergence result in the 2-Wasserstein distance (e.g., References [1,2,3]) all contain a $\log(\epsilon^{-1})$ factor, and diverge as $\epsilon\rightarrow0$. And to the best of our knowledge, our paper gives the first-ever convergence result for low-precision SGHMC algorithms.

We would like also to summarize the contributions of our work. This is the first work to provide a comprehensive study of low-precision SGHMC sampling algorithms. We consider both non-log-concave and strongly log-concave target distributions and propose three variants of low-precision SGHMC algorithms. We conduct a thorough theoretical analysis and compare the Wasserstein convergence rates for low-precision SGHMC and low-precision SGLD. We find that low-precision SGHMC converges faster and is more robust to low-precision error $\Delta$ because of the moving average effect brought by the update rules of SGHMC. Our theoretical results are supported by our simulations and real-data applications.

---

### Decision · Action_Editor_VCxV · 2024-04-06

**Recommendation:** Accept as is

**Comment:**

This paper studies low-precision sampling algorithms, focusing on stochastic gradient Hamiltonian Monte Carlo (SGHMC) with both low-precision and full-precision gradient accumulators. It provides theoretical convergence guarantees of these algorithms in the Wasserstein distance for both strongly log-concave and non-log-concave distributions. The numerical experiments show that low-precision SGHMC can outperform stochastic gradient Langevin dynamics (SGLD) in certain scenarios.

Reviewers expressed concerns about the clear picture of the paper, and suggest adding the definitions of lambda* and mu*, explaining that the Wasserstein distance bound is non-vanishing, and spectral gaps can be exponential. I agree with reviewers that the limitation of the results shall be clearly explained. After the authors modified accordingly, reviewers have agreed to accept the paper. I suggest acceptance.

**Audience:**

Yes. It is of interest to people working on stochastic sampling, as it studies a low-percision version of the algorithm stochastic gradient Hamiltonian Monte Carlo (SGHMC)  which achieves better bound than existing bound.

**Claims And Evidence:**

Yes. The paper claimed that SGHMC achieves quadratic improvement over SGLD, and the theoretical results validates the claim.